# In silico typing maps the natural diversity of *Escherichia coli* transporter-dependent capsules

Serotyping identifies bacterial variants based on surface antigens, traditionally using antibody-based assays, but has been increasingly replaced by in silico methods that infer serotypes from genomic sequences for faster, scalable and more reproducible analyses. However, traditional *Escherichia coli* capsule serotyping has largely fallen out of use since the 1990s, leaving gaps in our knowledge of capsule genetics, diversity, distribution and epidemiology. As capsules influence bacterial interactions with phages, host immune systems and the environment, this gap limits our understanding of *E. coli* ecology and pathogenicity as well as vaccine and diagnostic development. Here we established a definitive genotype–serotype map for 35 serologically identified and structurally characterized transporter-dependent capsules. We then surveyed 37,723 *E. coli* genomes, cataloguing 85 transporter-dependent capsule types (K-types), including 55 types that were not part of the reference collection. We leveraged this catalogue to develop a hidden Markov model-based in silico serotyping tool, kTYPr, and applied it to curated sets of 24,015 *E. coli* genomes and 2,762 metagenome-assembled genomes spanning diverse environmental and clinical sources. We found previously uncharacterized K-types enriched in undersampled environments and associated with *E. coli* disease. This study expands our understanding of *E. coli* surface structures, supporting efforts for precision targeting with phage therapy or vaccines.

Surface-exposed antigens drive extensive structural variation in *Escherichia coli* populations. While the O-antigen glycan repertoire is well characterized[1,2], our knowledge of the diversity of *E. coli* capsular polysaccharides remains limited[3]. Capsular polysaccharides are considered virulence factors, with described roles in innate and adaptive immune evasion[4,5], and are critical targets for protective antibodies and therapeutic bacteriophages[5,6]. During the past century, classical serology identified 80 different capsule serotypes in *E. coli*[7]. However, capsular polysaccharides are only weakly immunogenic and some mimic host structures[8,9], limiting the usefulness of classical cross-absorption and agglutination-based serological analyses[7]. Therefore, capsules have been detected via countercurrent immunoelectrophoresis, a slow and inefficient precipitin-line-based approach[10].

With the advent of PCR-based *E. coli* typing of O- and H-antigens, research into capsule serology has been stalled, with the last new serotype described in 1977[11].

*E. coli* capsules are classified into four groups, based on their biosynthesis and genetics[12]. Groups 1 and 4 are Wzx/Wzy-dependent polysaccharides, whereas groups 2 and 3 consist of polymers synthesized onto a phosphatidylglycerol oligo-β-3-deoxy-D-*manno*-octulosonic acid (Kdo) structure in the inner leaflet of the inner membrane and exported intact to the cell surface via an ABC transporter (Fig. 1a)[13]. The proteins required for oligo-β-Kdo synthesis (encoded by *kpsC* and *kpsS*)[13–15] and capsule export (encoded by *kpsE*, *kpsD*, *kpsM* and *kpsT*) are well conserved[16]. K-type-specific proteins extend the oligo-β-Kdo primer with serotype-specific sugars and other constituents, producing

✉e-mail: smiravet@ethz.ch; ecacace@ethz.ch; emma.slack@immune.engineering; ssunagawa@ethz.ch; timkeys@baxiva.com

the K antigen. It has been postulated that groups 2 and 3 dominate among extraintestinal pathovars, while groups 1 and 4 prevail in intestinal pathovars[17]. However, the lack of appropriate experimental and computational tools to identify capsule serotypes has hindered a comprehensive evaluation of this distribution[18].

With the increasing accessibility of whole-genome sequencing (WGS) and the escalating antibiotic resistance crisis, there is an urgent need for reliable genomic-based methods to identify capsular polysaccharides. Previous studies have proposed computational approaches for *E. coli* serotyping from genomic data, relying on basic local alignment search tool (BLAST)[1,19] or hidden Markov models (HMMs)[20], and have linked specific gene clusters to capsule serotypes[21–24], but the lack of a comprehensive genotype–phenotype map has prevented the assignment of novel K-types, leaving a critical knowledge gap.

Here we report major progress in closing this gap for transporter-dependent capsular polysaccharides. We recently established a genotype–serotype map for 35 group 2 and 3 K antigens belonging to the *E. coli* capsule reference strain collection[7,25], which we refined to 30 genetically distinguishable types. We extended this diversity to all group 2 and 3 capsule biosynthesis loci in 37,723 *E. coli* genomes, identifying 55 previously uncharacterized types. By integrating protein structural similarity, we functionally annotated biosynthetic loci, enabling structural inference for previously uncharacterized K-types and providing a corrected structural model for K6, supported by mass spectrometry. We leveraged these data to develop kTYPr, an HMM-based tool for scalable and sensitive capsule typing, which we applied to a globally distributed collection of 24,015 genomes and 2,762 stool-derived metagenome-assembled genomes (MAGs). This revealed a substantial overlap in K-type distributions between gut-colonizing and invasive *E. coli* strains and associations with human disease and understudied ecological niches. We envision that this integrated genotype–serotype reference and bioinformatic tool will enable microbiologists, epidemiologists and vaccinologists to explore the role of *E. coli* capsules in infection, transmission and protection from disease.

## Results

### A genotype–serotype map for established capsule types

To establish a definitive K antigen genotype–serotype map, we sequenced the reference strain collection established at the World Health Organization's Collaborative Centre for Research and Reference on *Escherichia* (Supplementary Table 1)[7,25]. Genetic information for four transporter-dependent K serotypes was obtained from publicly available sequence databases (K15 and K74) or from strains from the Culture Collection University of Gothenburg (K2a and K22). We identified a *kps* locus, including all common essential genes (*kpsEDCSMT*), in 35 of the 70 strains (Supplementary Table 1), confirming their designation as ABC transporter-dependent capsule types (groups 2 and 3)[12,26]. By contrast, these genes were not identified in strains designated as producers of Wzx/Wzy-dependent capsule types (groups 1 and 4).

In some reference strains, we observed identical or near-identical *kps* loci (Extended Data Fig. 1), corresponding to cross-reactive K serotypes[7] that share polysaccharide backbones with variable nonstoichiometric modifications[27–31]. For example, a single base insertion disrupting a putative acetyltransferase gene in K2a may underpin the distinction between K2a and its acetylated version K2ab (Extended Data Fig. 1a)[32]. Similarly, two missense mutations differentiate the candidate acetyltransferase genes in K13 and K23 (Extended Data Fig. 1b), potentially reflecting differing acetylation[31]. In other cases, the genes relevant to serotype differentiation may be found outside of the capsule gene cluster (Extended Data Fig. 1c,d)[32]. To cautiously assign these closely related K-types in which genetic differentiation is not possible, we grouped them under shared designations (K2a and K2ab = K2; K13 and K23 = K13_K23; K18a, K18ab and K22 = K18_K22; K54 and K96 = K96). Accordingly, our current kTYPr catalogue defines

30 K-types corresponding to the 35 reference transporter-dependent serotypes (Supplementary Table 1).

### An extended catalogue of transporter-dependent capsules

Although studies have identified more than the 35 established *kps* loci in *E. coli* genomes[6,19,32,33], the lack of a complete genotype–serotype map has prevented the assignment of new K-types. To bridge this gap, we catalogued *kps* loci from 37,723 *E. coli* genomes from the National Center for Biotechnology Information Reference Sequence Database (NCBI RefSeq; Fig. 1b and Methods) by first using *kpsC* as a marker gene, owing to its role in biosynthesis of the unique oligo-β-Kdo anchor common to this class of capsules[13]. We then dereplicated sequences extending 30 kb upstream and downstream of *kpsC*, filtered for the presence of essential genes (*kpsEDCSMT*) and progressively trimmed the sequences to a presumed locus based on the organization of conserved genes, functional annotations and identification of common flanking genes. The set of unique *kps* loci was further refined using a protein catalogue, excluding transposase-related open-reading frames (ORFs), with each set of closely related sequences (≥90% identity and coverage) used to build a protein HMM. In an iterative process, these HMMs were used to identify the *kps* locus flanking *kpsC* (±30 kb) and rebuilt, providing additional sequence diversity. A final set of 85 distinct *kps* loci was defined based on unique gene content, making our HMM-based K-type definitions robust to genetic rearrangements, and insertions or deletions in non-coding sequences. A comparison of the catalogue with the reference strains led us to assign 55 K-types, named between K104 and K163, that were not previously catalogued. This resulted in an updated catalogue of 85 transporter-dependent K-types (Supplementary Table 2 and Methods).

### Defining a fourth lineage of transporter-dependent capsules

The historically designated group 2 and 3 capsules represent two distinct genetic lineages[34]. A third group of plasmid-borne *kps* clusters (designated 3B) was identified as a recent acquisition in *E. coli*[6,32]. To investigate the evolutionary relationships across the 85 transporter-dependent K-types (Supplementary Table 2), we constructed phylogenetic trees based on the conserved capsule biosynthesis genes *kpsEDCSMT* (Fig. 1c). Trees for the two most conserved proteins, the outer membrane pore (KpsD) and the periplasmic adaptor (KpsE), revealed four distinct clades. Three correspond to the known lineages, and one to a previously unrecognized group that we designated 2B. We assigned each *kps* locus to a group (2A, 2B, 3A or 3B) based on the KpsDE phylogeny (Supplementary Table 2). While these four groups do not represent independently evolving lineages, they generally correlate with the presence of *kpsF* and *kpsU*, which encode enzymes involved in cytidine-5′-monophosphate (CMP)-Kdo biosynthesis and are redundant in the *E. coli* genome (Fig. 1a,d), and with the organization of conserved genes (Fig. 1e). One exception is the K150 cluster, which is assigned to group 3B but includes *kpsF* and *kpsU* genes typical of group 2A.

Phylogenetic trees of other conserved proteins highlight rearrangements and exchanges within and between groups. Phylogenetic trees of KpsC and KpsS share the KpsDE architecture with two known exceptions: *kpsS* genes in the 3A lineage appear to derive from the 2A lineage[32]; KpsC and KpsS from K15 form distinct branches incongruent with the other lineages, suggesting a possible origin outside *E. coli*[35,36]. Phylogenetic trees of KpsM and KpsT revealed higher diversity and evidence of exchanges between group 2A and 2B, consistent with frequent rearrangements of *kpsM*, *kpsT* and K-specific genes (for example, K7, K51, K52, K104, K105–K123, K94–K128 and K127–K156), with the result that group 2A clusters are not always flanked by *kpsM* (Fig. 2).

### Protein structure predictions support K-specific annotations

To characterize the catalogue's functional repertoire, we annotated serotype-specific genes identifying conserved domains

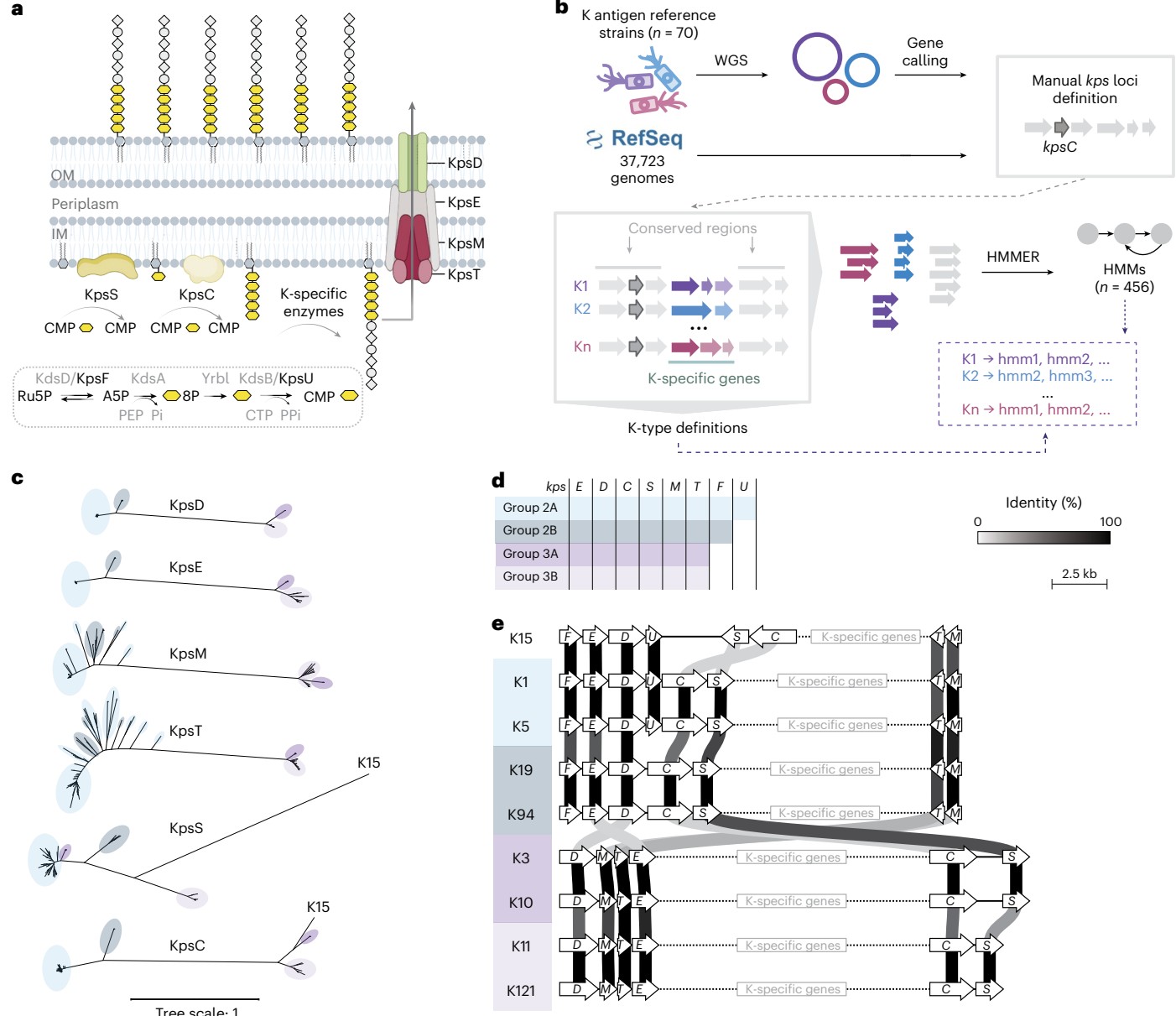

**Fig. 1 | Overview of capsule biosynthesis, cataloguing and *kps* lineages.**
**a**, Transporter-dependent capsule biosynthesis and export, showing conserved Kps proteins. Kdo residues are in yellow. OM, outer membrane; IM, inner membrane; CMP, cytidine 5′-monophosphate; CTP cytidine triphosphate; Ru5P, D-ribulose 5-phosphate; D-arabinose 5-phosphate, A5P; PEP, phosphoenol-pyruvate; Pi, inorganic phosphate; PPi, inorganic pyrophosphate. **b**, Workflow for building the capsule gene cluster catalogue by integrating *kps* loci from K antigen reference strains (*n* = 35) and RefSeq (*n* = 37,723), through WGS of reference strains, locus extraction, ORF identification and protein sequence clustering, and K-specific HMMs generation (Methods). **c**, Unrooted phylogenetic trees for essential proteins for capsule biogenesis KpsEDCSMT (Methods). Leaf colours correspond to lineages 2A, 2B, 3A and 3B. **d**, Presence (coloured) and absence (blank) of conserved capsule biosynthesis genes in the four *kps* lineages. **e**, Gene cluster arrangements representative of each lineage, including the K15 exception. Conserved genes (white arrows) are to scale; K-specific genes are not. Greyscale lines connecting genes indicate protein sequence identity.

(Supplementary Table 3) and glycosyltransferase (GT) families according to the Carbohydrate-Active enZYmes Database (CAZy; Supplementary Table 4)[37,38]. Sequence similarity allowed us to assign CAZy families to 50.5% (137 out of 271) of the GT domains. We further predicted structural similarities to Protein Data Bank entries, using AlphaFold2 (ref. [39]) and Foldseek[40] (Supplementary Table 5 and Methods), assigning a further 134 domains as putative GTs. Altogether, we assigned putative functions to 479 of 515 serotype-specific ORFs (Fig. 2), often in the absence of detectable protein sequence similarity. For example, 29 domains have predicted structural similarity to the GT99 (*n* = 19) and GT107 (*n* = 10) families[13], known for retaining Kdo transferase activity. This correlates with a Kdo constituent in the

reported polysaccharide structures (K13_K23, K14, K16, K19, K20, K74, K95 and K97) and suggests that GT99- and GT107-like domains catalyse the respective Kdo linkages (although the α-linked Kdo in K16 and K74 would require an inverting Kdo transferase).

Grouping *kps* clusters with related K-specific genes revealed 18 sets of K-types predicted to encode structurally related polysaccharides with similar backbone composition (Fig. 2). A total of 13 of these sets include at least one cluster with known polysaccharide, facilitating structural inferences for unknown polysaccharides. Notably, one set (K13_K23 to K19; Fig. 2) shares a large ORF (>1,100 amino acids) with little detected sequence similarity but three domains identified by predicted structural similarity: a GT99-like Kdo transferase and a dual

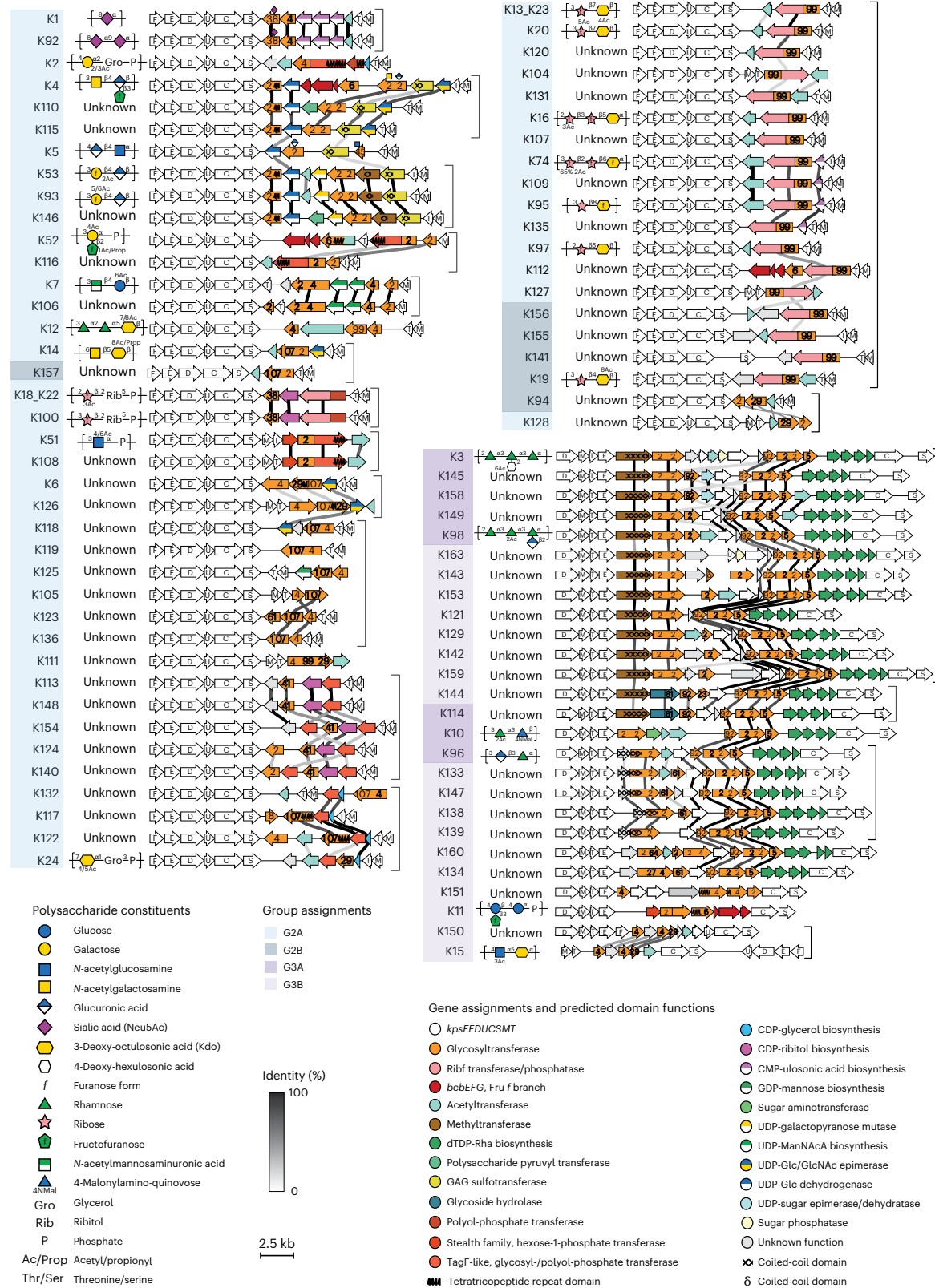

**Fig. 2 | Catalogue of *E. coli* transporter-dependent capsule gene clusters.** Conserved biosynthetic and export genes appear as white arrows. K-specific genes are coloured by known or predicted function. GT activity is indicated by the transferred GTs (above). CAZy family is indicated in black text on the gene. Putative GTs identified via structural similarity are labelled with their closest match's CAZy family in grey text on the gene. Greyscale lines indicate pairwise amino acid sequence identities (30–100%) between adjacent ORFs. Sequence similarity for *kpsFEDUCSMT* is omitted for clarity. Gene clusters are grouped by function (grey brackets). Polysaccharide repeat units, where known, are depicted according to symbol nomenclature for glycans (SNFG) conventions[91]. For accession numbers, gene cluster coordinates, and nucleotide and amino acid sequence identity, see Supplementary Tables 1, 2 and 6. dTDP, deoxythymidine diphosphate; GAG, glycosaminoglycan; CDP, cytidine diphosphate; CMP, cytidine monophosphate; GDP, guanosine diphosphate; UDP, uridine diphosphate; ManNAcA, N-acetylmannosaminuronic acid; Glc, glucose; GlcNAc, N-acetylglucosamine.

domain ribofuranosyltransferase–phosphatase, recently shown to incorporate ribofuranose (Rib*f*) in bacterial polysaccharides[41,42], consistent with Kdo- and Rib*f*-containing polysaccharides published for seven members of the group (Supplementary Table 1). This example shows how predicted structural similarity can reveal functional relationships obscured by sequence divergence and generate structural hypotheses for unknown capsular polysaccharides.

### Glycan structures clarify gene-function inconsistencies

Predicted gene cluster functions generally aligned with published polysaccharide structures (Fig. 2), except for two cases: K6 and K16.

The K6 gene cluster is reported to produce a Rib*f*–Rib*f*–Kdo repeat unit[43] but lacks predicted Rib*f*T domains. To clarify this inconsistency, we purified capsular polysaccharide from seven reference strains reported to produce Rib*f*–Kdo polysaccharides and analysed repeat unit composition by matrix-assisted laser desorption/ionization mass spectrometry (MALDI–MS) of permethylated glycans (Methods and Extended Data Fig. 2). Consistent with our analysis of the gene clusters, we observed Rib*f*–Kdo repeat units in all types except K6, which contained m/z species consistent with a Kdo homopolymer. Such a putative poly-Kdo structure is consistent with an ORF encoding a dual GT29–GT107 domain protein, where both GT families are known to transfer ulosonic acids, such as Kdo, from CMP-activated donor substrates. As we cannot be certain that the current K6-type strain is identical to the strain historically used in structural glycobiology studies[43], we conservatively assigned K6 an unknown polysaccharide structure (Fig. 2) pending further structural investigation.

K16 is reported to produce a Rib*f*–Rib*f*–Kdo trisaccharide repeat unit[44], but its predicted domains are identical to several clusters reported to produce a Rib*f*–Kdo disaccharide repeat (for example K13, K19, K20; Fig. 2 and Supplementary Tables 1 and 6). We confirmed with MALDI–MS both the reported disaccharide repeat unit (for K13, K19, K20, K23 and K97) and trisaccharide repeat unit (for K16; Extended Data Fig. 2). This discrepancy in K16 may be explained by an additional Rib*f*T encoded outside of the capsule gene cluster, or a bifunctional Rib*f*T capable of transferring Rib*f* to both Kdo and Rib*f* acceptors. This latter hypothesis would be supported by known single-domain bifunctional GTs, such as those in *Neisseria meningitidis* (serogroup L)[45] and *E. coli* (K92)[46].

### kTYPr as a genotype-to-serotype profiler in *E. coli*

We used the assembled *kps* HMM catalogue to develop kTYPr, an in silico capsule typing software applicable to both genomic and metagenomic data (Fig. 3a and Methods). kTYPr screens genomes for *kpsC* gene presence and processes positive ones with the full set of HMMs to identify conserved and K-type-specific genes that meet HMM-specific score cut-offs. K-types are defined by the presence of all essential genes (*kpsEDCSMT*) and K-type-specific gene sets. When criteria for multiple K-types are met, the decision is based on the highest accumulated bit scores. This allows to distinguish between closely related gene clusters (for example, K51 and K108) and prevents the incorrect assignment of gene subsets that are part of multiple K-types (for example, K121 is a subset of K153 and K129; Fig. 2). Bit score values are reported for all K-types considered, together with the complete set of gene hits against the HMMs for further inspection or parametrization, a GenBank file of the extracted *kps* locus and a clinker-based[47] visualization of its similarity to the predicted or closest K-type cluster (Fig. 3a).

We evaluated the performance of kTYPr on the source dataset (37,723 RefSeq *E. coli* genomes). Of these, 25.5% (*n* = 9,620) were assigned one of the transporter-dependent K-types in our catalogue based on the presence of all essential (*kpsEDCSMT*) and K-type-specific genes. Of the 25,602 genomes not assigned to a transporter-dependent K-type using default settings (Methods), 99.9% (*n* = 25,575 of 25,602) lacked a complete set of essential and K-specific genes (as well as *kpsC* in the majority of cases, *n* = 25,448). Only 27 of these unassigned

genomes contained a complete *kpsEDCSMT* set. All other genomes were assigned a K-type, but lacked complete conserved and/or specific sets of genes (*n* = 2,501, of which 1,081 had incomplete versions of both sets) (Extended Data Fig. 3a). These incomplete cases may represent disrupted gene clusters, genes missing in draft genomes or uncataloged *kps* cluster diversity. While all 85 K-types were identified in the source dataset, 10 were identified only once or twice (Extended Data Fig. 3b), indicating that our catalogue has deeply sampled the diversity of *E. coli kps* loci present in RefSeq, but RefSeq has not sampled their diversity in nature.

We compared the kTYPr catalogue and EC-K-Typing DB (v6.11.2025)[19] as implemented in the BLAST-based tool Kaptive[48]. The majority of *kps* loci in the two catalogues were equivalent in terms of functional gene content (*n* = 62 of 90 KL types in the EC-K-Typing DB and *n* = 62 of 85 K-types in the kTYPr catalogue; Supplementary Table 7 and Fig. 1). A total of 23 K-types were provided exclusively by kTYPr, while 6 found only in the EC-K-Typing DB catalogue were extremely rare in RefSeq (5 of 37,723). Of the 30 phenotypically established K-types identified by kTYPr, 7 were misidentified by Kaptive (Supplementary Fig. 1 and Tables 8–11), showing the value of both the genotype–phenotype map and sensitivity of our HMM-based approach. In addition, we identified 22 disrupted loci assigned as previously uncharacterized K-types in the EC-K-Typing DB that differ from the parental locus only by ORF disruptions caused by premature stop codons resulting from single-nucleotide polymorphisms (SNPs) or short indels, thus without genuine sequence divergence (Supplementary Fig. 2 and Table 8). By globally comparing the structure and function of all serotype-specific ORFs, we were able to identify such cases and avoid defining new K-types based on remnant fragments of functional proteins.

### Previously unknown K-types are enriched in understudied hosts and environments

To investigate the distribution of group 2 and 3 capsules across environments, hosts and body sites in humans, we used kTYPr to profile a collection of 32,043 metadata-curated, globally distributed genomes from NCBI (Methods). To reduce genomic redundancy due to clonal strains, or repeated isolation of strains from an individual over time, we dereplicated all genomes coupling a 99% average nucleotide identity (ANI) cut-off with metadata information (host type, sample type, geography, health state) and bacterial characteristics (sequence type; O-, H- and K-type), resulting in 23,188 representative genomes. We further extended this dataset with a collection of 827 *E. coli* genomes from urinary tract infection (UTI) cases, including 260 associated with invasive disease (Methods and Supplementary Table 12)[49].

The resulting collection (*n* = 24,015; Supplementary Table 13) was biased towards human hosts (48.2%; Fig. 3b) and westernized geographical areas (49.8% of genomes from Europe and North America; Extended Data Fig. 4a). A complete *kps* locus was detected in 26% of the genomes (*n* = 6,240), with higher proportions in humans (37.3%), pets (27.8%) and human-associated environments (32.4%) (Fig. 3b). Most of the assigned transporter-dependent capsules belonged to group 2 (91.9% of genomes, *n* = 5,754), of which only 22 genomes belonged to group 2B. A total of 486 genomes (7.8%) harboured group 3 capsules, with 395 and 91 genomes in groups 3A and 3B, respectively (Fig. 3c). Humans, human-associated environments and pets showed correlated K-type profiles (Fig. 3d and Extended Data Fig. 4b,c), whereas livestock and wild animals were similar, in agreement with previous reports of genetically distinct *E. coli* lineages between humans and livestock[50]. Animal hosts that are well represented in the collection correlate according to their K-type profiles beyond geographical groups (Extended Data Fig. 4c).

While 81.8% of the K-types in studied human hosts and associated environments were identified in the serological era, previously unknown K-types (K104–K163) were more frequent in understudied environments, such as non-domesticated animals (Fig. 3d,e), and more

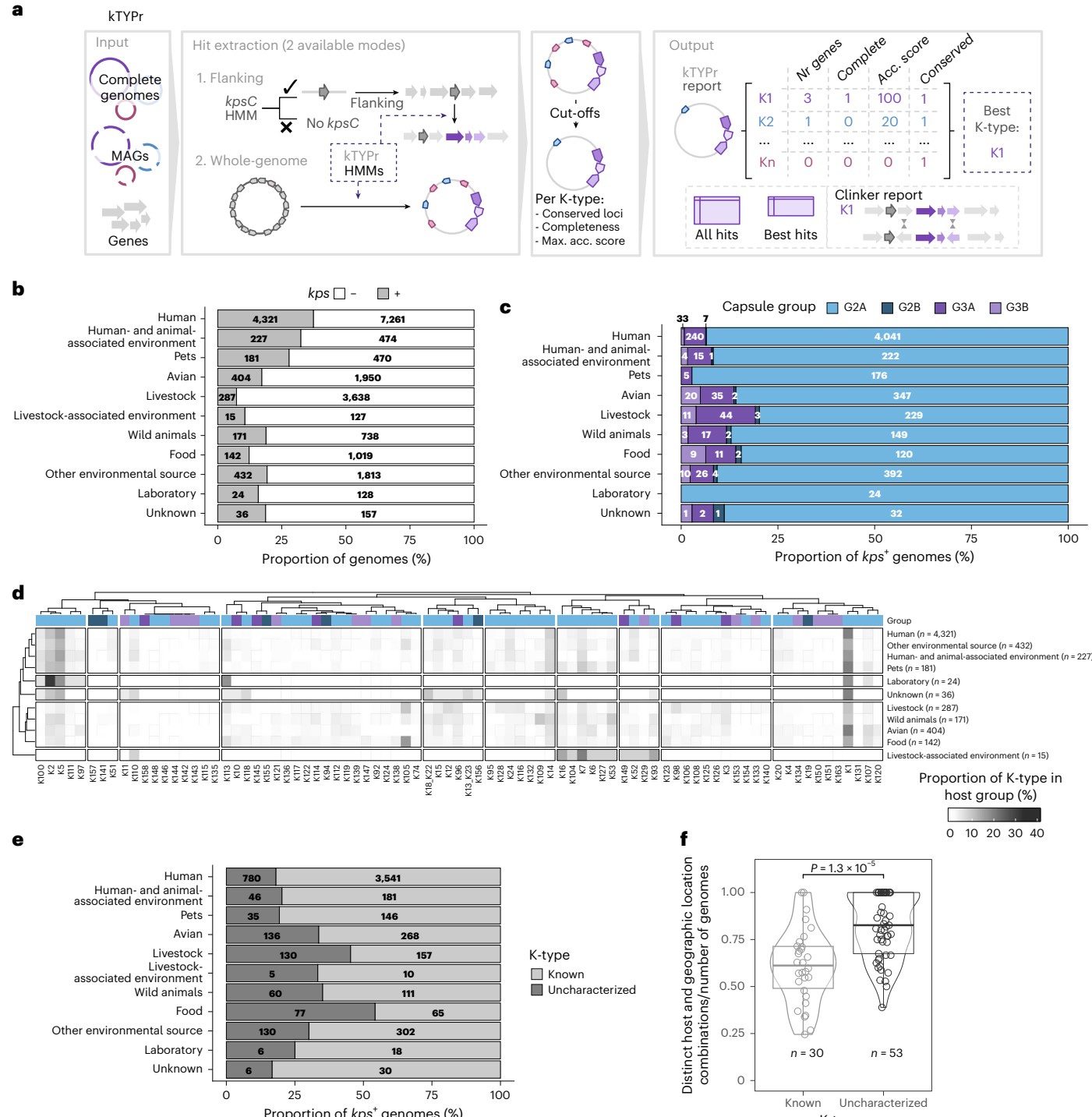

**Fig. 3 | kTYPr workflow and K-type diversity across hosts and environments.**
**a**, Schematic representation of the kTYPr tool, highlighting allowed inputs
(complete genomes, MAGs or genes) and two hit extraction modes (flanking
and whole-genome). Once the custom cut-offs are applied, the hits are processed to
produce a report output including, for each input genome, the accumulated bit
scores and completeness metrics for all K-types profiled, with the 'best match'
as the most complete K-type with the highest accumulated bit score. The output
includes all gene hits against the HMM collection, a formatted GenBank and a
clinker visualization of the similarity between the genome and the best hit(s)
(Methods). Nr genes, number of genes; Acc. score, accumulated bit score.
**b**, Fraction of *kps*-positive genomes (defined as containing a complete *kps* locus;
Methods) across hosts and environments. Absolute counts are indicated within

the bars. **c**, Capsule group proportion across hosts and environments. Only
*kps*-positive genomes are considered. Absolute counts are indicated within the
bars. **d**, K-type proportion in hosts and environments. Only *kps*-positive genomes
were considered. K-types and hosts are clustered according to Spearman
correlation. K-type groups are colour-coded as in **c**. **e**, Proportion of previously
uncharacterized and known K-types across hosts and environments. Only
*kps*-positive genomes are considered. **f**, Number of distinct hosts and geographic
locations divided by the number of genomes for each K-type. The *P* value between
known and previously uncharacterized K-types is shown (two-sided Wilcoxon
test); box limits correspond to the first and third quartiles, with the median
marked and whiskers extending to the most extreme data points up to 1.5 times
the interquartile range. Data point counts are indicated below the box plots.

diverse in terms of hosts and geography (Fig. 3f). The most frequent K-types overall were K1, K5, K2 and K14 (Extended Data Fig. 4d). This ranking reflected the bias of the collection towards human samples and, except for K1, was not conserved in other environmental and geographical niches (Extended Data Fig. 4e): for example, K105 and K109 were the most frequent types in food (15.5%) and wild animals (8.2%), respectively (Extended Data Fig. 4b). Overall, our approach unveiled a much higher diversity of transporter-dependent capsules than previously known from highly sampled, human-associated environments.

## K-type diversity in human health and disease

Consistent with previous reports[51], 64.4% of *E. coli* genomes from phylogroups B2, D and F harboured group 2 K-types (Extended Data Fig. 5a). This was driven by human samples (Fig. 4a), in which these phylogroups are associated with extraintestinal pathogenic *E. coli* (ExPEC) (Extended Data Fig. 5b)[52]. Because most *E. coli* genomes derive from clinical isolates, we have limited understanding of K-type prevalence and distribution in asymptomatic individuals. To address this gap, we profiled globally distributed *E. coli* MAGs from the stools of healthy humans from 25 published studies (*n* = 2,762 of which 921 were *kps* positive; Extended Data Fig. 5c and Supplementary Table 14)[53].

Although group 2 capsules were also more frequent in MAGs belonging to phylogroups B2, D and F, their proportion was higher in extraintestinal disease (for example, isolates from blood or cerebrospinal fluid) (Fig. 4b and Extended Data Fig. 5d). Phylogroups B2, D and F had also the highest K-type diversity, higher in invasive disease than in asymptomatic carriage, except for phylogroups B1 and B2-2 (Extended Data Fig. 5e–g). K-type profiles were more similar among MAGs and genomes from healthy individuals, patients with diseases not associated with *E. coli* and patients with intestinal pathogenic *E. coli* (InPEC), whereas genomes from UTIs, bloodstream infections (BSIs) and meningitis showed distinct separation (Fig. 4c). Transporter-dependent capsules were less common in InPEC (12.4%) compared with extraintestinal disease (52.7% in UTIs, 61.4% in BSI, 86.7% for meningitis), and with stool-derived MAGs (33.3%) or genomes (28.4%) from healthy individuals (Extended Data Fig. 5h). Altogether, while consistent with previous reports of group 2 K-types in ExPEC[54], our analysis revealed the extensive presence of typical ExPEC K-types (for example, K1, K2, K5)[17,18] in the gut microbiome of asymptomatic individuals by overcoming limitations of isolate-based databases.

The most frequent K-types (K1, K2, K5, K14) also showed the highest sequence type (ST) diversity (Fig. 4d). Several ExPEC-associated lineages (for example, ST95, ST73 and ST127) were dominated by single

capsule types, consistent with surface antigen adaptation to host immune pressure[55,56]. Conversely, ST131, despite being highly clonal at the core genome level[57], showed high K-type diversity, supporting capsule plasticity as part of its epidemiological success[58]. Ecological generalists (for example, ST10, ST38, ST69, ST405) showed extensive diversity, with multiple K-types shared across phylogenetically distant STs, indicating recurrent *kps* locus exchange[59] (Fig. 4e,f).

At the population level, K-type composition was strongly but incompletely associated with ST within phylogroups, even in genomes with incomplete K-type gene clusters, indicating lineage-level structuring beyond a broad phylogenetic background (Cramér's $V$ = 0.43–0.44; Fig. 4g). Capsule composition similarity between STs was more strongly associated with ecological similarity (Mantel test, Spearman's $\rho$ = 0.33, $P = 1 \times 10^{-4}$, 9,999 permutations) than with genomic background similarity (Spearman's $\rho$ = 0.14, $P = 0.011$; Fig. 4h and Methods). These results suggest that K-type diversity across the *E. coli* population is shaped by the combined forces of lineage-specific selection and horizontal *kps* locus exchange, aligning more tightly with ecological niche than with overall genomic relatedness. In agreement with the low frequency of transporter-dependent capsules in InPECs (Extended Data Fig. 5h), we did not detect these K-types in typical InPEC serotypes (for example, O157:H7 and O104:H4). These strains may still harbour group 1 or 4 capsules not detected by kTYPr. Conversely, typical ExPEC serotypes (for example, O25:H4, O1:H7) had the highest K-type diversity and included the O-types with the highest capsule prevalence (Extended Data Fig. 6a).

O and K antigen co-occurrence may also reveal functional interdependencies between glycan biosynthetic gene clusters: for example, K12 polysaccharide contains rhamnose[17,60], yet its *kps* cluster lacks the corresponding rhamnose-biosynthesis genes found in other rhamnose-containing K-types (Fig. 2). We hypothesized that these K-types instead harness the O-antigen's rhamnose-biosynthesis machinery. Consistent with this, K12 was exclusively found with rhamnose-containing O-types (O1, O4, O13/O135, O51, O18, O132) (Extended Data Fig. 6b). This shows how kTYPr can inform hypotheses on functional relationships between bacterial surface antigens with potential clinical relevance.

To quantify the association of K-types with invasive disease compared with asymptomatic carriage, we used a multivariable logistic regression model, with the invasive phenotype (defined as isolation from blood or cerebrospinal fluid in patients with BSIs or meningitis) as a binary outcome (Methods). The model confirmed previously proposed associations, including K2 and K14 (refs. 61,62). The K-types

**Fig. 4 | K-type diversity in human health and disease. a**, Proportion of capsule groups across phylogroups in *E. coli* complete genomes from humans (*n* = 11,536). Genomes to which a phylogroup could not be assigned were not considered (*n* = 20). Groups are colour coded as in Fig. 3c, with grey indicating *kps*-negative genomes. **b**, Proportion of capsule groups in *E. coli* genomes from humans across clinical categories (Supplementary Tables 12 and 13) and in asymptomatic carriers (MAGs from the stool of healthy individuals; Supplementary Table 14). Groups are colour coded as in Fig. 3c, with grey indicating *kps*-negative genomes. Non-*E. coli*-associated diseases are grouped as 'other disease'. **c**, Principal coordinates analysis (PCoA) of K-type profiles from the clinical categories in **b**. The PCoA was based on a Bray–Curtis dissimilarity matrix calculated from the proportion of K-types in each group. Axis labels indicate the percentage of variation explained, calculated from positive eigenvalues only. **d**, Number of distinct STs and genomes for each K-type in *E. coli* complete genomes (*n* = 4,170) and MAGs (*n* = 749) from human hosts. Only genomes with complete conserved and specific K-type genes and with assigned STs were considered. **e**, Shannon diversity of K-types for each ST. The same data as in **a** were used, excluding STs with less than 20 genomes (*n* = 3,942). **f**, K-type relative frequencies within each ST. The same data as in **e** were considered (*n* = 3,942). **g**, Association between K-type and ST quantified using Cramér's *V* excluding (*n* = 3,942) or including genomes with incomplete K-type conserved

and/or specific genes (*n* = 4,700). The observed association was compared with a permutation-based null model in which K-types were shuffled within phylogroups. Only STs assigned to at least 20 genomes were considered. **h**, Comparison between Jaccard similarity matrices between all ST pairs based on K-type composition, ecological diversity and genomic background (according to 99% ANI cluster membership). Human-associated complete genomes with annotated 99% ANI clusters, and STs at least 10 genomes, 4 K-types, 2 ecological niches (defined as number of unique hosts, isolation and geographical sources) and 2 99% ANI clusters were considered (*n* = 2,154). **i**, Odds ratio (OR) from a multivariable logistic regression model evaluating the association of K-types with invasiveness. The model adjusted for O-type, H-type, phylogroup, ST, host age, gender and geographic group, based on 4,923 *E. coli* genomes from distinct isolates and individuals including asymptomatic carriage (783 complete genomes and 2,762 MAGs; Supplementary Tables 13 and 14) and invasive *E. coli*-associated disease (1,378 genomes from isolates from blood or cerebrospinal fluid, Supplementary Tables 12 and 13). K-types with ≤5 isolates in either carriage or invasive group were grouped as 'Other'. Rare covariate levels (O-type <15; H-type and ST <10) were aggregated to ensure model stability (Methods). OR = 1 is indicated (dashed line). Data points are colour coded according to statistical significance, determined using two-sided Wald *z*-tests. Error bars represent 95% Wald confidence intervals (Supplementary Table 15).

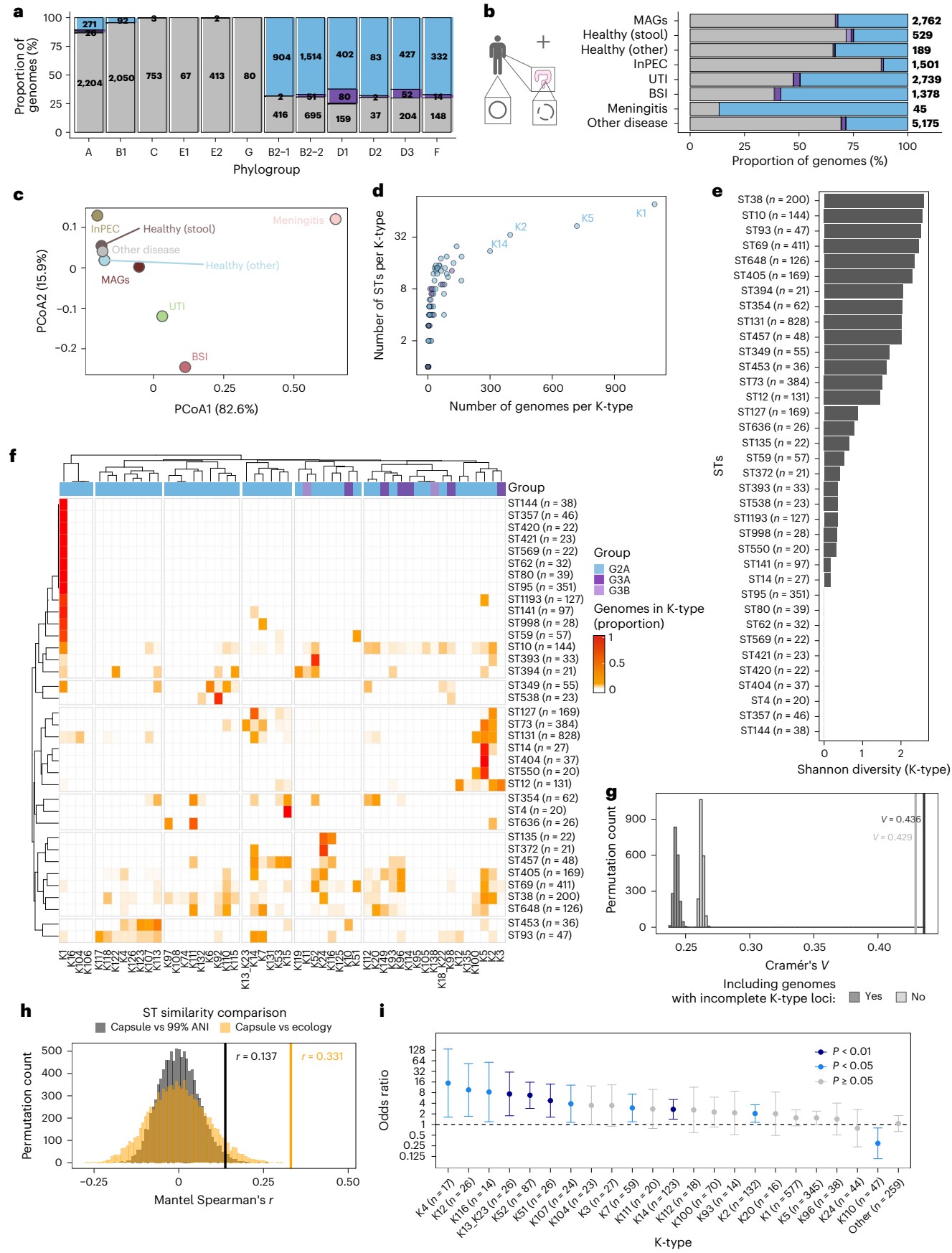

with the strongest association with invasiveness were K4, K12, K116, K13_K23, K52 and K51. Of the nine K-types that were significantly and positively associated with invasiveness, five were recently reported[19]. Of the four previously unknown types with invasive potential, two were defined in our study (K107, K116). In addition, the previously unknown K110 was negatively associated with invasiveness, suggesting a potential relationship with *E. coli* commensal lifestyle (Fig. 4i and Supplementary Table 15). Overall, this highlights a complex interplay between capsule diversity and human disease that warrants investigation in larger datasets and controlled mechanistic studies.

## Discussion

Historical classification of capsular polysaccharides through serology and bacteriophage sensitivity is limited in resolution, scalability and reproducibility[7,10]. On the basis of a ground-truth database including all established *E. coli* K antigens[25], and an extended catalogue of *kps* clusters derived from >37,000 *E. coli* genomes, we provide a robust, scalable, genome-driven approach to study *E. coli* transporter-dependent capsules.

Our catalogue of 85 K-type loci includes 55 previously uncharacterized types (Fig. 2), showing that *E. coli* capsule diversity extends far beyond the set collected by twentieth-century serologists[6,18,19,32,33]. Phylogeny of the conserved capsule biosynthesis machinery clarifies evolutionary relationships underlying group 2 and 3 classifications (Fig. 1c–e). This resolves group 2B, characterized by distinct KpsDE clades and the absence of *kpsU*, and expands group 3B[6,32] as plasmid-borne *kps* clusters. A limitation of our analysis is its reliance on one representative cluster per K-type, providing only a snapshot of intra-type diversity. Exceptional cases such as K15 and K150 suggest additional, unclassified groups and subgroups.

A key advance is our comprehensive characterization of the *kps* cluster catalogue, assigning putative functions to >90% of K-specific ORFs (Fig. 2). Building on evidence that TagF-like domains in capsule polymerases share fold and function despite only 15% sequence identity[63], we integrated structure prediction into our pipeline. This nearly doubled the identified GT domains linked with CAZy GT families. Our MALDI–MS results for K6 and K16 show that this annotation approach enables mechanistic hypotheses, clarifies structural model gaps and facilitates structural inference for previously uncharacterized K-types (Extended Data Fig. 2).

To open this resource, we developed kTYPr, an HMM-based tool for robust and scalable determination of K-types from genome sequences. Unlike BLAST-based approaches[19,33], kTYPr uses gene content-based K-type definitions and sensitive HMMs to recognize gene clusters that are highly divergent, reorganized, distributed across different genomic loci or fragmented in MAGs. While we have not investigated the presence of *kps* clusters on mobile genetic elements, kTYPr provides a scalable framework for such analyses and allows users to incorporate custom K-types of interest.

Our nomenclature strategy maintains continuity with established serotyping nomenclature, acknowledging that both serotyping and genotyping methods are indirect methods. Current typing relies on these approaches owing to the lack of high-throughput direct polysaccharide analysis. We expect the genetic catalogue underpinning kTYPr to be refined in future as our biochemical and structural understanding of capsules improves and modifying genes outside the *kps* locus are identified.

We propose that new K numbers are assigned when the serotype-specific region of the *kps* locus encodes a distinct gene set resulting in <90% identity at the protein level, excluding truncated or fragmented versions of otherwise identical ORFs. Conventions established in the serological era dictate that deleted K numbers are not reassigned and that, after merging two K-types, the higher number is deleted (for example, K62, K56 and K82 were deleted in favour of K2, K7 and K12)[7]. While two fimbrial antigens were originally

identified as K-types (K88 and K99)[7], we suggest that future K-type assignment is restricted to acidic polysaccharide antigens belonging to group 1 and 4 (Wzx/Wzy dependent) and 2 and 3 capsules (ABC transporter dependent).

Applying kTYPr to a curated global collection of *E. coli* genomes (Supplementary Tables 12 and 13), we found distinct K-type distributions in *E. coli* lineages, hosts and environments (Figs. 3b–d and 4g and Extended Data Fig. 4a–e). This diversity was tightly constrained in some epidemiologically successful lineages, while it remained dynamic in ecologically generalist STs (Fig. 4f,g), overall more closely reflecting ecological diversity than genetic background. This suggests evolutionary constraints within ecological niches, such as exposure to host immunity or phages.

The unknown K-types were more often found in undersampled hosts compared with the 30 known K-types (Fig. 3e,f). For example, half of the K-types identified in food and livestock were previously unknown, with K105 and K109 the most frequently observed types in food and wild animals, respectively (Fig. 3d). This analysis also hints at associations between transporter-dependent capsules and hosts and potentially geographical location (Extended Data Fig. 4c,e), which should be further investigated.

The inclusion of *E. coli* MAGs from healthy individuals as carriage control revealed unexplored associations of capsules with human health and disease, with a broadly similar K-type distribution between healthy gut and *E. coli*-associated disease (Fig. 4c and Extended Data Fig. 5h). These results confirm, at an unprecedented scale, the observation that ExPEC-associated serotypes are good gut colonizers[64] and that their K-type diversity could reflect opportunistic infection of vulnerable hosts[19].

A limitation of this study is that some established capsule serotypes cannot yet be distinguished genetically. For example, K18 and K22 share identical capsule gene clusters, yet differ in acetylation of the polysaccharide, probably controlled by an acetyltransferase gene located outside of the capsule locus (Extended Data Fig. 1c). These cases have forced us to depart from established nomenclature (for example, K2 instead of K2a and K2ab, hybrid names such as K18_K22, and left K54 unassigned), accounting for gaps in our understanding of capsule biosynthesis.

Discrepancies between reported polysaccharide structures and gene clusters (for example, K6, K54) may be explained by the fact that many capsule structures were determined decades ago sometimes using distinct isolates (Supplementary Table 2). Future progress will require integrating genomics with high-throughput structural analysis.

Overall, we presented kTYPr, a scalable, fast and flexible tool to type transporter-dependent capsules. We released three new resources: a catalogue of known and previously uncharacterized *kps* loci and two curated and dereplicated global collections of *E. coli* genomes (Supplementary Table 13) and MAGs (Supplementary Table 14). Our results enable structural inference for new capsule types, highlight key knowledge gaps in polysaccharide biochemistry and outline capsule diversity across ecological niches, including understudied environments and invasive disease. This study provides a foundation for future work on *E. coli* glycobiology, ecology and epidemiology and for the development of capsule-targeted vaccines, monoclonal antibodies or phages against ExPEC-associated infections.

## Methods

### Developing a catalogue of capsule gene clusters

To survey the diversity of ABC transporter-dependent capsules in *E. coli*, we catalogued the *kps* gene clusters, firstly in the genomes of a collection of K antigen reference strains (Supplementary Table 1; deposited in PRJEB83217)[25], then in a collection of 37,732 *E. coli* genomes available from RefSeq (downloaded on 19 May 2024). We used *kpsC* as a marker gene owing to its large size and central role in capsule biosynthesis[6,13,15].

*kpsC*-positive genomes were identified with BLASTn (version 2.16.0)[65,66] with the parameters -max-target-seqs 100000 -max_hsps 1, using *kpsC* genes from K1, K3, K11, K15 and K19 reference clusters as queries. We extracted the *kpsC* flanking region ±30 kb from 11,834 genomes. This ±30-kb window was chosen based on the largest intact cluster that we could identify, K3, which extends 23 kb downstream of *kpsC*. A total of 130 sequences where *kpsC* occurred on a contig of <8 kb were excluded. The remaining sequences were dereplicated at 90% identity and coverage using the mmseqs cluster algorithm (MMseqs2 version 15-6f452)[67] with the parameters –min-seq-id 0.9 -cov 0.9 cov-mode 0 -s 7.0. The unique representatives (*n* = 2,822) were annotated using DFAST Web API[68] with the parameter dataset=ecoli because of its sensitivity for the detection of *kpsFEDUCSMT* genes. Filtering for the presence of all essential genes (*kpsEDCSMT*) yielded 1,911 unique sequences with a potential *kps* locus. To account for K-specific genes often, but not always, being flanked by the common *kps* gene, we trimmed each sequence to an approximate cluster, including a minimum of 2 ORFs upstream and downstream of the *kps* start and end coordinates, or 10 ORFs upstream and downstream in cases in which no candidate K-specific ORFs were identified within the bounds of *kpsFEDUCSMT*. Dereplication of the trimmed sequences using the mmseqs cluster with -min-seq-id 0.98 yielded 292 representative sequences. Each sequence was visually inspected to define the locus start and end coordinates, considering the organization of *kpsFEDUCSMT*, functional annotations (DFAST) and identification of common flanking genes (Supplementary Table 16). Dereplication of the manually trimmed sequences at 98% identity yielded a set of 190 unique *kps* locus sequences.

We further refined the capsule locus catalogue based on gene content. To do this, we first generated a global protein catalogue by clustering all serotype-specific ORFs (excluding transposase-related sequences) at 90% amino acid identity and coverage. Each resulting cluster was aligned using MUSCLE (v3.8.1511)[69] and used to build a protein HMM (HMMER 3.4)[70]. In an iterative process, these HMMs were applied to automate identification of locus boundaries within each *kpsC* flanking region ±30 kb. This approach ensured the inclusion of functionally related genes that may have been missed during initial manual locus curation, and it allowed us to merge loci with the same gene content. In later iterations, we transitioned to K-type-specific protein HMMs, built exclusively with sequences assigned to that K-type by analysis with kTYPr. Exceptions to this rule are proteins with a common function across multiple K-types, that is, RfbBDAC, NeuDBACE and KpsFEDUCSMT. Because closely related gene clusters can produce serologically and chemically distinct polysaccharides—such as K1 and K92, which differ by a single ORF with 83% protein identity—we used a threshold of 90% identity across all ORFs for defining distinct K-types. This led us to a final set of 85 K-types (Supplementary Table 2) defined by a set of 420 protein HMMs.

### Phylogenetic analysis of conserved *kps* genes

Phylogenetic relationships among the conserved *kpsEDCSMT* genes were inferred to support the classification of *kps* loci into distinct groups. Protein sequences corresponding to each conserved gene (*kpsD*, *kpsE*, *kpsC*, *kpsS*, *kpsT* and *kpsM*) were individually aligned using MAFFT (v7.305)[71] with default parameters. Phylogenetic trees were constructed using FastTree (v2.1.10)[72] under the JTT + CAT model for maximum-likelihood inference. The resulting phylogenies were visualized using iTOL v7 (ref. 73).

### Definition and development of kTYPr

The kTYPr tool developed and used in this study is fully written in Python (v ≥ 3.9) and can be downloaded and installed following the details in https://github.com/SushiLab/kTYPr. This bioinformatic tool accepts GenBanks or two types of input in FASTA format: genomes or user-provided translated gene sequences. If a genome is provided, whether a complete genome or a MAG, the analysis begins with gene prediction using Pyrodigal (v3.4.1) and its find_genes function[74,75]. Once a set of predicted protein sequences is obtained, kTYPr uses PyHMMER (v0.10.11)[70,76] to scan for the presence of *kpsC* within the sequences. If a *kpsC* hit is detected, the tool extracts a ±30-kb region flanking the start and end coordinates of the hit. At this stage, no thresholds or best-hit filtering is applied, meaning that if multiple *kpsC* genes are present, regardless of score, all identified *kpsC* candidates (if multiple) and flanking genes are retained for further analysis.

The extracted subset of genes is profiled against our custom catalogue of 420 HMMs, each linked to specific K-type(s), generating an initial kTYPr output table that includes all identified genes, their corresponding HMM hits and key metrics such as bit scores. HMM hits are then filtered according to custom bit score cut-offs. By default, cut-offs were set to 60% of the maximum bit score observed for each HMM in the RefSeq collection (Supplementary Table 17). In a handful of cases, the thresholds were decreased to accommodate highly polymorphic genes in which the observed variation is known (for example, the K5 putative sulfotransferase, *kfiB*[77]) or not expected to be critical for K antigen production (for example, variation in length of the coiled-coil regions in the putative methyltransferase genes present in many group 3 clusters). However, the cut-offs for deoxythymidine diphosphate (dTDP)-rhamnose biosynthesis genes (*rfbABCD*) were increased to limit false-positive hits to homologues present in many *E. coli* O-antigen biosynthesis gene clusters. After these thresholds were applied, the filtered set of hits was analysed to determine the presence or absence of the conserved region, based on the identification of *kpsEDCSMT* and *kpsFU* genes (assigned a value of 1 if all genes are present and 0 otherwise). The same strategy is applied to each K-specific gene set associated with different K-types. The final output table reports the presence of the conserved region and, for each K-type, whether the cluster is complete, along with its accumulated bit score and this score normalized by the maximum accumulated bit score observed for each K-type identified in the RefSeq dataset. In addition, a separate set of columns designates the best K-type, prioritizing first the completeness of the cluster and then the highest accumulated bit score. This approach allows users to explore multiple or nearly complete clusters while still identifying the most probable gene-based K-type. The outputs are a detailed kTYPr report, the analysed predicted genes, a GenBank file for the extracted cluster and an HTML file with comparison of the query cluster with the predicted or closest match K-type cluster using clinker[47].

Alternatively, to enable kTYPr to run on incomplete genomes, we have implemented a 'whole-genome' mode as an alternative to the default 'flanking' mode described above. In this mode, the initial step of selecting neighbouring genes around *kpsC* is skipped; instead, every gene in the genome is evaluated throughout the process. Furthermore, the described bioinformatic process operates at the single-genome level; however, kTYPr also supports a 'collection' mode (see https://github.com/SushiLab/kTYPr). This mode allows users to provide a list of genome file paths, enabling the same analytical steps to be applied across an entire genome collection in parallel. In addition, multiprocessing capabilities are available to enhance performance, ensuring efficient processing of large datasets. In this mode, the final output consists of a comprehensive table that consolidates predictions for the entire collection, streamlining downstream comparative analyses.

To support the extension of kTYPr with new K-types and its integration into other pipelines or tools as a Python package, we provide detailed instructions in the repository. In addition, to facilitate direct reuse of our database in BLAST-based workflows such as Kaptive[48], we distribute a compatible multi-record GenBank file containing the best representative genomes associated with each K-type in our study.

All analyses presented in this study were performed using kTYPr in 'flanking mode', considering the custom definition of curated hit cut-offs.

## Definition of reference gene clusters

Some K antigen reference strains have disruptions in genes that are redundant in the *E. coli* genome (for example, *kpsF* and *galE*) that are not expected to impact capsule structure. For display in figures and for bioinformatic comparisons, we assigned a reference gene cluster for each of the 85 defined K-types (Supplementary Table 2). These were assigned first based on kTYPr results, using the cluster with the highest accumulated bit score for each K-type. The assigned reference was controlled by clinker comparison with all clusters (or a representative set of all clusters identified using MMSeqs2 with a minimum sequence identity threshold of 90%) to ensure that the gene organization was typical of the majority of clusters assigned to the given K-type. We note that the reference gene cluster is never taken from the corresponding K antigen reference strain, although in some cases the sequences are identical.

The reference gene clusters were compared in clinker, and K-specific ORFs extracted from the reference gene clusters were used for functional annotation of the gene clusters (Fig. 2).

## Evaluation of sequence similarity within the reference gene clusters and comparison with Kaptive

Percentage nucleotide and amino acid identity was calculated for each gene and protein within the reference *kps* clusters BLASTn and BLASTp[66] using default parameters. These results can be found in Supplementary Table 6.

The 85 gene clusters in the kTYPr catalogue were profiled using the Kaptive v.3.1.0 (ref. 48) assembly mode and using as database the EC-K-Typing DB (v6.11.2025)[19] with the command 'kaptive assembly/EC-K-typing_group2and3_v3.0.0.gbk <genome_list > -o output.tsv'. Results of the comparison can be found in Supplementary Tables 10 and 11. Six further previously uncharacterized K-types identified in the EC-K-Typing DB (v6.11.2025) were subsequently added to the kTYPr catalogue (K164–K169).

Instructions to reproduce these analyses have been deposited in https://github.com/SushiLab/kTYPr_EcoEpidem.

## HMM sensitivity analysis on the reference gene clusters

To assess the impact of partial alignments, which may result in lower HMM scores and failure to meet internal coverage thresholds, we performed a systematic perturbation and HMM sensitivity analysis. Specifically, we took all K antigen cluster genes from the reference collection and artificially degraded their sequences in four ways: (1) introducing random mutations to reduce sequence identity, (2) trimming the N-terminus, (3) trimming the C-terminus and (4) trimming both termini simultaneously. We then ran the same HMMs on these modified sequences and evaluated whether they passed the minimum cut-offs for a positive hit as defined by kTYPr (Supplementary Table 18). For the identity perturbation, sequences were mutated from 0% to 70% in 10% increments. For the terminal trimming, we tested 10% to 40% truncation per terminus, as well as combinations of N- and C-terminal trimming. This analysis shows that HMMs still detect hits even when a portion of protein termini is trimmed without affecting detection (Supplementary Table 19).

## Functional annotation of ORFs

Functional annotation of serotype-specific ORFs was performed using InterProScan (v5.62-94.0)[78] and dbCAN3 (ref. 37) with default parameters. InterProScan result columns 'orf_id', 'length', 'domain_start', 'domain_end', 'DB', 'DB_hit', 'DB_hit_description', 'e_value' and 'gene_ontology' were taken to group results by the ORF identifier while preserving the information by the database in which the hit was retrieved. This tab-delimited output and the standard output from EggNOG were then combined by ORF identifier for further comparison and analysis.

Protein structure predictions were performed using ColabFold (v1.3.0)[79]. The AlphaFold2 version used within ColabFold was v2.1.14

(ref. 39). Multiple sequence alignments (MSAs) were generated using MMseqs2 (ref. 80). To maximize sequence coverage, searches were conducted against the UniRef30 (ref. 81) and ColabFold environmental[82] databases. MSAs were obtained using the colabfold_search command with default parameters, specifying both databases for alignment.

Protein structure predictions were then performed using the precomputed MSAs with the colabfold_batch function. Five AlphaFold2 models were generated per query, with a maximum of three recycles. Model confidence was ranked based on predicted local distance difference test scores, and structures were retained according to this ranking. Template-based modelling and Amber refinement were not used. The models were generated using an ensemble size of one, and the pairing mode was set to 'unpaired + paired' to optimize MSA depth. The MMseqs2 search mode included both UniRef30 and environmental databases to improve structural predictions.

Serotype-specific ORFs were clustered according to sequence and structural similarity using MMseqs2 (ref. 67) and Foldseek[40], respectively. Sequence-based clustering was conducted using MMseqs2 (v14) with a minimum sequence identity threshold of 30% and 90% coverage. Clusters were generated using single-linkage clustering, ensuring that closely related sequences were grouped while maintaining sensitivity to divergent homologues. Structural homology clustering was performed using Foldseek (v8), using a TM-score threshold of 0.6 and a minimum coverage of 90%. In addition, ORFs were searched against the Protein Data Bank[83] to identify potential structural homologues using Foldseek (v8) and default parameters.

## Purification, hydrolysis, permethylation and MS analysis of capsular polysaccharides

K antigen reference strains (Supplementary Table 1) were cultivated to stationary phase in TB media at 37 °C shaking at 180 rpm in Erlenmeyer flasks. Total surface polysaccharides, including lipopolysaccharide (LPS) and capsular polysaccharide, were extracted from approximately 25 g (wet weight) cell pellets according to the hot phenol–water procedure[84,85] with slight modifications as follows. Membrane lipids were extracted from the biomass once with 500 ml of ethanol overnight with stirring, then twice with 500 ml of acetone for 2 h with stirring. The cell mass was recovered after each extraction by centrifugation at 2,500 g, 20 °C, 30 min, with slow deceleration; the supernatant was discarded. After the extraction of membrane lipids, the cell mass was air-dried and weighed, typically yielding 3–5 g dry weight. Dry cell mass was crushed to a fine powder with a spatula, then resuspended with 60 ml of 50 mM Tris–HCl, pH 8.0, at 65 °C under vigorous stirring in Teflon-coated centrifuge buckets. Subsequently, 60 ml of preheated phenol at 65 °C was added to the bucket, and vigorous stirring of the aqueous phenol mix continued for 15 min at 65 °C. The aqueous phenol mix was then cooled in an ice bath for 10 min, then centrifuged for 45 min at 4 °C with slow deceleration to reduce disruption of the separated phases. The upper aqueous phase containing LPS and capsular polysaccharide was collected in a clean vessel. The remaining phenol phase was extracted a second time with 60 ml of 50 mM Tris–HCl, pH 8.0, using the same procedure, and the upper aqueous phases were pooled. The pooled aqueous extract was dialysed 5 times over 5 days with 40-fold volume of water using 14-kDa Spectra/Por 4 regenerated cellulose membranes to remove phenol. The dialysed extract was lyophilized to dryness, weighed and resuspended at 50 mg ml[−1] in 20 mM Tris–HCl, pH 8.0, with 1 mM MgCl$_2$, and treated with 500 units of Benzonase for 1 h at 37 °C. Lipid A and short LPS molecules were removed by three successive extractions with Triton X-114 as follows: 2% by volume of Triton X-114 was added to ice-cold samples then vortexed until completely dissolved. After 10 min on ice, samples were warmed to 42 °C in a water bath to trigger phase separation of Triton X-114, then centrifuged for 10 min at 20,000 g in a hot rotor. The aqueous supernatant was removed to a fresh tube, taking care not to disturb the Triton X-114 extracted material at the bottom of the tube. Capsular

polysaccharides were then selectively precipitated by adding 1 volume of 10% cetyltrimethylammonium bromide (CTAB). The precipitated material was recovered by centrifugation for 15 min at 4,500 g and dissolving the pellet in 7 ml of 1 M NaCl. Polysaccharides were then precipitated from 1 M NaCl by the addition of 3 volumes (27 ml) of ice-cold ethanol, and centrifugation at 4 °C and 8,000 g for 15 min. Supernatant was discarded and the pellet was resuspended in 7 ml of water. Purified capsular polysaccharides were partially hydrolysed in 50 mM TFA at 25 °C overnight preceded by a 1-h incubation step at 40 °C (for K13, K20, K23, K97) or 60 °C (for K19). No pre-incubation was performed for K6 and K16.

Hydrolysates were neutralized by the addition of 7 ml 50 mM NaOH, then adjusted to 50 ml total volume with 20 mM Tris–HCl, pH 7.5. Capsular polysaccharide fragments were then loaded on a 10-ml Source 15Q column, washed with 20 mM Tris, pH 7.5, and eluted in a step gradient up to 1 M NaCl. The eluted polysaccharides were dialysed against water, then lyophilized to dryness and resuspended in water.

Polysaccharide fragments were permethylated to enhance glycan ionization efficiency. For each sample, 50 μl (50 μg of polysaccharide) was dried in a Teflon-lined screw-capped glass tube. A slurry was prepared by grinding NaOH pellets in dry DMSO, and about 0.5 ml was then added to the sample, followed by about 0.5 ml methyl iodide. The mixture was shaken vigorously for 20 min at room temperature. The reaction was quenched by the dropwise addition of water, followed by extraction with 2 ml chloroform. The chloroform layer was washed repeatedly with water until clear, then dried under a stream of nitrogen. Permethylated glycans were dissolved in 20 μl of pure acetonitrile, and 1 μl was mixed with 1 μl of 2,5-dihydroxybenzoic acid matrix (10 mg ml$^{-1}$) on the MALDI-target plate. Samples were analysed using a Bruker Rapiflex matrix-assisted laser desorption/ionization time-of-flight/time-of-flight (MALDI–TOF–TOF) mass spectrometer in positive ion mode.

### Genome collection preprocessing and annotation

For the NCBI collection, we collected 39,460 genomes reported as *E. coli* from NCBI (11 October 2024) and removed all deprecated ones and duplicated accession numbers, leading to 32,043 genomes. O- and H-types were annotated using the tool ECTyper (v1.0)[1], and sequence types were determined using the MLST tool (v2.23) available at https://github.com/tseemann/mlst. The genomes were classified into phylogroups using a published Mash (v2.3) distance-based approach[52] using a distance cut-off of 0.04 from the reference genome for that phylogroup. For phylogroup C, an alternative reference genome was selected (GCF_001515725.1) from the Microreact database, based on its classification under multiple criteria and highest sequence scores[49]. Metadata available from the Bacterial and Viral Bioinformatics Resource Center (BV-BRC)[86] were further integrated with the supplementary material of the corresponding publications and with associated NCBI BioSample records. As NCBI genomes are biased towards hosts and environments (for example, clinical isolates) and include time series from the same source, we dereplicated all genomes according to a 99% ANI cut-off using the tool skani (v0.2.2)[87]. The command skani triangle -l genome_list.txt --fast was used to generate an ANI percentage matrix by performing all-against-all genome comparisons within the collection (with genome file paths specified in the input text file). The fast mode was selected to enhance scalability, given that results for genomes with high N50 and >95% expected ANI, as in the case of this study, are comparable to those obtained using the standard mode[87]. This matrix was then processed to define clusters of genomes with at least 99% ANI. Coupling this information with metadata on sample origin, we considered as distinct genomes with distinct 99% ANI clustering, geographic origin, host type, gender, age and health state when available and unique in sequence type and O-, H- and K-type. This resulted in 23,188 distinct entries that we used for all downstream analyses (Supplementary Table 13).

For the 2,762 MAGs from the stool of healthy individuals, raw sequencing data from 25 published metagenomics studies were preprocessed and taxonomically profiled using mOTUs v4 (ref. 53). Metadata were obtained from the original studies and manually curated (Supplementary Table 14). Only MAGs associated with the *E. coli* mOTU (mOTUv4.0_000063) and from healthy individuals were kept for downstream analyses.

We used a strict definition for assigning *kps*-positive genomes, which fulfil the following criteria: (1) a complete set of essential common genes (*kpsEDCSMT*) and (2) a complete set of serotype-specific genes.

### Association between K-types, STs and O-types

The association between K-types and STs was quantified using Cramér's *V* computed from ST × K-type contingency tables, considering only STs represented by at least 20 genomes and K-types observed at least 5 times, both excluding and including genomes with incomplete K-type loci to assess robustness. Statistical significance was evaluated using a phylogroup-stratified permutation null model in which K-type labels were randomly shuffled within phylogroups.

Pairwise similarity between STs was quantified based on shared K-types, ecological niches or genomic background. Ecological niches were defined as unique combinations of host category, isolation source and geographic groups, while genomic similarity was based on shared 99% ANI cluster membership. ST-by-feature incidence matrices were converted to ST–ST Jaccard similarity matrices. Analyses were restricted to STs represented by at least 10 genomes. Associations between capsule-based similarity and ecological or genomic similarity (based on 99% ANI clusters) were assessed using Mantel tests based on Spearman's rank correlation with 9,999 permutations.

O-type polysaccharide annotations were downloaded and parsed from ECODAB[2,88].

### Statistical inference on strain invasiveness

To assess the contribution of K-types to the invasive phenotype, the NCBI collection was filtered to include only human hosts and samples annotated as 'Bloodstream Infection,' 'Meningitis,' 'Neonatal Bacteremia' and isolated from blood and cerebrospinal fluid (n = 1,118), or 'Healthy' (n = 783, Supplementary Table 13). This dataset was further integrated with 260 samples from bloodstream infections[49] (Supplementary Table 12) and with publicly available, metadata-curated MAGs from the stool of healthy humans (n = 2,762; Supplementary Table 14). We performed a multivariable logistic regression to estimate the association between K-types and invasiveness, using the glm function (family = binomial) from the R package stats (v4.5.2). The model encoded invasiveness as a binary outcome, with K-type, O-type, H-type, phylogroup, ST, host age, host gender and geographic group included as fixed effects. To ensure model stability and prevent complete separation, categories with low representation were aggregated: O-types with <15 total occurrences, H-types and STs with <10 total occurrences and K-types with ≤5 occurrences in either the carriage or invasive group were grouped as 'Other'. Statistical significance for each fixed effect was determined using two-sided Wald *z*-tests. Odds ratios and 95% Wald confidence intervals were calculated by exponentiating the model coefficients and their corresponding standard errors. Odds ratios, confidence intervals, *P* values and counts for each K-type are reported in Supplementary Table 15. All statistical analyses were conducted in R (v4.5.2).

### Reporting summary

Further information on research design is available in the Nature Portfolio Reporting Summary linked to this article.

## Data availability

The genomes sequenced in this study can be found at the European Nucleotide Archive (ENA) under the BioProject ID accession code

PRJEB83217 (https://www.ebi.ac.uk/ena/browser/view/PRJEB83217). Publicly available genomes used in this study are always identified by their RefSeq complete accession identifiers and are available via GitHub at https://github.com/SushiLab/kTYPr_EcoEpidem and via Zenodo at https://doi.org/10.5281/zenodo.18924720 (ref. 89). MAGs use unique identifiers from the mOTUs database (https://motus-db.org/). These include the ENA BioSample identifier to indicate the specific sample from which each genome was reconstructed. Mass spectrometry data were deposited as peak tables and raw Bruker Daltonics MALDI files in the ETH Research Collection at https://doi.org/10.3929/ethz-c-000797044. All information required to reproduce the presented analysis is available in Supplementary Tables and Supplementary Information (more information is presented in Code Availability). Source data are provided with this paper.

## Code availability

The software and code required to run the kTYPr tool are available via GitHub at https://github.com/SushiLab/kTYPr and via Zenodo at https://doi.org/10.5281/zenodo.18923626 (ref. 90). The code and input data used for all analyses reported in Figs. 3 and 4 and Extended Data Figs. 3–6 are available via GitHub at https://github.com/SushiLab/kTYPr_EcoEpidem and via Zenodo at https://doi.org/10.5281/zenodo.18924720 (ref. 89). also contains the code used in the ANI dereplication, tool comparison, sequence homology and HMM sensitivity analyses.

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

## Acknowledgements

This study was supported by funding from the Basel Research Centre for Child Health to E.S. and S.S., the Swiss National Science Foundation through project grants 40B2-0_180953 and 310030_185128 to E.S., the NCCR Microbiomes (51NF40_180575 and 51NF40_225148) to E.S. and S.S. an Innosuisse grant (117.143 IP-LS) to T.G.K. and E.S., and core funding from ETH Zürich. E.S. acknowledges support from a European Research Council Consolidator Grant (865730). E.S. and S.S. are supported by the LOOP Zurich mTORUS project. S.M.-V. acknowledges funding from the Human Frontier Science Program through the fellowship LT0050/2023-L (https://doi.org/10.52044/HFSP.LT00502023-L.pc.gr.171942). E.C. acknowledges support from SNSF Spark (CRSK-3_228959) and Novartis Freenovation (FN24-0000000703). A.E. was supported by an unrestricted research grant from the University of Zurich. T.F. group is funded by the Deutsche Forschungsgemeinschaft (German Research Foundation) under grant numbers 412824531 and 553100037. AlphaFold runs were performed on the ETH Euler cluster. We acknowledge the support of the IT Service and HPC facilities of ETH Zürich.

## Author contributions

Conceptualization: E.S., S.S. and T.G.K. Data curation: S.M.-V., E.C. and C.R.M. Formal analysis: S.M.-V., E.C., C.R.M. and T.G.K. Funding acquisition: E.S., S.S. and T.G.K. Investigation: S.M.-V., E.C., C.R.M., C.R., K.V., C.-w.L., E.C.B., E.M., R.S., D.J.B., T.F., M.S. and T.G.K. Methodology: S.M.-V., E.C., D.J.B. and T.G.K. Software: S.M.-V. Resources: S.M.-V., H.-J.R., A.C. and A.E. Supervision: E.S., S.S. and T.G.K. Validation: S.M.-V., E.C. and T.G.K. Visualization: S.M.-V., E.C., C.R.M. and T.G.K. Writing—original draft: S.M.-V., E.C., C.R.M. and T.G.K. Writing—review and editing: S.M.-V., E.C., E.S., S.S. and T.G.K.

## Competing interests

C.R., E.S. and T.G.K. are listed as inventors on a European patent application (EP24168910) describing methods and reagents for the preparation of capsule-targeting vaccines. C.R. and T.G.K. are founders and shareholders of Baxiva AG. The other authors declare no competing interests.

## Additional information

**Extended data** is available for this paper at https://doi.org/10.1038/s41564-026-02323-5.

**Correspondence and requests for materials** should be addressed to Samuel Miravet-Verde, Elisabetta Cacace, Emma Slack, Shinichi Sunagawa or Timothy G. Keys.

Article

**Samuel Miravet-Verde** ©[1,16] ✉, **Elisabetta Cacace** ©[1,16] ✉, **Carine Roese Mores**[1,13,16], **Christoph Rutschmann** ©[2,3], **Chia-wei Lin** ©[4], **Hans-Joachim Ruscheweyh** ©[1], **Aline Cuénod**[5], **Elisa Cappio Barazzone** ©[2], **Enora Marrec**[1], **Kateryna Vershynina**[2,14], **Raffael Schumann** ©[2,15], **Dan J. Bower** ©[6], **Mario Schubert** ©[7], **Adrian Egli**[5], **Timm Fiebig** ©[8,9], **Emma Slack** ©[2,10,11,12] ✉, **Shinichi Sunagawa** ©[1] ✉ & **Timothy G. Keys** ©[2,3] ✉

[1]Department of Biology, Institute of Microbiology and Swiss Institute of Bioinformatics, ETH Zurich, Zurich, Switzerland. [2]Department of Health Sciences and Technology, Institute of Food, Nutrition and Health, ETH Zurich, Zurich, Switzerland. [3]Baxiva AG, Schlieren, Switzerland. [4]Functional Genomics Centre Zurich, Zurich, Switzerland. [5]Institute for Medical Microbiology, University of Zurich, Zurich, Switzerland. [6]QuadraticMed GmbH, Baar, Switzerland. [7]Department of Biology, Chemistry and Pharmacy, Freie Universität Berlin, Berlin, Germany. [8]German Center for Infection Research (DZIF), Partner Site Hannover-Braunschweig, Hannover, Germany. [9]Institute of Clinical Biochemistry, Hannover Medical School, Hannover, Germany. [10]Basel Research Center for Child Health, Basel, Switzerland. [11]Sir William Dunn School of Pathology, University of Oxford, Oxford, UK. [12]Botnar Institute for Immune Engineering, Basel, Switzerland. [13]Present address: Department of Infectious Diseases and Hospital Epidemiology, University Hospital Zurich, University of Zurich, Zurich, Switzerland. [14]Present address: Institute of Cell Biology, University of Bern, Bern, Switzerland. [15]Present address: MRC Laboratory of Molecular Biology and University of Cambridge, Cambridge, UK. [16]These authors contributed equally: Samuel Miravet-Verde, Elisabetta Cacace, Carine Roese Mores. ✉e-mail: smiravet@ethz.ch; ecacace@ethz.ch; emma.slack@immune.engineering; ssunagawa@ethz.ch; timkeys@baxiva.com

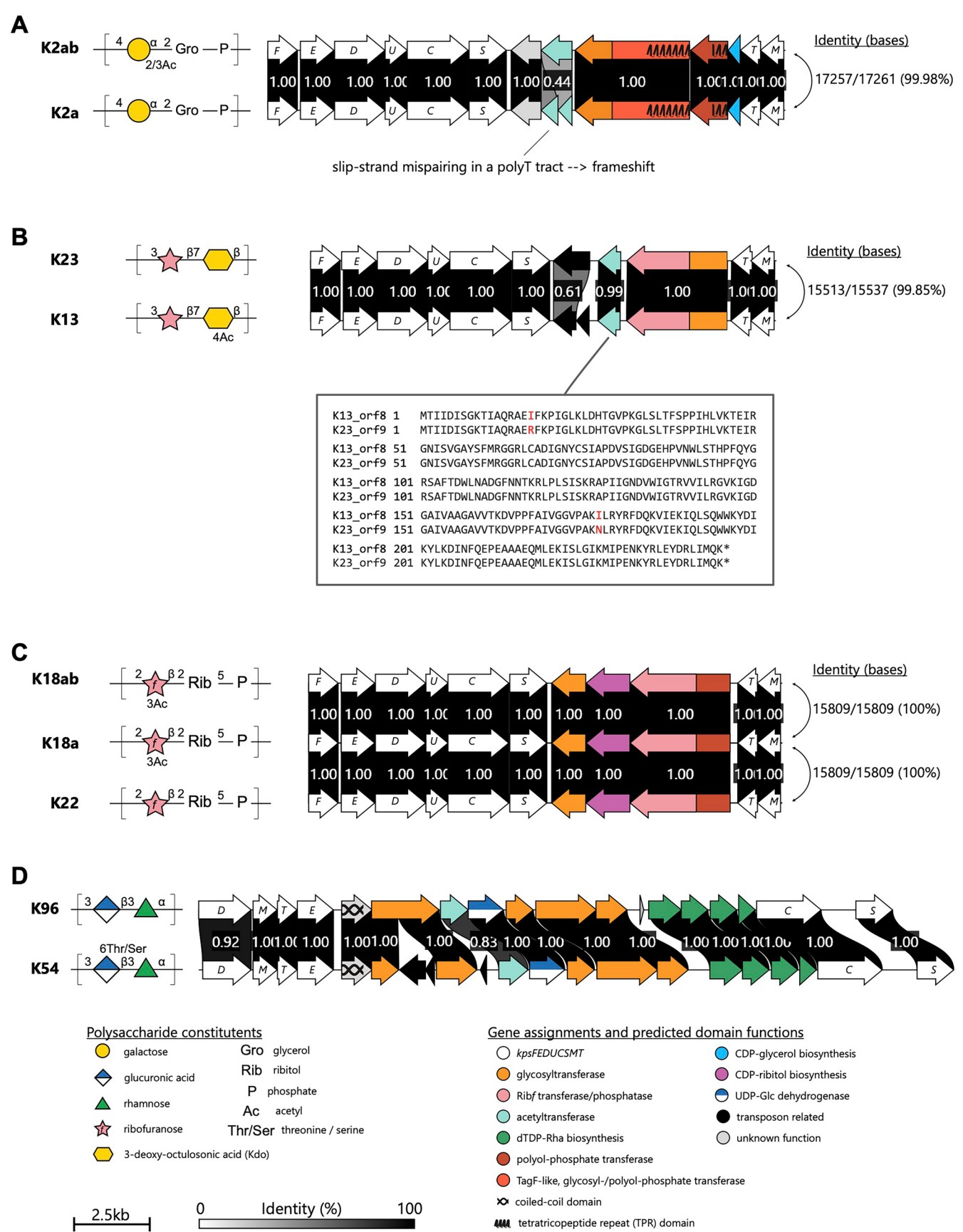

**Extended Data Fig. 1 | See next page for caption.**

**Extended Data Fig. 1 | Closely related capsule gene clusters identified in different K serotype reference strains do not support robust genetic differentiation of the respective serotypes, related to Fig. 2.** Capsule gene clusters were compared and rendered using clinker. The proportion of identical amino acids is displayed numerically and in grey-scale between opposing ORFs. The reported base identities are from global Needle-Wunsch alignments and the protein alignments were made with the Smith-Waterman algorithm. Gene assignments and functional predictions were made as described in Methods. Polysaccharides are depicted according to the conventions of the Symbol Nomenclature for Graphical Representation of Glycans and references for the displayed polysaccharide structures are provided in Supplementary Table 1. **A**. K2a and K2ab differ by acetylation of galactose in the repeat unit, consistent with a frameshift mutation in the putative acetyltransferase ORF in K2a. **B**. The K13 and K23 polysaccharides differ by acetylation of the Kdo residue in the repeat unit and the candidate acetyltransferases differ by two missense mutations. Although the observed mutations may explain the K2a/K2ab and K13/K23 serotypes, they do not provide a robust basis for genetically distinguishing these serotypes. Therefore, these serotypes are grouped under the names K2 and K13_K23, respectively, in our catalog (Supplementary Table 2). **C**. The K18 polysaccharide is an acetylated version of the K22 backbone, but the capsule gene clusters are identical (100% identity), and there is no candidate acetyltransferase gene within the clusters. The serological distinction between K18a and K18ab is unclear because chemical analysis of the polysaccharides revealed identical repeat unit structures. As there is currently no genetic basis for distinguishing these serotypes, they are grouped under the assignment K18_K22 in our catalog (Supplementary Table 2). **D**. Our analysis of the K54 and K96 gene clusters is consistent with earlier reports suggesting that the gene(s) responsible for threonine modification of the polysaccharide backbone are located outside of the *kps* locus. Pending identification of these gene(s), the K54 and K96 clusters are grouped under the assignment K96 in the current version of the catalog (Supplementary Table 2).

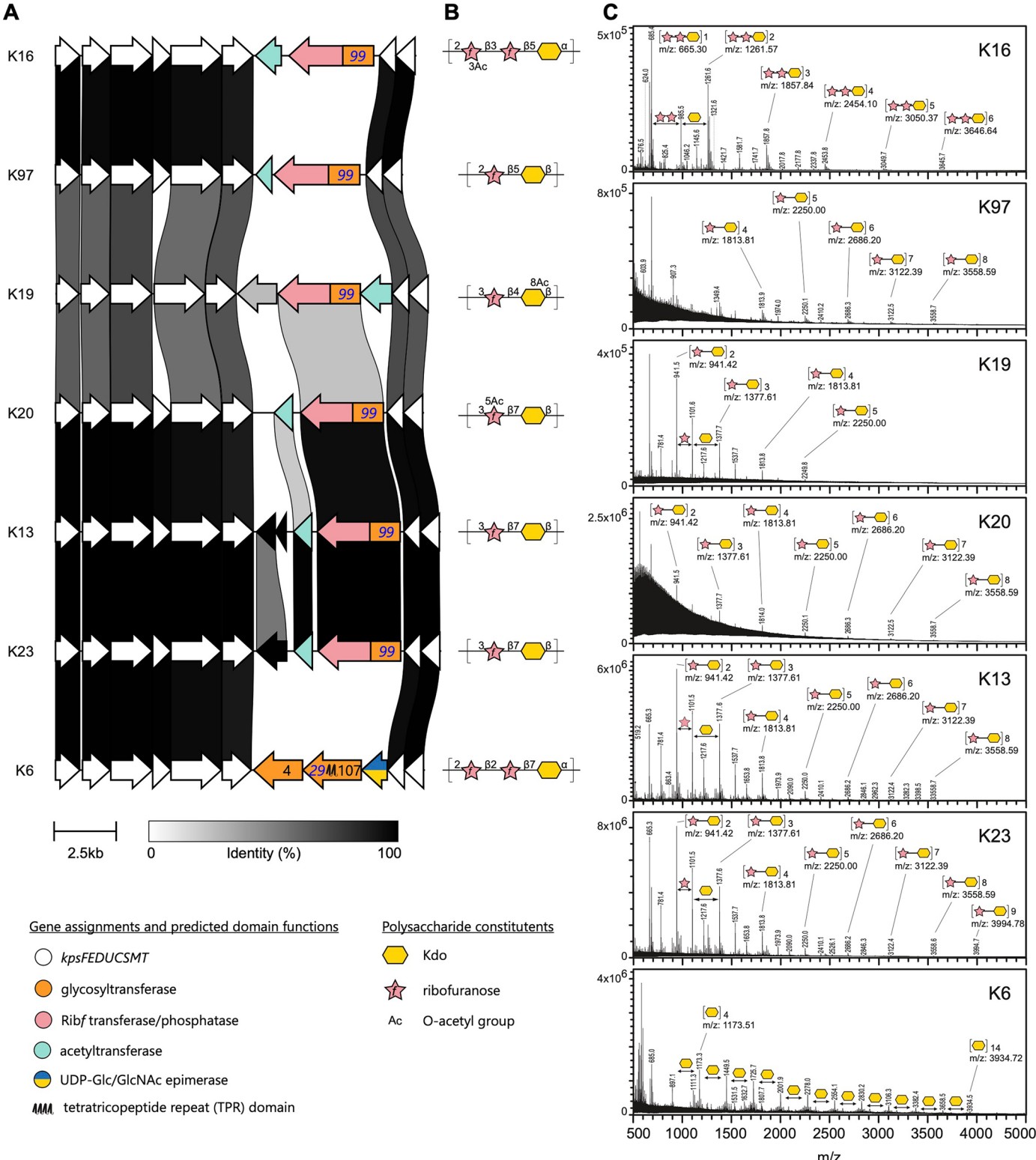

**Extended Data Fig. 2 | Mass spectrometry clarifies K antigen structures, related to Fig. 2. A.** Capsule gene clusters from K serotype reference strains were compared and rendered with clinker, then colored according to gene assignments and predicted domain functions. Amino acid identity > 30% is displayed in greyscale between opposing genes. **B.** Published polysaccharide repeat unit structures for each K serotype (see references in Supplementary Table 1). **C.** Deconvoluted MALDI-TOF-TOF spectra of partially hydrolyzed and permethylated capsular polysaccharide (Methods). Peak labels indicate the assigned structure and corresponding expected m/z species calculated in GlycoWorkbench 2 (ref. 92). We note that O-acetyl groups are expected to be lost during the permethylation procedure and were not considered when assigning peaks.

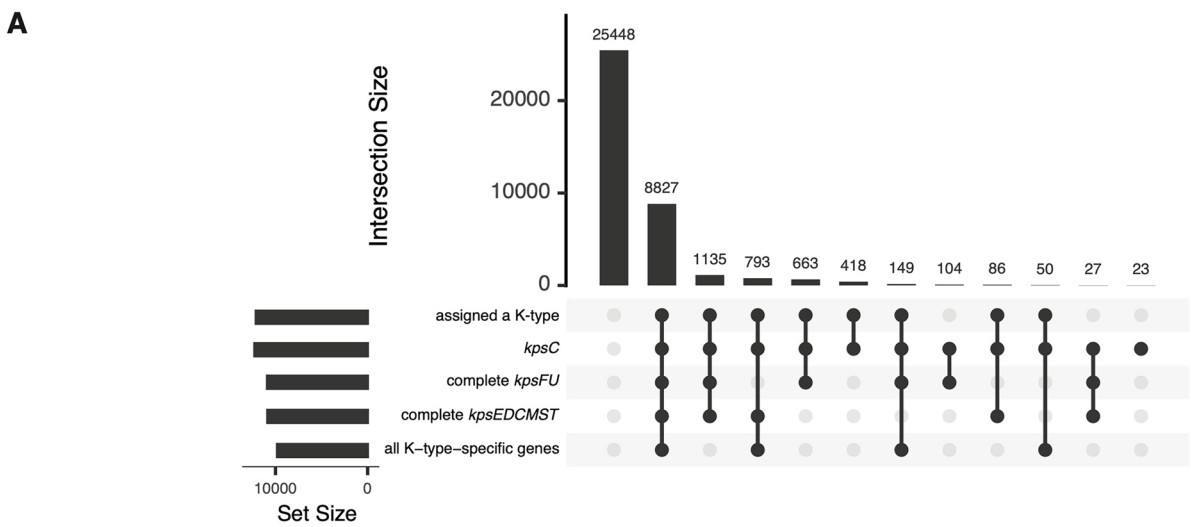

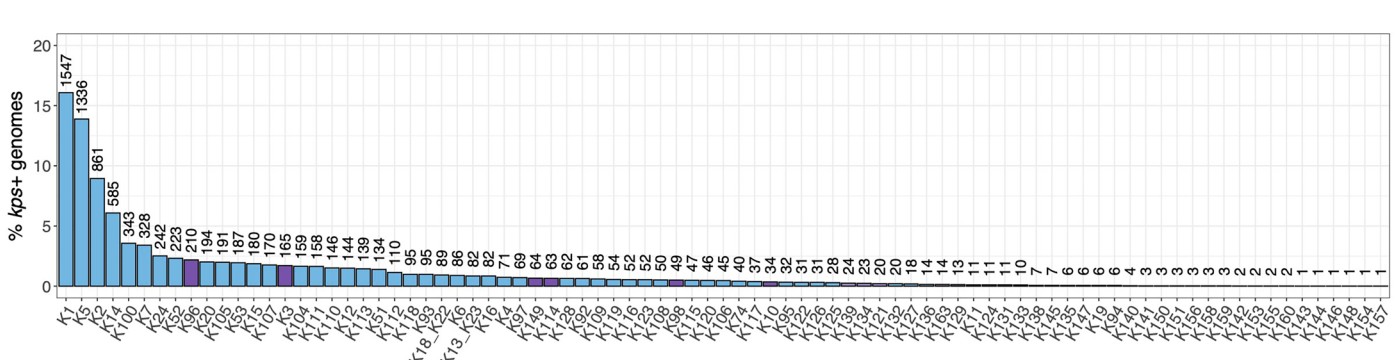

**Extended Data Fig. 3 | Evaluation of kTYPr on the RefSeq source data set. A**. Upset plot of kTYPr results on 37,723 *E. coli* genomes from RefSeq, showing the distribution of *E. coli* genomes with complete and incomplete capsule biosynthesis loci. **B**. Absolute counts of K-types identified within the same RefSeq data set (n = 37,723).

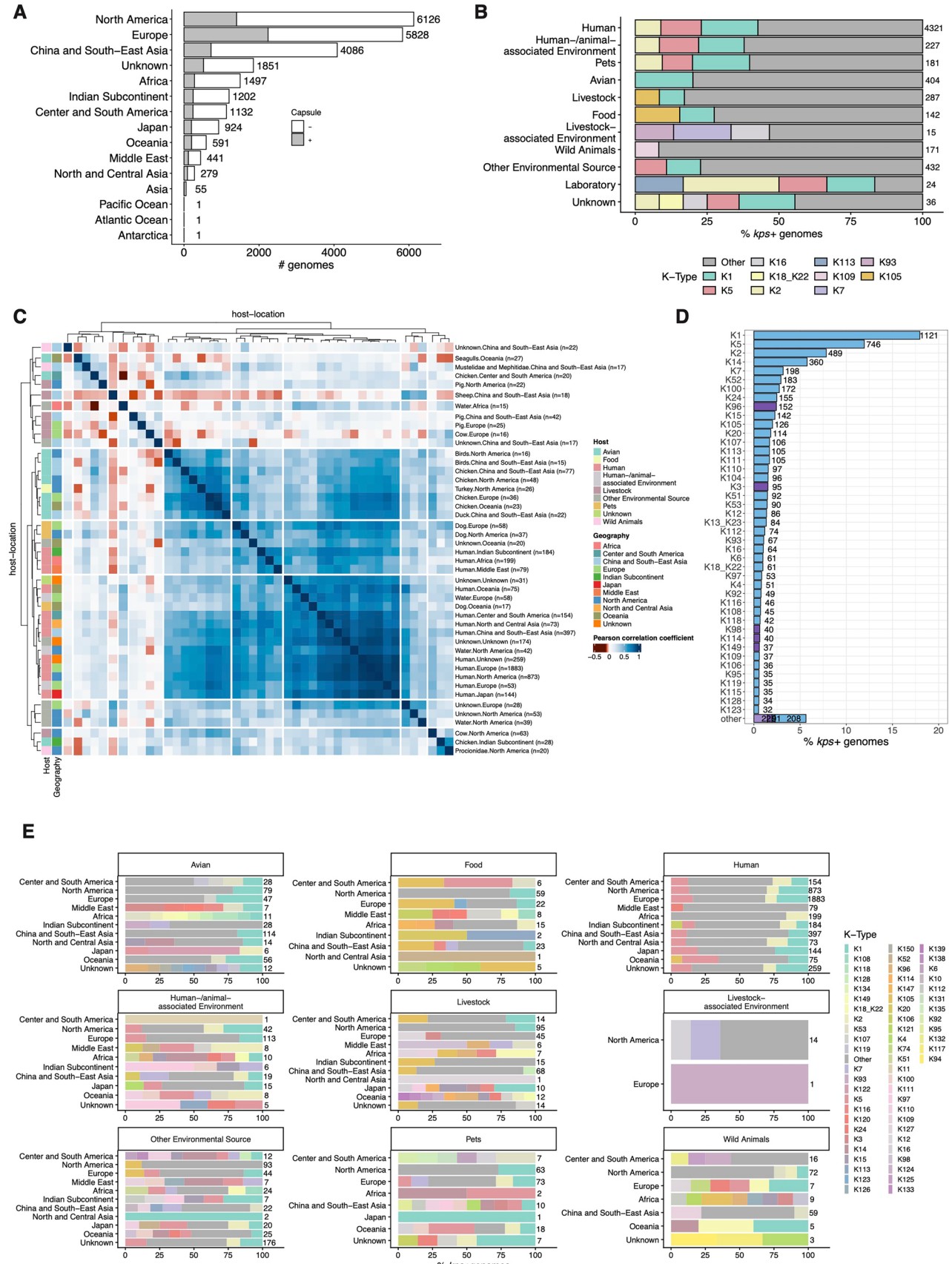

**Extended Data Fig. 4 | See next page for caption.**

**Extended Data Fig. 4 | Global diversity of K-types. A**. Fraction of *kps*-positive genomes (defined as containing a complete *kps* locus, Methods) across geography. **B**. K-type proportion in hosts and environments. K-types with frequency < 5% are annotated as 'other'. **C**. K-type proportion in individual hosts and environments. K-types and sample origin are clustered according to Pearson's correlation. Host-environment combinations with < 15 occurrences were not considered. K-type groups are color-coded as in Fig. 3c. Hosts are color-coded according to their lower resolution groups shown in Fig. 3b. **D**. Proportion and absolute counts of K-types, color-coded according to groups as in Fig. 3c. K-types with proportion < 0.5% were annotated as 'other'. Only *kps*-positive genomes were considered. **E**. K-type proportion across geographic regions, grouped by hosts and environments. K-types with frequency < 8% are annotated as 'other'. Only *kps*-positive genomes were considered.

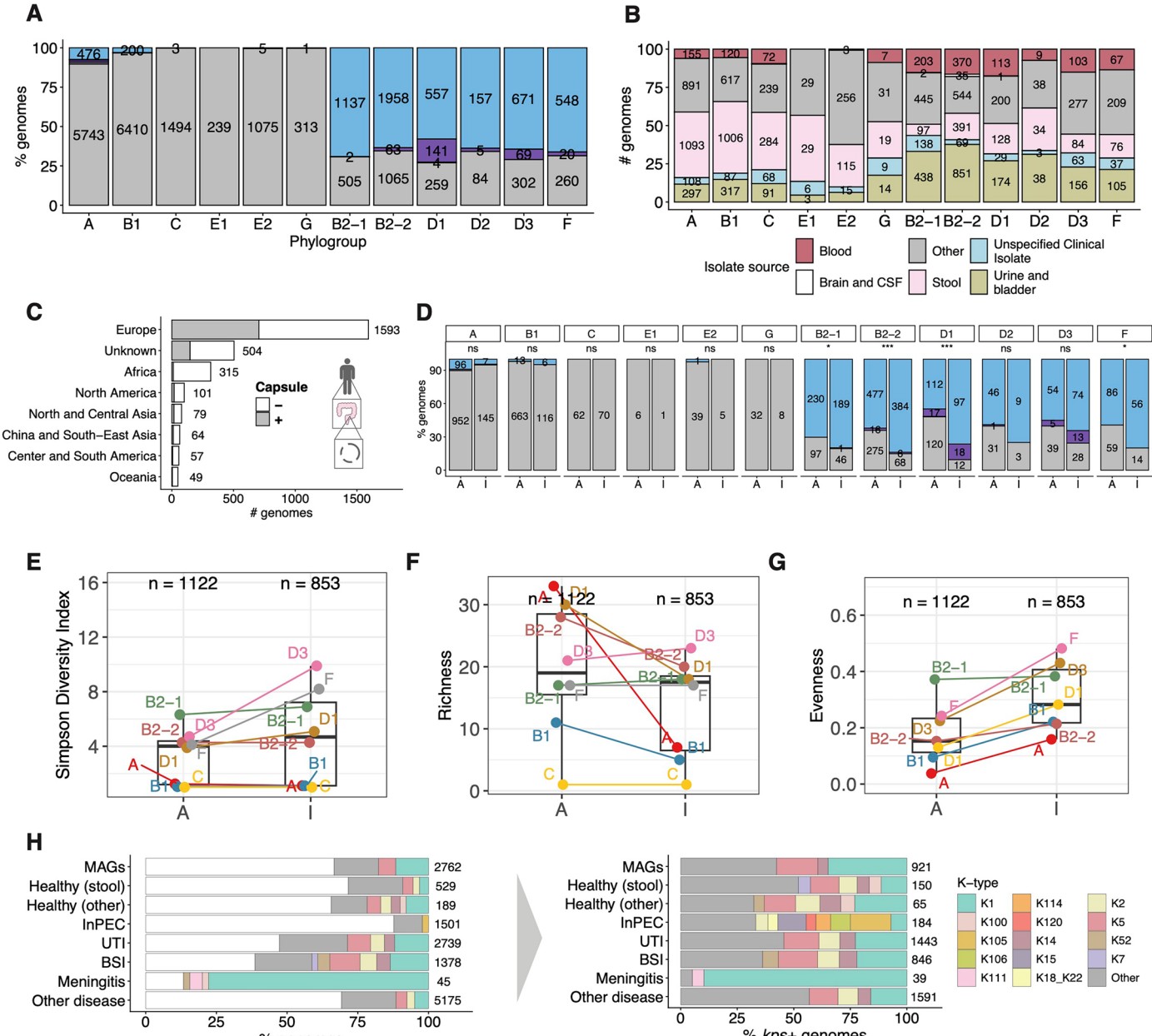

**Extended Data Fig. 5 | K-type associations with phylogroups and human health and disease. A.** Proportion of capsule groups across phylogroups in *E. coli* genomes from all hosts and environments. Groups are color-coded as in Fig. 3c, with grey indicating *kps*-negative genomes. **B.** Source of *E. coli* human isolates in the collection, grouped by phylogroup (n = 11,556). 20 genomes to which no phylogroup could be assigned were removed. **C.** Fraction of *kps*-positive genomes (defined as containing a complete *kps* locus, Methods) in *E. coli* human gut metagenomes (Supplementary Table 13). **D.** Capsule group proportion in each phylogroup in asymptomatic carriage (A, corresponding to 2,762 *E. coli* MAGs from healthy individuals, Supplementary Table 13) and invasive *E. coli*-associated disease (I, 1,118 genomes of isolates from blood or cerebrospinal fluid from NCBI and 260 from blood from a published study on

urosepsis), Methods. ns p > 0.05; * p ≤ 0.05; ** p ≤ 0.01; *** p ≤ 0.001 (two-sided Fisher's exact test, Bonferroni correction). Groups are color-coded as in Fig. 3c, with grey indicating *kps*-negative genomes. **E-G.** K-type diversity (expressed as Simpson index, **E**), richness (**F**) and evenness (**G**) of phylogroups. The same data as in D were considered, filtering out phylogroups with ≤ 25 genomes in each group (asymptomatic, A or invasive, I). The number of *kps*-positive genomes in the two groups is indicated above each boxplot. Box limits correspond to first and third quartiles, with the median marked, and whiskers extending to the most extreme data points up to 1.5 times the interquartile range (IQR). **H.** K-type proportion across health groups defined as in Fig. 4b, considering all (left) or only *kps*-positive genomes (right). K-types with proportion < 2% (left) or < 4% (right) are grouped as 'other'.

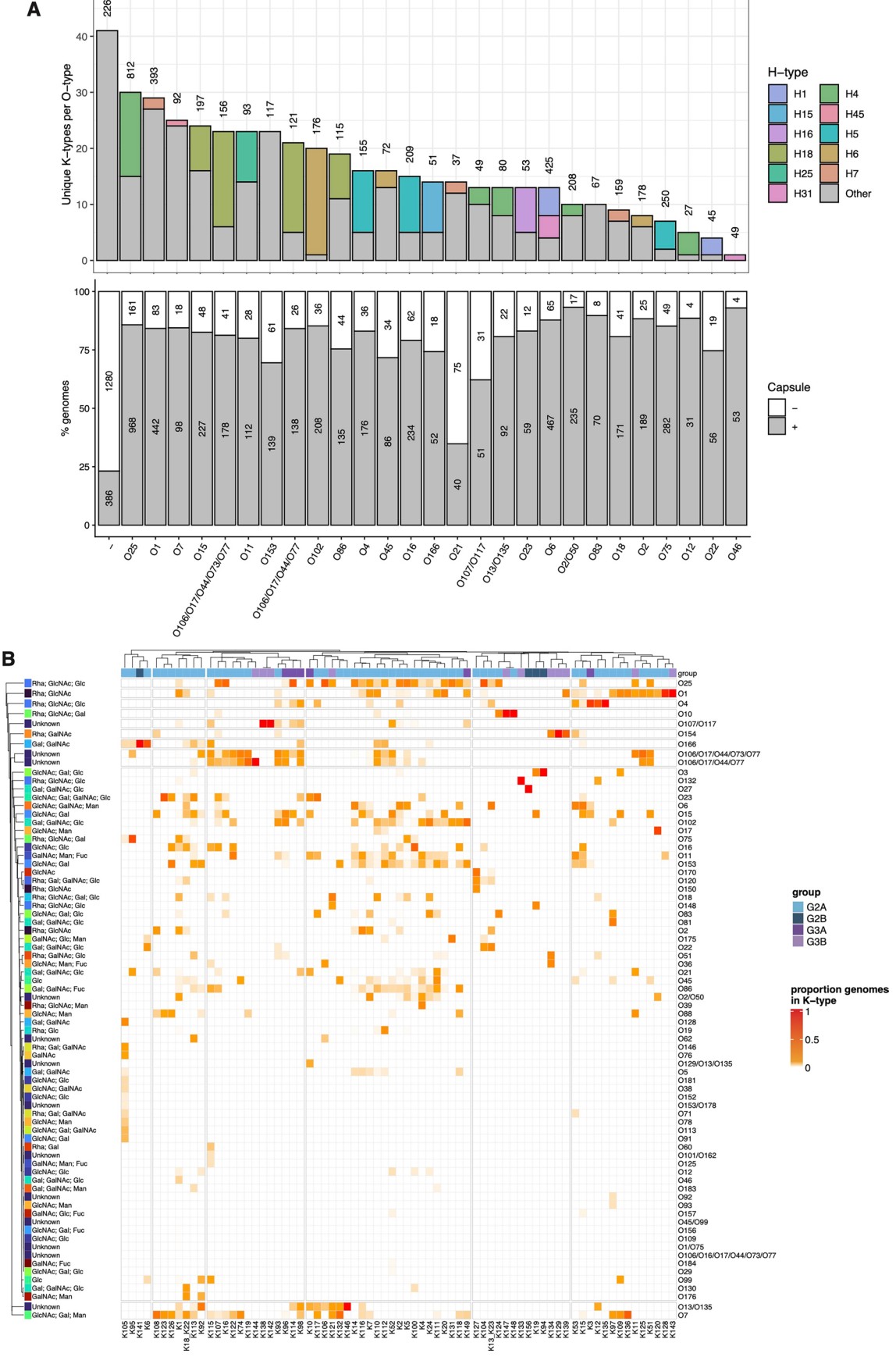

**Extended Data Fig. 6 | See next page for caption.**

**Extended Data Fig. 6 | K-type associations with O and H antigens in human-associated E. coli genomes. A.** Proportion of *kps*-positive genomes (top) and the number of unique K-types (bottom) per O-type. H-type composition is indicated for each O-type (bottom). Only O-types with > 25 genomes are shown. H-types with proportion < 30% are grouped as 'Other'. Genomes where no O-type-encoding locus was detected, and no O-type could therefore be assigned (Methods), are shown as '-'. The same data as in Fig. 4d was considered. **B.** O-type proportion for each K-type. K- and O-negative genomes are not considered (n = 4,699). K-type groups are color-coded as in Fig. 3c. O-type sugar composition was obtained from ECODAB[2,88]. Only unique sugars are listed. K12 co-occurrence with rhamnose-containing O-types is highlighted.

# Reporting Summary

## Statistics

For all statistical analyses, confirm that the following items are present in the figure legend, table legend, main text, or Methods section.

| n/a | Confirmed | |
|---|---|---|
| ☐ | ☒ | The exact sample size (*n*) for each experimental group/condition, given as a discrete number and unit of measurement |
| ☐ | ☒ | A statement on whether measurements were taken from distinct samples or whether the same sample was measured repeatedly |
| ☐ | ☒ | The statistical test(s) used AND whether they are one- or two-sided<br>*Only common tests should be described solely by name; describe more complex techniques in the Methods section.* |
| ☐ | ☒ | A description of all covariates tested |
| ☐ | ☒ | A description of any assumptions or corrections, such as tests of normality and adjustment for multiple comparisons |
| ☐ | ☒ | A full description of the statistical parameters including central tendency (e.g. means) or other basic estimates (e.g. regression coefficient) AND variation (e.g. standard deviation) or associated estimates of uncertainty (e.g. confidence intervals) |
| ☐ | ☒ | For null hypothesis testing, the test statistic (e.g. *F*, *t*, *r*) with confidence intervals, effect sizes, degrees of freedom and *P* value noted<br>*Give P values as exact values whenever suitable.* |
| ☒ | ☐ | For Bayesian analysis, information on the choice of priors and Markov chain Monte Carlo settings |
| ☒ | ☐ | For hierarchical and complex designs, identification of the appropriate level for tests and full reporting of outcomes |
| ☐ | ☒ | Estimates of effect sizes (e.g. Cohen's *d*, Pearson's *r*), indicating how they were calculated |

*Our web collection on statistics for biologists contains articles on many of the points above.*

## Software and code

Policy information about availability of computer code

| Data collection | All data is available as Extended Data Tables or, in the case of K-antigen reference strains, their acquisition and sequencing is described in the Genome Announcement by Roese-Mores et al. provided with the submission. A collection of 37,723 E. coli genomes was downloaded from Refseq on 19.05.2024. MALDI samples were analyzed using a Bruker Rapiflex MALDI-TOF mass spectrometer in positive ion mode. |
|---|---|
| Data analysis | PyHMMER (v0.10.11), Pyrodigal (v3.4.1), BLASTN (v 2.16.0), MMseqs2 v 15-6f452, MUSCLE v3.8.1511, HMMER 3.4, MAFFT (v7.305), FastTree v2.1.10, iTOL v7, ColabFold (v2.1.14), ECTyper (v1.0), MLST (v2.23), skani (v0.2.2), R (v4.5.2), Kaptive (v3.1.0), and EC-K-typing_v3.0.0 (https://github.com/rgladstone/EC-K-typing/blob/main/DB/EC-K-typing_group2and3_v3.0.0.gbk). The software and code required to run kTYPr tool can be found in the repository https://github.com/SushiLab/kTYPr and at Zenodo (https://doi.org/10.5281/zenodo.18923442). The code and input data used for all analyses reported in Figures 3-4 and Extended Data Figures 3, 6-9 can be found in the repository https://github.com/SushiLab/kTYPr_EcoEpidem and at Zenodo (https://doi.org/10.5281/zenodo.18924720). This repository also contains the code employed in the ANI dereplication, tool comparative, sequence homology and HMM sensitivity analyses. |

For manuscripts utilizing custom algorithms or software that are central to the research but not yet described in published literature, software must be made available to editors and reviewers. We strongly encourage code deposition in a community repository (e.g. GitHub). See the Nature Portfolio guidelines for submitting code & software for further information.

# Data

Policy information about <u>availability of data</u>

All manuscripts must include a <u>data availability statement</u>. This statement should provide the following information, where applicable:

- Accession codes, unique identifiers, or web links for publicly available datasets
- A description of any restrictions on data availability
- For clinical datasets or third party data, please ensure that the statement adheres to our <u>policy</u>

All data is available as Supplementary Tables and Source Data of figures. The genomes sequenced for this study can be found at the European Nucleotide Archive (ENA) under the study accession number PRJEB83217 (https://www.ebi.ac.uk/ena/browser/view/PRJEB83217). Publicly available genomes employed in this study are always identified by their RefSeq complete accession identifier in the supplementary data provided. Metagenome-assembled genomes use unique identifiers from the mOTUs database (https://motus-db.org/). These include the ENA biosample identifier to indicate the specific sample from which each genome was reconstructed. Mass spectrometry data was deposited as peak tables and raw Bruker Daltonics MALDI files in the ETH Research Collection with DOI:10.3929/ethz-c-000797044.
The software and code required to run kTYPr tool can be found in the repository https://github.com/SushiLab/kTYPr and at Zenodo (https://doi.org/10.5281/zenodo.18923442). The code and input data used for all analyses reported in Figures 3-4 and Extended Data Figures 3, 6-9 can be found in the repository https://github.com/SushiLab/kTYPr_EcoEpidem and at Zenodo (https://doi.org/10.5281/zenodo.18924720). This repository also contains the code employed in the ANI dereplication, tool comparative, sequence homology and HMM sensitivity analyses.

# Research involving human participants, their data, or biological material

Policy information about studies with <u>human participants or human data</u>. See also policy information about <u>sex, gender (identity/presentation), and sexual orientation</u> and <u>race, ethnicity and racism</u>.

| | |
|---|---|
| Reporting on sex and gender | *Use the terms sex (biological attribute) and gender (shaped by social and cultural circumstances) carefully in order to avoid confusing both terms. Indicate if findings apply to only one sex or gender; describe whether sex and gender were considered in study design; whether sex and/or gender was determined based on self-reporting or assigned and methods used. Provide in the source data disaggregated sex and gender data, where this information has been collected, and if consent has been obtained for sharing of individual-level data; provide overall numbers in this Reporting Summary. Please state if this information has not been collected. Report sex- and gender-based analyses where performed, justify reasons for lack of sex- and gender-based analysis.* |
| Reporting on race, ethnicity, or other socially relevant groupings | *Please specify the socially constructed or socially relevant categorization variable(s) used in your manuscript and explain why they were used. Please note that such variables should not be used as proxies for other socially constructed/relevant variables (for example, race or ethnicity should not be used as a proxy for socioeconomic status). Provide clear definitions of the relevant terms used, how they were provided (by the participants/respondents, the researchers, or third parties), and the method(s) used to classify people into the different categories (e.g. self-report, census or administrative data, social media data, etc.) Please provide details about how you controlled for confounding variables in your analyses.* |
| Population characteristics | *Describe the covariate-relevant population characteristics of the human research participants (e.g. age, genotypic information, past and current diagnosis and treatment categories). If you filled out the behavioural & social sciences study design questions and have nothing to add here, write "See above."* |
| Recruitment | *Describe how participants were recruited. Outline any potential self-selection bias or other biases that may be present and how these are likely to impact results.* |
| Ethics oversight | *Identify the organization(s) that approved the study protocol.* |

Note that full information on the approval of the study protocol must also be provided in the manuscript.

# Field-specific reporting

Please select the one below that is the best fit for your research. If you are not sure, read the appropriate sections before making your selection.

☒ Life sciences   ☐ Behavioural & social sciences   ☐ Ecological, evolutionary & environmental sciences

For a reference copy of the document with all sections, see <u>nature.com/documents/nr-reporting-summary-flat.pdf</u>

# Life sciences study design

All studies must disclose on these points even when the disclosure is negative.

| | |
|---|---|
| Sample size | n/a |
| Data exclusions | n/a |
| Replication | n/a |

| Randomization | n/a |
| Blinding | n/a |

# Reporting for specific materials, systems and methods

We require information from authors about some types of materials, experimental systems and methods used in many studies. Here, indicate whether each material, system or method listed is relevant to your study. If you are not sure if a list item applies to your research, read the appropriate section before selecting a response.

## Materials & experimental systems

| n/a | Involved in the study |
|---|---|
| ☒ | Antibodies |
| ☒ | Eukaryotic cell lines |
| ☒ | Palaeontology and archaeology |
| ☒ | Animals and other organisms |
| ☒ | Clinical data |
| ☒ | Dual use research of concern |
| ☒ | Plants |

## Methods

| n/a | Involved in the study |
|---|---|
| ☒ | ChIP-seq |
| ☒ | Flow cytometry |
| ☒ | MRI-based neuroimaging |

## Plants

| Seed stocks | *Report on the source of all seed stocks or other plant material used. If applicable, state the seed stock centre and catalogue number. If plant specimens were collected from the field, describe the collection location, date and sampling procedures.* |
|---|---|
| Novel plant genotypes | *Describe the methods by which all novel plant genotypes were produced. This includes those generated by transgenic approaches, gene editing, chemical/radiation-based mutagenesis and hybridization. For transgenic lines, describe the transformation method, the number of independent lines analyzed and the generation upon which experiments were performed. For gene-edited lines, describe the editor used, the endogenous sequence targeted for editing, the targeting guide RNA sequence (if applicable) and how the editor was applied.* |
| Authentication | *Describe any authentication procedures for each seed stock used or novel genotype generated. Describe any experiments used to assess the effect of a mutation and, where applicable, how potential secondary effects (e.g. second site T-DNA insertions, mosiacism, off-target gene editing) were examined.* |

