## [Peer Review File · Nature Microbiology]

In silico typing maps the natural diversity of *E. coli* transporter-dependent capsules

Corresponding Author: Professor Shinichi Sunagawa

Version 0:

Reviewer comments:

Reviewer #1

(Remarks to the Author)

This study by Roesse Mores and coauthors builds, describes and applies kTYPr for the in silico serotyping of transporter-dependent capsule types (K-types) in *E. coli*. Testament to the high novelty, applying kTYPr to 37,723 RefSeq genomes identified 55 novel transporter-dependent capsule types (i.e. not defined by classical serology). Further, proving its utility to uncover novel biology, the authors apply kTYPr to select datasets (strains from diverse hosts, colonization vs infection, etc.) and notably report on the association of newly identified K-types with invasive disease in human. Strengths of this study were: a remarkable effort in the curation and annotation of the *kps* gene/cluster catalog, a protein-structure aware typing approach (HMMs + structural inference) which reduces reliance on strict NT or AA% cutoffs to resolve functionally equivalent but sequence-divergent loci, a biochemical validation mapping genotype to polysaccharide structure for 35 reference strains, and a clear/concise/thoughtful manuscript with top quality illustrations. Weaknesses identified by this reviewer were: the absence of cross-validation/comparisons against other K-types in silico prediction tools, a possible limitation in the *kps* catalog/definition, and a lack of association of K-types with known, prevalent lineages of *E. coli*. These are detailed below for the authors to consider.

1/ In a recent pre-print (<https://doi.org/10.1101/2024.11.22.24317484>), Gladstone and coauthors released a K-typing database (90 loci with unique capsular gene presence-absence patterns identified from extensive genome mining) formatted for use with the tool Kaptive (<https://github.com/rgladstone/EC-K-typing>). While this pre-print is indeed cited (reference 20), it would be very informative for the reader and future users of kTYPr to perform a direct cross-validation (i.e. run kTYPr on the curated 90 KL references and report concordance/discordance) between kTYPr and the Kaptive database. At minimum, this would expose how HMM/structure vs gene-cluster thresholds will differ in practice (probably worth discussing in a couple lines). Another important consideration would be to report overlaps in novel types (i.e. which of the 55 putative novel k-types match any of the 90 KL references) and maybe discuss a naming convention for novel types going forward.

2/ The HMM/structure approach heavily depends on the reference set quality. When cataloguing the diversity of k-loci in RefSeq (draft?) genomes, the authors first identified the *kpsC* gene and then screened 30 kb of flanking sequences to identify and extract the *kps* loci. How was this arbitrary 30 kb window chosen? Why not searching the full contig carrying *kpsC*? Are there examples of genomes being characterized as missing some of the K-type-specific genes because the genes fell outside of that 30 kb window? It has been documented that a significant fraction of K-loci carry IS which are notorious for breaking contigs in draft genome assemblies, did the authors quantify (this time a blast-based approach may be useful) how many possible genomes with complete k-type-specific genes may have been missed?

3/ It would be very informative for epidemiologists and infectiologists to provide a population-level analysis of the 85 identified K-types and their association with major/prevalent lineages (sequence-types) of *E. coli*. The raw data is provided in Table S3 but no analysis was performed. Such correlation was done at the level of phylogroups (Fig. S5) but sequence types will be more informative as MLST remains one of the most used typing strategy to track lineages in a variety of hosts at both local and global scales.

(Remarks on code availability)

Successful install and analysis of *E. coli* genomes.
Working as expected.

Reviewer #2

(Remarks to the Author)

Mores and colleagues report on the construction of a comprehensive catalog of capsule types in *E. coli* and describe their associations with different environments and disease states. First, sequenced representatives of 35 known serotypes were used to carefully construct a genotype-phenotype map. Next, 37K public *E. coli* genomes were analyzed, with capsule loci clustered by gene content to identify 55 novel genetically-defined capsule types. Using this curated database of capsule types, the authors

created an HMM-based tool to predict capsule types from genomes and MAGs. Finally, the authors use their approach to describe the prevalence of capsule types in different environments, and their association with human disease states. Important observations include the enrichment of understudied environments (e.g. food, livestock) in novel types, the frequency of group 2/3 capsule types in healthy stool and the association of specific capsule types with invasive infection. Overall, this is an outstanding contribution, providing a rigorous framework enabling future studies of capsule biology and the impact of capsule of colonization and human infection. I have only a few comments, primarily requesting more details on the computational framework for capsule type prediction.

Major comments

1. The premise of using HMMs to define the presence of capsule loci versus a BLAST approach is well grounded. However, I presume that the HMM approach is much more computationally demanding. It would therefore be of value to the community to compare the HMM approach to defining capsule types with a more naïve BLAST-based approach, to demonstrate the value added.
2. I appreciate the author's decision to provide a score for each capsule locus, so that the relative confidence can be assessed by the user. However, it would also be helpful to put these scores in context (e.g. best possible score for a given capsule type, or score for verified hits). In this way the user can have a sense of whether all hits are relatively weak and should be treated with greater caution.
3. The github page was not accessible, so the software could not be evaluated.

(Remarks on code availability)

Github page not accessible.

Reviewer #3

(Remarks to the Author)

The manuscript "The natural diversity of *E. coli* transporter-dependent capsules" identifies transporter dependent capsule loci in serologically-determined reference isolates, and extends this dataset by detecting further diversity in publicly available genome sequences. The association between types, phylogroups and isolation sources was also explored in curated collections, which were filtered for redundancy to address known issues with sampling bias. This work advances our understanding of *E. coli* capsules by addressing a critical gap in our knowledge; that is the lack of direct association between defined serotypes and sequence. However, it is narrowly focused on a subset of capsules and does not represent the full set. I have several comments that need to be addressed, particularly regarding the use of HMMs for accurate and definitive typing.

Major comments

- 1) The interchangeable use of the same nomenclature for serotype, locus and structure is difficult to follow as it is not entirely clear what (genotype or phenotype) is being referred to. This is complicated by several serotypes having the same locus (Fig S1) and conversely those with different loci that have the same structure. I appreciate that the authors have tried to address this by collapsing some serotypes into one (e.g. K18_K22) but the structures still differ in their non-carbohydrate additions and hence deserve different names. In other cases, only one name is used (e.g. K2 or K96) despite being grouped with another type. Using the same numbers where possible to link serotype and sequence is important, but a separate naming system for serotypes, loci and structures needs to be used in order to distinguish them and avoid generating confusion.
- 2) I do not have the expertise to provide comment on the kTYPr code. However, I question whether an HMM-based tool is the best method to detect specific loci for typing purposes. Firstly, polysaccharide loci are usually low %GC increasing the chance of sequence fragmentation and mis-translated proteins when quality is low, which could lead to false negative hits. Secondly, HMMs may be detected in truncated/remnant sequences, and important SNPs or mutations that influence type may go undetected. Thirdly, the tool can detect genes in O-antigen loci or elsewhere in the genome and this could lead to inaccurate assignment of type. Additionally, while the method would be robust to genetic rearrangements as the authors suggest, rearrangements and inversions are important from an epidemiological and evolutionary perspective and are useful markers for surveillance. Therefore, a sequence-based approach or hybrid approach combining sequence with HMMs would be more valuable and would address these issues. It is not entirely clear why a tool was developed to locate and type capsule loci given that several other effective tools are now available. The authors state that the kTYPr approach provides more accurate detection (L443) but comparison/benchmarking against other tools is not performed. This is a shortcoming of the work and needs to be addressed.
- 3) The designation of 2B as a new group is sound and supported by Fig 1C results. However, the authors must define in the text what constitutes the assignment of a new group. This is particularly important given that the level of sequence divergence observed can be influenced by recombination and the choice of representative sequences used, which can obscure the finding of new groups. For example, K150 is placed with Group 3 but includes *kpsF/U* genes defining it as Group 2A (as per Fig 1D). This suggests that it is a novel hybrid group. On the other hand, sequence divergence suggests that K15 is a new group. Hence, rules for classifying groups need to be defined and the K150 arrangement needs to be discussed in text.
- 4) The placement of *kpsM/T* genes in Group 2 loci is an interesting finding as it challenges the dogma that these genes are located at the terminal end of the locus, representing what is referred to as 'region 3' in the literature. The authors should highlight this more in the text to point out that there are, and may be more, variant forms of Group 2 where *kpsM* is not the locus boundary.

5) The tool was tested on the same set of genomes used to catalogue loci, but the results show that ~5% with detectable HMMs could not be typed (L281). Were these genomes manually inspected to ensure that the full breadth of diversity was captured? How does the tool handle new cases where all genes for a K type are present (leading to the assignment of a known type) but the locus includes additional novel genes that could modify the structure?

6) I am unable to follow the numbers on L276-286 vs Fig S3A:

Fig S3A - 25,879 plus the other columns add to 37,689. The total should be 37,723

L279 - 37,723 minus 9,607 is 28,116 but the text states that 28,082 were not assigned

L281 - 28,082 minus 25,879 = 2,203 (which matches the number in the figure) but the text states that 1,436 are remaining

7) While structures are available for some capsules, authors should note in the text that these structures were determined using a different set of isolates and therefore the correlations between sequence and structure are not direct. This could potentially explain the issue with K6, and other cases where genes outside the capsule locus are suspected.

8) It is not clear why K74 was not directly compared with K16 in the MS analyses as the structures are clearly related. MS analyses should also be performed on K74.

9) The Kdo residue in K16 and K74 is alpha linked and would require an inverting GT. This is different to the other types with the GT99 protein, where Kdo is beta linked and requires a retaining GT. This could explain the sequence divergence seen in Fig S2 but conflicts with the statement made on L199-200. Please revise.

10) The authors should use established gene nomenclature for previously characterised loci where possible to acknowledge the work of others. Additionally, ref 27 and others should be referred to for previous structure correlations.

11) Please define what criteria and cutoffs were used to call types for each genome. For MAGs, if a low cutoff was used, types could potentially be novel or inaccurately assigned.

12) Fig 2 - Nucleotide sequence identity between loci needs to be shown. Namely because it is not clear for example how different K51 is from K108, as well as the following pairs: K113/K148, K18/K100, K53/K93, K1/K92, K16/K107. If these are pairs have the same sequence, this will affect typing results.

13) Table S1 – Granted that these genomes are integral to the manuscript, please provide the genome accessions in this table.

14) Table S2 – please provide further justification on why the serotype-referenced isolates were not used for the representative locus sequences. Table S2 is referred to in-text and Fig 2 legend for locus accession numbers. However, only genome accessions are listed. Please provide the precise locus location (contig accession number with base positions).

Minor points

Throughout the text and Fig legends, “capsule-positive” is used to describe genomes that have kps genes. While this is defined in some places, it is not defined in others. This can lead to inferring that genomes with Group 1 and 4 genes are capsule negative. Please revise to “kps-positive” in all instances.

Were the genomes screened for species-specific markers to confirm the taxon?

L55-64 – Please define K-antigen as it is not always CPS i.e. K88 and others are protein

L58 – Many capsule types are poorly immunogenic due to molecular mimicry with host structures as reported in the literature. In addition, Ref 10 does not mention E. coli. Please revise

L65 – Suggest use of zero possessive i.e. “E. coli capsular polysaccharides”

L67 – Not all E. coli O-antigens are synthesised by the Wzx/Wzy-dependent pathway. Revise to “...in a manner analogous to many E. coli O-antigens”.

L72 – move “(Fig 1A)” to line 70.

L72 – Revise to “Further sugars and/or constituents are added to the conserved oligo-b-Kdo base...”

L74-L76 – Please add reference(s) for this statement

L82-83 – Please include references for other tools including SerotypeFinder, ECTyper and SRST2

L88 – Sequencing was not reported in this work. Revise sentence to “We recently reported the genome sequences for...”

L90 – “...diversity of Group 2 and Group 3 capsule biosynthesis...”

L115 – Fig. 1A does not show a locus – remove reference to figure or add a generic representation of a kps locus with regions 1, 2 and 3 marked.

L115 – Move Table S1 citation after “70 strains”

L121 - Fig. 1A – Please define yellow shape as Kdo in legend or key

L123 - Fig. 1B legend – “...integrating Group 2 and 3 sequences for serotype reference strains...”

L139 - Was the K2a mutation confirmed, or could it be a sequencing error?

L153 – italicise kps

L161 – please name the common flanking genes

L166 – “HMM-based K-type definitions” seems to be referring to locus designations rather than the K type or structure

L199 – Please provide numbers for how many encode the GT99 vs GT107 proteins, rather than the collective total of 29.

L277 – “...were assigned a known (previously catalogued) transporter-dependent K-type...”

L300 – complete or known kps locus?

L312 – what is meant by novel? Types K104-K163? Or those not characterised in previous sections?

L369 – Table S3 not Table 3?

L422 – replace “types” with “loci”

L608-609 – This section is misleading as the bacterial strains and sequencing/assembly are published in another paper. Suggest removing or revising it to collating the genome set for analyses.

L613 – Please include that the catalogue was prepared initially using the reference strains then extended by analysing RefSeq genomes.

L619-620 -if sequences were dereplicated at 90%, shouldn't the -min-seq-id parameter be 0.9 not 0.98? Same comment for line 629

L635 – specify tool used to perform protein clustering and the tool used for identification of IS

L637 – Were HMMs trimmed?

L718 – “were” > “where”

L797 – please specify the tool and criteria used to pull and collate the metadata (geographical regions and isolation sources) associated with all genomes.

Fig 2 – For some structures, acetyl groups are shown above sugars and in others they are shown below. Some structures are very close together making it hard to distinguish which structure has the acetyl group e.g. K13_K22 vs K20. Please correct.

Fig 2 – it is difficult to distinguish blue and black italics on orange genes. Please revise.

Fig S4E and S5H – please make colour scheme (at least for common types) the same as panel B.

FigS5A – Please define grey colour

Fig S6A – bottom plot – what is meant by “-“ in first column? Are these the genomes with O-types represented in <25 genomes

Fig S7 – not provided?

(Remarks on code availability)

Decision Letter:

28th November 2025

Dear Professor Sunagawa,

Thank you for your patience while your manuscript "The natural diversity of *E. coli* transporter-dependent capsules" was under peer-review at Nature Microbiology. It has now been seen by 3 referees, whose expertise and comments you will find at the end of this email. Although they find your work of some potential interest, they have raised a number of concerns that will need to be addressed before we can consider publication of the work in Nature Microbiology.

In particular, the referees note a need to compare and benchmark the kTPYr tool against existing tools as well as to recent analyses of *E. coli* capsules. Several referees noted that it would be important to validate the use of HMMs to justify the use of this and demonstrate its benefits. In addition there were a few other comments raised by the referees, including to integrate these analyses with information on the major *E. coli* lineages, which should be addressed in a revised manuscript. Should further experimental data allow you to address these criticisms, we would be happy to look at a revised manuscript.

Given that this is a competitive area, with related preprint online, we also wanted to ask you how long you think might be needed to complete these revisions. We think it is very important to move forward quickly here, and were wondering if it would be possible to revise this study within a month?

Please include a data availability statement as a separate section after Methods but before references, under the heading "Data Availability". This section should inform readers about the availability of the data used to support the conclusions of your study. This information includes accession codes to public repositories (data banks for protein, DNA or RNA sequences, microarray, proteomics data etc...), references to source data published alongside the paper, unique identifiers such as URLs to data repository entries, or data set DOIs, and any other statement about data availability. At a minimum, you should include the following statement: "The data that support the findings of this study are available from the corresponding author upon request", mentioning any restrictions on availability. If DOIs are provided, we also strongly encourage including these in the Reference list (authors, title, publisher (repository name), identifier, year). For more guidance on how to write this section please see: <http://www.nature.com/authors/policies/data/data-availability-statements-data-citations.pdf>

* If you have not done so already we suggest that you begin to revise your manuscript so that it conforms to our Article format

instructions at <http://www.nature.com/nmicrobiol/info/final-submission>. Refer also to any guidelines provided in this letter.

When submitting the revised version of your manuscript, please pay close attention to our [href="https://www.nature.com/nature-portfolio/editorial-policies/image-integrity">Digital Image Integrity Guidelines.](https://www.nature.com/nature-portfolio/editorial-policies/image-integrity) and to the following points below:

EXTENDED DATA FIGURES

Link Redacted

Note: This url links to your confidential homepage and associated information about manuscripts you may have submitted or be reviewing for us. If you wish to forward this e-mail to co-authors, please delete this link to your homepage first.

Nature Microbiology is committed to improving transparency in authorship. As part of our efforts in this direction, we are now requesting that all authors identified as 'corresponding author' on published papers create and link their Open Researcher and Contributor Identifier (ORCID) with their account on the Manuscript Tracking System (MTS), prior to acceptance. This applies to primary research papers only. ORCID helps the scientific community achieve unambiguous attribution of all scholarly contributions. You can create and link your ORCID from the home page of the MTS by clicking on 'Modify my Springer Nature account'. For more information please visit www.springernature.com/orcid.

If you wish to submit a suitably revised manuscript we would hope to receive it within a month. If you cannot send it within this time, please let us know. We will be happy to consider your revision, even if a similar study has been accepted for publication at Nature Microbiology or published elsewhere (up to a maximum of 6 months).

Yours sincerely,

Reviewer Expertise:

- Referee #1: Bioinformatics Genomics
- Referee #2: Genomics Epidemiology Evolution
- Referee #3: Capsule Biology

Reviewer Comments:

Reviewer #1 (Remarks to the Author):

This study by Roese Mores and coauthors builds, describes and applies kTYPr for the in silico serotyping of transporter-dependent capsule types (K-types) in E. coli. Testament to the high novelty, applying kTYPr to 37,723 RefSeq genomes identified 55 novel transporter-dependent capsule types (i.e. not defined by classical serology). Further, proving its utility to uncover novel biology, the authors apply kTYPr to select datasets (strains from diverse hosts, colonization vs infection, etc.) and notably report on the association of newly identified K-types with invasive disease in human. Strengths of this study were: a remarkable effort in the curation and annotation of the kps gene/cluster catalog, a protein-structure aware typing approach (HMMs + structural inference) which reduces reliance on strict NT or AA% cutoffs to resolve functionally equivalent but sequence-divergent loci, a biochemical validation mapping genotype to polysaccharide structure for 35 reference strains, and a clear/concise/thoughtful manuscript with top quality illustrations. Weaknesses identified by this reviewer were: the absence of cross-validation/comparisons against other K-types in silico prediction tools, a possible limitation in the kps catalog/definition, and a lack of association of K-types with known, prevalent lineages of E. coli. These are detailed below for the authors to

consider.

1/ In a recent pre-print (<https://doi.org/10.1101/2024.11.22.24317484>), Gladstone and coauthors released a K-typing database (90 loci with unique capsular gene presence-absence patterns identified from extensive genome mining) formatted for use with the tool Kaptive (<https://github.com/rgradstone/EC-K-typing>). While this pre-print is indeed cited (reference 20), it would be very informative for the reader and future users of kTYPr to perform a direct cross-validation (i.e. run kTYPr on the curated 90 KL references and report concordance/discordance) between kTYPr and the Kaptive database. At minimum, this would expose how HMM/structure vs gene-cluster thresholds will differ in practice (probably worth discussing in a couple lines). Another important consideration would be to report overlaps in novel types (i.e. which of the 55 putative novel k-types match any of the 90 KL references) and maybe discuss a naming convention for novel types going forward.

2/ The HMM/structure approach heavily depends on the reference set quality. When cataloguing the diversity of k-loci in RefSeq (draft?) genomes, the authors first identified the kpsC gene and then screened 30 kb of flanking sequences to identify and extract the kps loci. How was this arbitrary 30 kb window chosen? Why not searching the full contig carrying kpsC? Are there examples of genomes being characterized as missing some of the K-type-specific genes because the genes fell outside of that 30 kb window? It has been documented that a significant fraction of K-loci carry IS which are notorious for breaking contigs in draft genome assemblies, did the authors quantify (this time a blast-based approach may be useful) how many possible genomes with complete k-type-specific genes may have been missed?

3/ It would be very informative for epidemiologists and infectiologists to provide a population-level analysis of the 85 identified K-types and their association with major/prevalent lineages (sequence-types) of *E. coli*. The raw data is provided in Table S3 but no analysis was performed. Such correlation was done at the level of phylogroups (Fig. S5) but sequence types will be more informative as MLST remains one of the most used typing strategy to track lineages in a variety of hosts at both local and global scales.

Reviewer #1 (Remarks on code availability):

Successful install and analysis of *E. coli* genomes.
Working as expected.

Reviewer #2 (Remarks to the Author):

Mores and colleagues report on the construction of a comprehensive catalog of capsule types in *E. coli* and describe their associations with different environments and disease states. First, sequenced representatives of 35 known serotypes were used to carefully construct a genotype-phenotype map. Next, 37K public *E. coli* genomes were analyzed, with capsule loci clustered by gene content to identify 55 novel genetically-defined capsule types. Using this curated database of capsule types, the authors created an HMM-based tool to predict capsule types from genomes and MAGs. Finally, the authors use their approach to describe the prevalence of capsule types in different environments, and their association with human disease states. Important observations include the enrichment of understudied environments (e.g. food, livestock) in novel types, the frequency of group 2/3 capsule types in healthy stool and the association of specific capsule types with invasive infection.

Overall, this is an outstanding contribution, providing a rigorous framework enabling future studies of capsule biology and the impact of capsule of colonization and human infection. I have only a few comments, primarily requesting more details on the computational framework for capsule type prediction.

Major comments

1. The premise of using HMMs to define the presence of capsule loci versus a BLAST approach is well grounded. However, I presume that the HMM approach is much more computationally demanding. It would therefore be of value to the community to compare the HMM approach to defining capsule types with a more naïve BLAST-based approach, to demonstrate the value added.

2. I appreciate the author's decision to provide a score for each capsule locus, so that the relative confidence can be assessed by the user. However, it would also be helpful to put these scores in context (e.g. best possible score for a given capsule type, or score for verified hits). In this way the user can have a sense of whether all hits are relatively weak and should be treated with greater caution.

3. The github page was not accessible, so the software could not be evaluated.

Reviewer #2 (Remarks on code availability):

Github page not accessible.

Reviewer #3 (Remarks to the Author):

The manuscript "The natural diversity of *E. coli* transporter-dependent capsules" identifies transporter dependent capsule loci in serologically-determined reference isolates, and extends this dataset by detecting further diversity in publicly available genome sequences. The association between types, phylogroups and isolation sources was also explored in curated collections, which were filtered for redundancy to address known issues with sampling bias. This work advances our understanding of *E. coli* capsules by addressing a critical gap in our knowledge; that is the lack of direct association between defined serotypes and

sequence. However, it is narrowly focused on a subset of capsules and does not represent the full set. I have several comments that need to be addressed, particularly regarding the use of HMMs for accurate and definitive typing.

Major comments

1) The interchangeable use of the same nomenclature for serotype, locus and structure is difficult to follow as it is not entirely clear what (genotype or phenotype) is being referred to. This is complicated by several serotypes having the same locus (Fig S1) and conversely those with different loci that have the same structure. I appreciate that the authors have tried to address this by collapsing some serotypes into one (e.g. K18_K22) but the structures still differ in their non-carbohydrate additions and hence deserve different names. In other cases, only one name is used (e.g. K2 or K96) despite being grouped with another type. Using the same numbers where possible to link serotype and sequence is important, but a separate naming system for serotypes, loci and structures needs to be used in order to distinguish them and avoid generating confusion.

2) I do not have the expertise to provide comment on the kTYPr code. However, I question whether an HMM-based tool is the best method to detect specific loci for typing purposes. Firstly, polysaccharide loci are usually low %GC increasing the chance of sequence fragmentation and mis-translated proteins when quality is low, which could lead to false negative hits. Secondly, HMMs may be detected in truncated/remnant sequences, and important SNPs or mutations that influence type may go undetected. Thirdly, the tool can detect genes in O-antigen loci or elsewhere in the genome and this could lead to inaccurate assignment of type. Additionally, while the method would be robust to genetic rearrangements as the authors suggest, rearrangements and inversions are important from an epidemiological and evolutionary perspective and are useful markers for surveillance. Therefore, a sequence-based approach or hybrid approach combining sequence with HMMs would be more valuable and would address these issues. It is not entirely clear why a tool was developed to locate and type capsule loci given that several other effective tools are now available. The authors state that the kTYPr approach provides more accurate detection (L443) but comparison/benchmarking against other tools is not performed. This is a shortcoming of the work and needs to be addressed.

3) The designation of 2B as a new group is sound and supported by Fig 1C results. However, the authors must define in the text what constitutes the assignment of a new group. This is particularly important given that the level of sequence divergence observed can be influenced by recombination and the choice of representative sequences used, which can obscure the finding of new groups. For example, K150 is placed with Group 3 but includes kpsF/U genes defining it as Group 2A (as per Fig 1D). This suggests that it is a novel hybrid group. On the other hand, sequence divergence suggests that K15 is a new group. Hence, rules for classifying groups need to be defined and the K150 arrangement needs to be discussed in text.

4) The placement of kpsM/T genes in Group 2 loci is an interesting finding as it challenges the dogma that these genes are located at the terminal end of the locus, representing what is referred to as 'region 3' in the literature. The authors should highlight this more in the text to point out that there are, and may be more, variant forms of Group 2 where kpsM is not the locus boundary.

5) The tool was tested on the same set of genomes used to catalogue loci, but the results show that ~5% with detectable HMMs could not be typed (L281). Were these genomes manually inspected to ensure that the full breadth of diversity was captured? How does the tool handle new cases where all genes for a K type are present (leading to the assignment of a known type) but the locus includes additional novel genes that could modify the structure?

6) I am unable to follow the numbers on L276-286 vs Fig S3A:

Fig S3A - 25,879 plus the other columns add to 37,689. The total should be 37,723

L279 - 37,723 minus 9,607 is 28,116 but the text states that 28,082 were not assigned

L281 - 28,082 minus 25,879 = 2,203 (which matches the number in the figure) but the text states that 1,436 are remaining

7) While structures are available for some capsules, authors should note in the text that these structures were determined using a different set of isolates and therefore the correlations between sequence and structure are not direct. This could potentially explain the issue with K6, and other cases where genes outside the capsule locus are suspected.

8) It is not clear why K74 was not directly compared with K16 in the MS analyses as the structures are clearly related. MS analyses should also be performed on K74.

9) The Kdo residue in K16 and K74 is alpha linked and would require an inverting GT. This is different to the other types with the GT99 protein, where Kdo is beta linked and requires a retaining GT. This could explain the sequence divergence seen in Fig S2 but conflicts with the statement made on L199-200. Please revise.

10) The authors should use established gene nomenclature for previously characterised loci where possible to acknowledge the work of others. Additionally, ref 27 and others should be referred to for previous structure correlations.

11) Please define what criteria and cutoffs were used to call types for each genome. For MAGs, if a low cutoff was used, types could potentially be novel or inaccurately assigned.

12) Fig 2 - Nucleotide sequence identity between loci needs to be shown. Namely because it is not clear for example how different K51 is from K108, as well as the following pairs: K113/K148, K18/K100, K53/K93, K1/K92, K16/K107. If these are pairs have the same sequence, this will affect typing results.

13) Table S1 – Granted that these genomes are integral to the manuscript, please provide the genome accessions in this table.

14) Table S2 – please provide further justification on why the serotype-referenced isolates were not used for the representative

locus sequences. Table S2 is referred to in-text and Fig 2 legend for locus accession numbers. However, only genome accessions are listed. Please provide the precise locus location (contig accession number with base positions).

Minor points

Throughout the text and Fig legends, “capsule-positive” is used to describe genomes that have kps genes. While this is defined in some places, it is not defined in others. This can lead to inferring that genomes with Group 1 and 4 genes are capsule negative. Please revise to “kps-positive” in all instances.

Were the genomes screened for species-specific markers to confirm the taxon?

L55-64 – Please define K-antigen as it is not always CPS i.e. K88 and others are protein

L58 – Many capsule types are poorly immunogenic due to molecular mimicry with host structures as reported in the literature. In addition, Ref 10 does not mention E. coli. Please revise

L65 – Suggest use of zero possessive i.e. “E. coli capsular polysaccharides”

L67 – Not all E. coli O-antigens are synthesised by the Wzx/Wzy-dependent pathway. Revise to “...in a manner analogous to many E. coli O-antigens”.

L72 – move “(Fig 1A)” to line 70.

L72 – Revise to “Further sugars and/or constituents are added to the conserved oligo-b-Kdo base...”

L74-L76 – Please add reference(s) for this statement

L82-83 – Please include references for other tools including SerotypeFinder, ECTyper and SRST2

L88 – Sequencing was not reported in this work. Revise sentence to “We recently reported the genome sequences for...”

L90 – “...diversity of Group 2 and Group 3 capsule biosynthesis...”

L115 – Fig. 1A does not show a locus – remove reference to figure or add a generic representation of a kps locus with regions 1, 2 and 3 marked.

L115 – Move Table S1 citation after “70 strains”

L121 - Fig. 1A – Please define yellow shape as Kdo in legend or key

L123 - Fig. 1B legend – “...integrating Group 2 and 3 sequences for serotype reference strains...”

L139 - Was the K2a mutation confirmed, or could it be a sequencing error?

L153 – italicise kps

L161 – please name the common flanking genes

L166 – “HMM-based K-type definitions” seems to be referring to locus designations rather than the K type or structure

L199 – Please provide numbers for how many encode the GT99 vs GT107 proteins, rather than the collective total of 29.

L277 – “...were assigned a known (previously catalogued) transporter-dependent K-type...”

L300 – complete or known kps locus?

L312 – what is meant by novel? Types K104-K163? Or those not characterised in previous sections?

L369 – Table S3 not Table 3?

L422 – replace “types” with “loci”

L608-609 – This section is misleading as the bacterial strains and sequencing/assembly are published in another paper. Suggest removing or revising it to collating the genome set for analyses.

L613 – Please include that the catalogue was prepared initially using the reference strains then extended by analysing RefSeq genomes.

L619-620 -if sequences were dereplicated at 90%, shouldn't the -min-seq-id parameter be 0.9 not 0.98? Same comment for line 629

L635 – specify tool used to perform protein clustering and the tool used for identification of IS

L637 – Were HMMs trimmed?

L718 – “were” > “where”

L797 – please specify the tool and criteria used to pull and collate the metadata (geographical regions and isolation sources) associated with all genomes.

Fig 2 – For some structures, acetyl groups are shown above sugars and in others they are shown below. Some structures are very close together making it hard to distinguish which structure has the acetyl group e.g. K13_K22 vs K20. Please correct.

Fig 2 – it is difficult to distinguish blue and black italics on orange genes. Please revise.

Fig S4E and S5H – please make colour scheme (at least for common types) the same as panel B.

FigS5A – Please define grey colour

Fig S6A – bottom plot – what is meant by “-” in first column? Are these the genomes with O-types represented in <25 genomes

Fig S7 – not provided?

Version 1:

Reviewer comments:

Reviewer #1

(Remarks to the Author)

I thank the authors for positively answering all my suggestions.

I particularly appreciate Fig. S8 (maybe consider including as a main figure?) and the associated discussion about k-types and ST. A small edit will be needed on the figure as the bottom left corner of panel C is not readable.

The rigorous work comparing kTYPr and the EC-K-typing database will also very valuable for the readers and users of these tools.

I have also reviewed the author's revisions and thorough responses to all reviewers, and I have no further questions.

(Remarks on code availability)

Reviewed in original submission only

Reviewer #2

(Remarks to the Author)

I thank the authors for their thoughtful and comprehensive response to reviewer critiques. I have no additional concerns.

(Remarks on code availability)

Reviewer #3

(Remarks to the Author)

The authors have addressed all comments and have performed the additional analyses needed. Benchmarking of kTYPr against Kaptive enhances the manuscript and confidence in the tool, and the inclusion of a multi-record gbk file for use via a BLAST-based approach further alleviates concerns regarding users who may be interested in SNPs, rearrangements, inversions etc. I would also like to thank the authors for providing additional discussion surrounding the complex issue of nomenclature. I appreciate that this will continue to be a challenging area that will continue to evolve as more genotype-serotype-structure maps become available. Defining the approach used in this work and including a proposal on how to handle future typing are valuable additions and highlight the importance of this issue moving forward.

I have only one further comment for the authors to consider. I suggest retrospectively adding the six novel types found in the Kaptive database to the kTYPr catalog for future use if this hasn't already been done.

(Remarks on code availability)

Decision Letter:

Our ref: NMICROBIOL-25093591A

6th February 2026

Dear Dr. Sunagawa,

Thank you for submitting your revised manuscript "The natural diversity of E. coli transporter-dependent capsules" (NMICROBIOL-25093591A). It has now been seen by the original referees and their comments are below. The reviewers find that the paper has improved in revision, and therefore we'll be happy in principle to publish it in Nature Microbiology, pending minor revisions to satisfy the referees' final requests and to comply with our editorial and formatting guidelines.

Thank you again for your interest in Nature Microbiology Please do not hesitate to contact me if you have any questions.

Sincerely,

Reviewer #1 (Remarks to the Author):

I thank the authors for positively answering all my suggestions.

I particularly appreciate Fig. S8 (maybe consider including as a main figure?) and the associated discussion about k-types and ST. A small edit will be needed on the figure as the bottom left corner of panel C is not readable.

The rigorous work comparing kTYPr and the EC-K-typing database will also very valuable for the readers and users of these tools.

I have also reviewed the author's revisions and thorough responses to all reviewers, and I have no further questions.

Reviewer #1 (Remarks on code availability):

Reviewed in original submission only

Reviewer #2 (Remarks to the Author):

I thank the authors for their thoughtful and comprehensive response to reviewer critiques. I have no additional concerns.

Reviewer #3 (Remarks to the Author):

The authors have addressed all comments and have performed the additional analyses needed. Benchmarking of kTYPr against Kaptive enhances the manuscript and confidence in the tool, and the inclusion of a multi-record gbk file for use via a BLAST-based approach further alleviates concerns regarding users who may be interested in SNPs, rearrangements, inversions etc. I would also like to thank the authors for providing additional discussion surrounding the complex issue of nomenclature. I appreciate that this will continue to be a challenging area that will continue to evolve as more genotype-serotype-structure maps become available. Defining the approach used in this work and including a proposal on how to handle future typing are valuable additions and highlight the importance of this issue moving forward.

I have only one further comment for the authors to consider. I suggest retrospectively adding the six novel types found in the Kaptive database to the kTYPr catalog for future use if this hasn't already been done.

Version 2:

Decision Letter:

13th March 2026

Dear Professor Sunagawa,

I am pleased to accept your Article "In silico typing maps the natural diversity of E. coli transporter-dependent capsules" for publication in Nature Microbiology. Thank you for having chosen to submit your work to us and many congratulations.

Authors may need to take specific actions to achieve compliance with funder and institutional open access mandates. If your research is supported by a funder that requires immediate open access (e.g. according to [a href="https://www.springernature.com/gp/open-science/plan-s-compliance"> Plan S principles](https://www.springernature.com/gp/open-science/plan-s-compliance) or the [a href="https://www.springernature.com/gp/open-science/us-federal-agency-compliance"> NIH public access policy](https://www.springernature.com/gp/open-science/us-federal-agency-compliance)) then you should select the gold OA route, and we will direct you to the compliant route where possible. Because authors warrant under our subscription licensing terms that they haven't committed to licensing any version of their article under a licence inconsistent with the terms of our agreement – including the applicable embargo period – publication under the subscription model isn't suitable for authors whose funders require no embargo.

With kind regards,

P.S. Click on the following link if you would like to recommend Nature Microbiology to your librarian
<http://www.nature.com/subscriptions/recommend.html#forms>

** Visit the Springer Nature Editorial and Publishing website at http://editorial-jobs.springernature.com?utm_source=ejP_NMicro_email&utm_medium=ejP_NMicro_email&utm_campaign=ejp_NMicro for more information about our career opportunities. If you have any questions please click [here](mailto:editorial.publishing.jobs@springernature.com). **

Open Access This Peer Review File is licensed under a Creative Commons Attribution 4.0 International License, which permits use, sharing, adaptation, distribution and reproduction in any medium or format, as long as you give appropriate credit to the original author(s) and the source, provide a link to the Creative Commons license, and indicate if changes were made. In cases where reviewers are anonymous, credit should be given to 'Anonymous Referee' and the source. The images or other third party material in this Peer Review File are included in the article's Creative Commons license, unless indicated otherwise in a credit line to the material. If material is not included in the article's Creative Commons license and your intended use is not permitted by statutory regulation or exceeds the permitted use, you will need to obtain permission directly from the copyright holder.

Editor

28th November 2025

Dear Professor Sunagawa,

Thank you for your patience while your manuscript "The natural diversity of *E. coli* transporter-dependent capsules" was under peer-review at Nature Microbiology. It has now been seen by 3 referees, whose expertise and comments you will find at the end of this email. Although they find your work of some potential interest, they have raised a number of concerns that will need to be addressed before we can consider publication of the work in Nature Microbiology.

In particular, the referees note a need to compare and benchmark the kTPYr tool against existing tools as well as to recent analyses of *E. coli* capsules. Several referees noted that it would be important to validate the use of HMMs to justify the use of this and demonstrate its benefits. In addition there were a few other comments raised by the referees, including to integrate these analyses with information on the major *E. coli* lineages, which should be addressed in a revised manuscript. Should further experimental data allow you to address these criticisms, we would be happy to look at a revised manuscript.

Given that this is a competitive area, with related preprint online, we also wanted to ask you how long you think might be needed to complete these revisions. We think it is very important to move forward quickly here, and were wondering if it would be possible to revise this study within a month?

Please include a data availability statement as a separate section after Methods but before references, under the heading "Data Availability". This section should inform readers about the availability of the data used to support the conclusions of your study. This information includes accession codes to public repositories (data banks for protein, DNA or RNA sequences, microarray, proteomics data etc...), references to source data published alongside the paper, unique identifiers such as URLs to data repository entries, or data set DOIs, and any other statement about data availability. At a minimum, you

should include the following statement: “The data that support the findings of this study are available from the corresponding author upon request”, mentioning any restrictions on availability. If DOIs are provided, we also strongly encourage including these in the Reference list (authors, title, publisher (repository name), identifier, year). For more guidance on how to write this section please see: <http://www.nature.com/authors/policies/data/data-availability-statements-data-citations.pdf>

* Include a “Response to referees” document detailing, point-by-point, how you addressed each referee comment. If no action was taken to address a point, you must provide a compelling argument. This response will be sent back to the referees along with the revised manuscript.

* If you have not done so already we suggest that you begin to revise your manuscript so that it conforms to our Article format instructions at <http://www.nature.com/nmicrobiol/info/final-submission>. Refer also to any guidelines provided in this letter.

When submitting the revised version of your manuscript, please pay close attention to our [href="https://www.nature.com/nature-portfolio/editorial-policies/image-integrity">Digital Image Integrity Guidelines](https://www.nature.com/nature-portfolio/editorial-policies/image-integrity). and to the following points below:

EXTENDED DATA FIGURES

When re-submitting your manuscript, please ensure that any supplementary figures and tables that are crucial to the manuscript’s conclusions are converted into Extended Data figures and tables to increase visibility of these data. Extended Data figures and tables are online-only (present in the online PDF and full-text HTML versions of the paper), peer-reviewed display items that provide essential background to the article but are not included in the main article due to space constraints. A maximum of ten Extended Data display items (figures and tables) is permitted.

Please use the link below to submit a revised paper: <link removed>

Nature Microbiology is committed to improving transparency in authorship. As part of our efforts in this direction, we are now requesting that all authors identified as ‘corresponding author’ on published papers create and link their Open Researcher and Contributor Identifier (ORCID) with their account on the Manuscript Tracking System (MTS), prior to acceptance. This applies to primary research papers only.

ORCID helps the scientific community achieve unambiguous attribution of all scholarly contributions. You can create and link your ORCID from the home page of the MTS by clicking on 'Modify my Springer Nature account'. For more information please visit www.springernature.com/orcid.

If you wish to submit a suitably revised manuscript we would hope to receive it within a month. If you cannot send it within this time, please let us know. We will be happy to consider your revision, even if a similar study has been accepted for publication at Nature Microbiology or published elsewhere (up to a maximum of 6 months).

Yours sincerely,

Reviewer Expertise:

Referee #1: Bioinformatics Genomics

Referee #2: Genomics Epidemiology Evolution

Referee #3: Capsule Biology

Reviewer Comments:

Reviewer #1

This study by Roese Mores and coauthors builds, describes and applies kTYPr for the in silico serotyping of transporter-dependent capsule types (K-types) in E. coli. Testament to the high novelty, applying kTYPr to 37,723 RefSeq genomes identified 55 novel transporter-dependent capsule types (i.e. not defined by classical serology). Further, proving its utility to uncover novel biology, the authors apply kTYPr to select datasets (strains from diverse hosts, colonization vs infection, etc.) and notably report on the association of newly identified K-types with invasive disease in human. Strengths of this study were: a remarkable effort in the curation and annotation of the kps gene/cluster catalog, a protein-structure aware typing approach (HMMs + structural inference) which reduces reliance on strict NT or AA% cutoffs to resolve functionally equivalent but sequence-divergent loci, a biochemical validation mapping genotype to polysaccharide structure for 35 reference strains, and a clear/concise/thoughtful manuscript with top quality illustrations. Weaknesses identified by this reviewer were: the absence of cross-validation/comparisons against other K-types in silico prediction tools, a possible limitation in the kps catalog/definition, and a lack of association of K-types with known, prevalent lineages of E. coli. These are detailed below for the authors to consider.

1/ In a recent pre-print (<https://doi.org/10.1101/2024.11.22.24317484>), Gladstone and coauthors released a K-typing database (90 loci with unique capsular gene presence-absence patterns identified from extensive genome mining) formatted for use with the tool Kaptive (<https://github.com/rgladstone/EC-K-typing>). While this pre-print is indeed cited (reference 20), it would be very informative for the reader and future users of kTYPr to perform a direct cross-validation (i.e. run kTYPr on the curated 90 KL references and report concordance/discordance) between kTYPr and the Kaptive database. At minimum, this would expose how HMM/structure vs gene-cluster thresholds will differ in practice (probably worth discussing in a couple lines). Another important consideration would be to report overlaps in novel types (i.e. which of the 55 putative novel k-types match any of the 90 KL references) and maybe discuss a naming convention for novel types going forward.

We thank the reviewer for these constructive comments and we agree that a comparison of kTYPr and Kaptive assignments and novel types is important. The EC-K-typing DB associated with the preprint from Gladstone and colleagues remains under active development, with 4 updates in the last six months, including major changes to the KL assignments and numbering. Therefore, we note that the following analyses were performed using the EC-K-typing database version released on 6th November, 2025 (which corresponds to the latest available and distributed along their repository <https://github.com/rgladstone/EC-K-typing/tree/main/DB> at the time of this reply).

We applied Kaptive to our 85 reference genomes (Extended Data Fig. 4 and Extended Data Table 10) and ran kTYPr on the 90 KL reference loci from Kaptive EC-K-typing database as suggested (Extended Data Tables 8 and 9). Detailed examination of the resulting assignments and direct comparison of the underlying sequences allowed us to identify a set of 62 equivalent clusters (same functional gene content) in the two databases (Extended Data Table 7). We furthermore ran Kaptive on the 37,723 *E. coli* genomes from RefSeq used to construct the kTYPr catalog. The results of this analysis are now included in an additional Extended Data Table 11.

Analysis of the kTYPr reference catalog by Kaptive and EC-K-typing database

Of the 30 phenotypically established K-types identified by kTYPr, 23 are correctly identified by Kaptive (Extended Data Fig. 4 and Extended Data Table 10). Missing and wrong assignments made by Kaptive are:

- K18_K22 misidentified as KL100 (K100)
- K94 is misidentified as KL19 (K19)
- K95 is misidentified as KL113
- K74 is equivalent to KL113
- K6 is equivalent to KL124
- K51 is equivalent to KL110
- K97 is equivalent to KL116

Of the 55 putative novel types identified by kTYPr, 33 have an equivalent KL (Extended Data Fig. 4 and Extended Data Table 7 and 10). The remaining 22 are not included in the EC-K-typing database resulting in misidentification (Extended Data Fig. 4 and Extended Data Table 8 and 10). In 15 cases, loci from the

kTYPr catalog are assigned as “Typeable” despite having a different set of serotype-specific ORFs (Extended Data Fig. 4 and Extended Data Table 10). Examples include:

- K146 which is misidentified as KL93 despite different serotype-specific gene content (Extended Data Fig. 4).
- K141, K155, K156, K157 which are all identified as KL19 despite having, in some cases, completely unrelated sets of serotype-specific genes. In fact, all group G2B clusters (including K94) are identified as KL19. The KL19 assignment is based on similarity of the common genes (*kpsFEDCSMT*) not on the serotype-specific genes (Extended Data Fig. 4). This reflects the fact that the EC-K-typing database and Kaptive do not distinguish between common and serotype-specific genes.
- K121, K129, K142, K149 and K163 are all identified as KL140 despite differences in serotype-specific gene content (Extended Data Fig. 4).
- K133, K147, K138, K139 are all identified as KL132 despite differences in serotype-specific gene content (Extended Data Fig. 4).

Finally, we note that Gladstone and colleagues use over 660,000 *E. coli* assemblies to construct their *kps* locus catalog, 17-times more sequences than the 37,723 *E. coli* genomes from RefSeq used to construct the kTYPr catalog. However, the larger number of assemblies does not necessarily reflect a higher genomic diversity, probably due to the similar geographical and ecological source of the collection. In fact, we identified 22 novel *kps* loci that are not present in EC-K-typing DB v06.11.2025, whereas only six novel *kps* clusters were included in the EC-K-typing database but not in the kTYPr catalog, all of which are exceedingly rare. Given the diversity of *E. coli* and undersampling of the global microbiome, we expect future sequencing efforts to produce a very long tail of singleton/rare capsule types. To account for this rapidly increasing knowledge, we included in kTYPr the possibility for users to employ kTYPr with their custom K-types of interest (we have included instructions in the tool documentation on how to add new K-type assignments). Together this is a strong validation of the thoroughness of our approach and the depth of coverage achieved by the kTYPr catalog.

Analysis of EC-K-typing database by kTYPr

Of the 90 KL in EC-K-typing database (v06.11.2025),

- 62 KLs have an equivalent in the kTYPr database (Extended Data Table 7)
- 6 KLs are genuine novel *kps* loci that have no equivalent in kTYPr (Extended Data Table 8). We note that these novel KLs are exceedingly rare, each occurring less than five times in ~37,000 *E. coli* genomes from RefSeq (Extended Data Table 11). Because the novel KLs (KL160, KL162, KL163, KL165, KL168, KL172) do not exist in the kTYPr database, they are misidentified by kTYPr as follows:
 - KL160 is misidentified as incomplete K151 (4/7 genes)
 - KL162 is misidentified as incomplete K160 (8/13 genes)
 - KL163 is misidentified as incomplete K133 (8/12 genes)
 - KL165 is misidentified as incomplete K115 (2/5 genes)
 - KL168 is misidentified as K139 (10/10 genes)
 - KL172 is misidentified as incomplete K93 (4/7 genes)

Notably, because kTYPr extracts the respective cluster and provides a visual comparison with the respective reference locus, these cases are readily identifiable by the user as novel clusters.

- 22 KLs are spurious, corresponding to disrupted/degraded versions of a parental *kps* cluster because they only differ from the parental locus by deletion of some part of the sequence or disruption of some ORFs by SNPs, i.e. with no genuine sequence divergence (Extended Data Fig. 5 and Extended Data Table 8).
- KL82 is misassigned. The basis for assignment of KL82 to the K82 phenotype by Gladstone and colleagues is unclear. According to the 1977 review by Orskov and Jann, K82 was deleted in favor of K12 based on cross-reactivity (Orskov et al., *Bacteriological Reviews*, 41:3, 667-710). Given that the KL82 locus is unrelated to K12, this assignment is likely to be spurious.

Approximately one quarter (22/90, 24.4%) of the KLs in EC-K-typing database (v06.11.2025) correspond to disrupted or degraded *kps* loci. Many hundreds of such degraded versions of *kps* loci can be identified in the RefSeq collection. In building the kTYPr catalog, we actively sought to avoid assigning disrupted or degraded loci as candidate novel K-types with two steps that are unique to our work: (i) we used a global protein catalog to identify the parental locus composition and avoid assigning new K-types to clusters that were a simple subset (except in specific cases such as K121, which is a subset of two different K-types, as noted in line 295), or disrupted version, of another K-type; (ii) detailed structural and functional annotation of all serotype-specific ORFs enabled us to recognize remnant fragments of functional proteins, and thereby avoid using them as defining ORFs for new K-types. The following examples demonstrate how defining disrupted K-loci as novel types leads to artificial inflation of capsule diversity, misleading predictions of “typeable” strains that are unlikely to produce a capsule, and loss of relevant biochemical information.

Example 1 – KL148, KL156, KL175

These KLs are rare, disrupted versions of the K4 locus (Extended Data Fig. 5D). Each is identified less than five times by Kaptive in the ~37k RefSeq collection (Extended Data Table 11). They each have significant regions of high identity with K4, disrupted by indels or SNPs, but without additional ORFs. Accordingly, they are identified by kTYPr as incomplete K4 clusters, with 6-8 of the 9 serotype-specific ORFs that define K4 (Extended Data Fig. 5 and Extended Data Table 8). Comparison with the K4 locus indicates that these KLs encode an intact chondroitin polymerase (>99% identity with K4) and that the disrupted ORFs are primarily related to the fructofuranose side-branch. This biochemical composition is consistent with production of a capsule with the K4 backbone lacking the fructofuranose side-branch. Thus, classification of KL148, KL156, and KL175 as distinct loci obscures their clear derivation from K4, it inflates apparent capsule diversity, and adds confusion to an already complex field.

Example 2 - KL121

A more extreme example is KL121, a degraded *kps* cluster comprising only the common genes *kpsFEDUCSMT* and no serotype-specific ORFs. There is no evidence that such a locus supports capsule production. Kaptive assigns 190 genomes in the RefSeq collection as “Typeable” with KL121 (Extended Data Table 11), including it in the top 20 most frequently reported capsule types. In contrast, kTYPr reports

the identified genes in distinct categories – common (*kpsFEDUCSMT*) and serotype-specific (by K-type) – and explicitly evaluates the completeness of each category. Among the 190 genomes assigned to KL121 by Kaptive, kTYPr does not identify any case with a complete set of serotype-specific genes. Classification of KL121 as a “Typeable” capsule locus reflects the absence of a distinction between common and serotype-specific genes in the Kaptive approach, and highlights this distinction as a strength of kTYPr’s protein-function based catalog and approach.

We thank the reviewer for the suggestion to clarify nomenclature going forward. We have added a corresponding paragraph in the discussion that explicitly addresses this point with a focus on maintaining continuity with the established serotyping system (see lines 530-547 in Discussion).

Action points: we performed a systematic cross-comparison of kTYPr and Kaptive, using the EC-K-typing-DB from Gladstone et al, on our respective reference genomes/loci. This resulted in two new Extended Data Figures (4 and 5), five new Extended Data Tables (7-11) and additional paragraphs in the Results (lines 314-330) and Discussion (lines 530-547).

2/ The HMM/structure approach heavily depends on the reference set quality. When cataloguing the diversity of k-loci in RefSeq (draft?) genomes, the authors first identified the *kpsC* gene and then screened 30 kb of flanking sequences to identify and extract the *kps* loci. How was this arbitrary 30 kb window chosen? Why not searching the full contig carrying *kpsC*? Are there examples of genomes being characterized as missing some of the K-type-specific genes because the genes fell outside of that 30 kb window? It has been documented that a significant fraction of K-loci carry IS which are notorious for breaking contigs in draft genome assemblies, did the authors quantify (this time a blast-based approach may be useful) how many possible genomes with complete k-type-specific genes may have been missed?

We agree that the HMM approach depends on the reference set quality. We note that the 30 kb window is 30 kb upstream plus 30 kb downstream of the *kpsC* gene, providing a total window of ~62 kb to identify the complete *kps* locus (including *kpsC* gene length which is ~2kb). The window was initially chosen based on the K3 cluster which extends ~23 kb past the *kpsC* gene. Despite searching a considerably larger window, we did not find any *intact* clusters larger than K3. We did identify disrupted clusters larger than K3, such as the K10 cluster (see figure below) with an inversion resulting in several genes that normally flank the *kps* locus (grey genes) appearing inside the *kps* locus. Together, this suggests that *E. coli* does not harbor intact K-loci larger than K3, which is corroborated by the Gladstone catalog where, despite using a different approach, and searching a larger number of *E. coli* genomes, no cluster larger than K3 was identified.

Further to this point, for each *kps* locus investigated for inclusion in the catalog, we checked at least 2, and up to 10, ORFs up- and down-stream of the identified cluster (defined by the common genes, *kpsFEDUCSMT*) for candidate serotype-specific genes (as outlined in Methods lines 835-839). This allowed us to identify clusters such as K132 where three of the serotype-specific ORFs are located outside of the common genes.

We deliberately avoided searching entire contigs or complete genomes (which may exceed 5 Mb) when assembling the catalog. Beyond physical linkage to the *kps* locus, there is no reliable way to determine whether a putative glycosyltransferase or other CAZyme contributes specifically to capsule biosynthesis rather than to unrelated cellular processes. Restricting the search space to the genomic neighborhood of the *kps* region therefore reduces spurious associations and improves the biological specificity of the inferred K-locus gene content. On the other hand, when kTYPr is analyzing genomes, the possibility of K-loci that are disrupted by contig breaks can be accounted for by “whole genome” mode. In this mode, the entire genome is searched for all genes in the kTYPr catalog regardless of their physical location on the same or multiple contigs. The completeness of the essential genes (*kpsEDCSMT*) and serotype-specific genes (for each K-type) is reported in the output.

Action point: the justification for the 30 kb window is now added to the relevant section of Methods (lines 828-829).

3/ It would be very informative for epidemiologists and infectiologists to provide a population-level analysis of the 85 identified K-types and their association with major/prevalent lineages (sequence-types) of *E. coli*. The raw data is provided in Table S3 but no analysis was performed. Such correlation was done at the level of phylogroups (Fig. S5) but sequence types will be more informative as MLST remains one of the most used typing strategy to track lineages in a variety of hosts at both local and global scales.

We agree with the reviewer that this is a relevant point and we now show K-type distribution across sequence types. K-type diversity was high in ecologically generalist sequence types and low in known human-associated STs, with the notable exception of the most epidemiologically successful ST, ST131, which exhibited a high K-type diversity, consistent with recent reports of capsule plasticity as a key determinant of this ST’s success.

To formally quantify the relationship between K-type and lineage, we measured the strength of association between K-type and ST using Cramér's V, and assessed its significance using a permutation-based null model in which K-types were shuffled within phylogroups to control for deep population structure. In addition, we compared ST–ST similarity based on K-type composition, ecological niches, and genome background (based on ANI99 cluster membership) by constructing Jaccard similarity matrices and evaluating their correspondence using a Mantel test. ST-ST similarity based on K-type composition was more strongly associated with ecological similarity (based on unique hosts, geographical regions and isolation sources) than with genomic background.

Together, this analysis suggests that K-type composition is structured at the ST level beyond phylogroup ancestry, but remains sufficiently dynamic to allow extensive K-type sharing across divergent genomic backgrounds, aligning with ecological niches. This reinforces the known role of capsules as determinants of ecological plasticity.

Action point: we added the results of this analysis to the main text (lines 439-458), discussion (lines 549-556), methods (lines 1094-1107) and a new Figure (updated Extended Data Fig. 8).

Reviewer #1 (Remarks on code availability):

Successful install and analysis of E. coli genomes.
Working as expected.

Thank you for testing kTYPr. We are glad it is working as expected.

Reviewer #2

Mores and colleagues report on the construction of a comprehensive catalog of capsule types in E. coli and describe their associations with different environments and disease states. First, sequenced representatives of 35 known serotypes were used to carefully construct a genotype-phenotype map. Next, 37K public E. coli genomes were analyzed, with capsule loci clustered by gene content to identify 55 novel genetically-defined capsule types. Using this curated database of capsule types, the authors created an HMM-based tool to predict capsule types from genomes and MAGs. Finally, the authors use their approach to describe the prevalence of capsule types in different environments, and their association with human disease states. Important observations include the enrichment of understudied environments (e.g. food, livestock) in novel types, the frequency of group 2/3 capsule types in healthy stool and the association of specific capsule types with invasive infection.

Overall, this is an outstanding contribution, providing a rigorous framework enabling future studies of capsule biology and the impact of capsule of colonization and human infection. I have only a few comments, primarily requesting more details on the computational framework for capsule type prediction.

We thank the reviewer #2 for the constructive comments and the overall impression shared.

Major comments

1. The premise of using HMMs to define the presence of capsule loci versus a BLAST approach is well grounded. However, I presume that the HMM approach is much more computationally demanding. It would therefore be of value to the community to compare the HMM approach to defining capsule types with a more naïve BLAST-based approach, to demonstrate the value added.

In addition to the comparison between HMM- or BLAST-based approaches in response to reviewer 1, point 1, we have performed a comprehensive comparative analysis in terms of runtimes when applying Kaptive (BLAST-based approach) and kTYPr (HMM-based) in flanking-region and whole-genome mode. To simulate different scenarios, we tested with 1, 10, 50, 100, and 1000 genome sets randomly picked from the NCBI collection employed in our study (Extended Data Table 12). Each iteration was replicated requesting 1, 8, 32, 64, 128 threads to simulate different computational resource configurations, shown below in linear and log scale.

This analysis shows that kTYPr achieves runtimes comparable to Kaptive, both in flanking mode (orange) and whole-genome mode (green), independently of the number of genomes. Notably, Kaptive runtimes plateau beyond ~1,000 genomes when using ≥ 8 threads, whereas kTYPr continues to benefit from parallelization, exhibiting improved scaling with increasing dataset size and thread count. Together, these highlight that our HMM-based approach is not necessarily more computationally expensive and it can be scaled efficiently to profile large genome collections as we show in our study.

Independently, we recognize that some users may prefer to leverage our curated K-type genome collection using a BLAST-based strategy, as implemented in EC-K-typing by Gladstone *et al.* To support this use case, we now distribute a multi-record GenBank file containing the collection of best representative genomes associated with each K-type in our study to directly reuse our database in BLAST-based pipelines such as Kaptive, following the same strategy described in Gladstone *et al.*

We have included this multi-record GenBank file in the GitHub repository and added dedicated instructions in the README.md describing how to run kTYPr in this specific configuration using a BLAST-based approach. We also introduced a corresponding section in the Methods (lines 915-919).

2. I appreciate the author’s decision to provide a score for each capsule locus, so that the relative confidence can be assessed by the user. However, it would also be helpful to put these scores in context (e.g. best possible score for a given capsule type, or score for verified hits). In this way the user can have a sense of whether all hits are relatively weak and should be treated with greater caution.

We thank the reviewer for the constructive comment. We analyzed the bitscore distribution across each K-type in the RefSeq dataset (considering cases where all essential *kps* genes - *kpsEDCSMT* - and K-type-specific genes are present).

For the most frequent K-types (K1, K2, K4, K5, K14) we found very few cases with lower accumulated bitscore: for example, only 15/232 K1 genomes had bitscores lower than two standard deviations below the mean for K1. Therefore, we agree with the reviewer that incorporating the mean or maximum score

detected for that type as reference in the tool output would be very helpful for users to gauge the quality of K-type assignments, even for cases where all conserved and specific genes are present.

We would also like to note that the bitscore distribution is limited by manually curated bitscore cutoffs that we have assigned for each HMM, as described in Methods lines 881-891, and accessible here: https://github.com/SushiLab/kTYPr/blob/master/ktypr/data/hmm_cutoffs_v20250704.tsv

Action point: for each assignment, we now provide in the kTYPr output the bitscore as a fraction of the maximum bitscore observed for that K-type in our RefSeq collection. We have included this information in the Methods section (lines 897-898).

3. The github page was not accessible, so the software could not be evaluated.

The software and test data were provided to the reviewers together with the submission as a zip archive, as confirmed by reviewer #1. We will make the GitHub page (<https://github.com/SushiLab/kTYPr>) public upon publication.

Reviewer #2 (Remarks on code availability):
Github page not accessible.

Please see above.

Reviewer #3

The manuscript “The natural diversity of *E. coli* transporter-dependent capsules” identifies transporter dependent capsule loci in serologically-determined reference isolates, and extends this dataset by detecting further diversity in publicly available genome sequences. The association between types, phylogroups and isolation sources was also explored in curated collections, which were filtered for redundancy to address known issues with sampling bias. This work advances our understanding of *E. coli* capsules by addressing a critical gap in our knowledge; that is the lack of direct association between

defined serotypes and sequence. However, it is narrowly focused on a subset of capsules and does not represent the full set. I have several comments that need to be addressed, particularly regarding the use of HMMs for accurate and definitive typing.

Major comments

1) The interchangeable use of the same nomenclature for serotype, locus and structure is difficult to follow as it is not entirely clear what (genotype or phenotype) is being referred to. This is complicated by several serotypes having the same locus (Fig S1) and conversely those with different loci that have the same structure. I appreciate that the authors have tried to address this by collapsing some serotypes into one (e.g. K18_K22) but the structures still differ in their non-carbohydrate additions and hence deserve different names. In other cases, only one name is used (e.g. K2 or K96) despite being grouped with another type. Using the same numbers where possible to link serotype and sequence is important, but a separate naming system for serotypes, loci and structures needs to be used in order to distinguish them and avoid generating confusion.

We thank the reviewer for drawing attention to the important and challenging topic of nomenclature.

We address the final point first: the need for separate naming systems for serotypes, loci, and structures. Here our approach is to maintain continuity with the established nomenclature and its common usage in the literature. It is the case that, historically, the K antigen nomenclature has been used to refer to three closely related but distinct entities: the serotype, the *kps* locus (genotype), and the capsular polysaccharide structure. Thus “K-type” is used also in this work as an integrative designation that links three levels of information: the historical serotype, the *kps* locus or genotype, and, where available, the capsular polysaccharide structure. Where necessary, clarity is achieved by the use of complete noun phrases such as “the K1 locus”, “the K1 polysaccharide”, or “the K1 serotype.” However, in the majority of cases clarity is achieved by context. To give a concrete example from the manuscript, when we state “The most frequent K-types overall were K1, K5, K2 and K14...” it is clear from the context, and from the methods section, that we are referring to *in silico* detection of the corresponding genotypes using a very specific set of programmatically defined parameters, but it is also clear from the broader context that we infer the presence of corresponding polysaccharide structures. Thus, we have carefully reviewed the manuscript for any unintentional ambiguous use of the K-type nomenclature and sought to clarify these by use of complete noun phrases or by adding context.

Action point: We have clarified our approach to nomenclature in the Discussion (lines 530-537). We have also provided a proposal for nomenclature and assigning new K-types going forward in the Discussion (lines 538-547).

We also acknowledge the specific challenges that arise from several serotypes having the same locus and vice versa. Each of the cases mentioned represent a fundamental knowledge gap in our ability to infer capsule structure from genomic information. For this reason, we have sought to highlight these cases openly in the manuscript as areas where further research is required (e.g. Extended Data Fig. 1). The names that we provide for the respective K-types reflect these limitations. For example, K18_K22 does

not collapse two serotypes into one but it highlights that we are not able to distinguish them using current knowledge of genotype-phenotype relationships. K54 is not assigned because (i) the cluster is not identical to K96, and (ii) the biochemical and genetic basis for the Thr/Ser modification of the polysaccharide backbone is not understood. As we acknowledge in the text, until the respective genes are identified, the high similarity between the K96 and K54 loci will result in the grouping of these K-types under the K96 assignment by the current version of kTYPr.

Action point: We have clarified in the Results (lines 150-153) that merged/missing designations reflect limitations of the current catalog to be addressed with future research. Furthermore, we added a paragraph in the Discussion (lines 573-581) which acknowledges these points as limitations of our study, and not as suggestions to change nomenclature or delete certain types.

2) I do not have the expertise to provide comment on the kTYPr code. However, I question whether an HMM-based tool is the best method to detect specific loci for typing purposes. Firstly, polysaccharide loci are usually low %GC increasing the chance of sequence fragmentation and mis-translated proteins when quality is low, which could lead to false negative hits. Secondly, HMMs may be detected in truncated/remnant sequences, and important SNPs or mutations that influence type may go undetected. Thirdly, the tool can detect genes in O-antigen loci or elsewhere in the genome and this could lead to inaccurate assignment of type. Additionally, while the method would be robust to genetic rearrangements as the authors suggest, rearrangements and inversions are important from an epidemiological and evolutionary perspective and are useful markers for surveillance. Therefore, a sequence-based approach or hybrid approach combining sequence with HMMs would be more valuable and would address these issues. It is not entirely clear why a tool was developed to locate and type capsule loci given that several other effective tools are now available. The authors state that the kTYPr approach provides more accurate detection (L443) but comparison/benchmarking against other tools is not performed. This is a shortcoming of the work and needs to be addressed.

We thank the reviewer for pointing out important and reasonable aspects regarding the kTYPr tool.

Regarding the point about potential false negatives caused by gene fragmentation, we acknowledge that truncated protein sequences can impact identification. Such partial alignments may yield lower HMM scores and fail to meet the internal coverage and score thresholds employed by our tool. To evaluate the impact of this effect, we performed a systematic perturbation and HMM scoring sensitivity analysis. For this, we took all K antigen cluster genes from the reference collection, degraded their sequences by trimming the N-, C-, or both termini regions, ran the same HMMs, and then evaluated whether the degraded sequences passed the minimum cutoffs for a positive hit employed by the tool. This analysis reveals that:

- N-termini: for 240 HMMs, 30% (relative to total protein length) from N-termini regions can be trimmed without affecting the detection. 180 HMMs still identify a hit when 40% is trimmed from the queries.

- C-termini: For 4 HMMs, 20% can be trimmed without affecting the detection. 271 HMMs still identify a hit when 30% from is trimmed, and 145 HMMs identify proper hits even after trimming 40% of the protein query.

The sensitivity results per HMM are now part of the Extended Data Table 17 and we have included a brief description on the analysis performed in the section ‘HMM sensitivity analysis on the reference gene clusters’ in Methods (lines 949-959). Nevertheless, evaluating whether the truncated/remnant sequences preserve their function requires a case-by-case study per gene considered, but overall these results demonstrate that our method is robust to substantial terminal sequence loss while maintaining reliable K-type identification.

Regarding important SNPs, HMM-based approaches are generally more robust than BLAST-based methods because they model the entire protein family profile rather than relying on individual sequence similarity. An HMM captures conserved positions, patterns, and the probability of amino acid substitutions across the family, allowing it to tolerate single-residue changes or small insertions/deletions without losing detection. In contrast, BLAST performs pairwise sequence alignment, which is highly sensitive to mismatches; even a few critical SNPs in key positions can reduce the alignment score below significance thresholds, leading to potential false negatives. Therefore, for detecting genes with variations or mutations, HMMs provide a more sensitive and reliable measure of functional homology, making them better suited for capturing K antigen cluster genes even in the presence of SNPs.

Moreover, we acknowledge the reviewer’s observation regarding our tool potentially identifying O antigen loci or hits elsewhere in the genome. This is indeed a possible scenario, but it occurs only when employing the whole-genome mode, which is not the default. In the analyses presented in the main manuscript, and as described in the tool usage instructions, kTYPr is run in flanking mode, where only genes flanking the essential *kpsC* marker gene are considered; any gene present in a different locus is ignored. Nevertheless, the whole-genome mode can still be useful for exploratory purposes, as it allows the identification of possible antigen-related genes that may not be present within a canonical cluster configuration or which are located on a different contig due to quality or completeness of the genome sequence. Furthermore, in the rare cases where closely related homologs are commonly present in K and O antigen biosynthesis gene clusters (e.g. *rfbABCD*) we tuned the bitscore thresholds for the respective HMMs to reduce false positive hits (see Methods, lines 889-891).

Regarding the implementation of a sequence homology-based approach and the advantages that kTYPr provides compared to other tools, we refer to our responses to Comment 1 from Reviewer 1 and Comment 1 from Reviewer 2, where we detail the observed differences between our approach and classic BLAST-based analyses in terms of profiling and computational runtime. In addition, as noted in our response to Reviewer 2, we now provide our database in a format compatible with BLAST-based pipelines such as Kaptive, following the same strategy described in Gladstone *et al.*

Finally, regarding the comment that rearrangements and inversions are important from an epidemiological and evolutionary perspective, kTYPr implements clinker to visualize such rearrangements (see for example figure appended to Comment 2 from Reviewer 1). While the genes are still detected as

hits, users can directly explore these structural variations by comparing the observed gene organization to the expected reference cluster.

Independently, it is worth noting that HMM-based approaches are generally expected to be more robust than BLAST-based methods for the challenges described. HMMs model the full profile of a protein family, capturing both conserved positions and tolerated variation across sequences. This allows them to detect genes even in the presence of truncations, minor insertions/deletions, or single-residue changes, which might otherwise reduce alignment scores in a BLAST search below significance thresholds. In contrast, BLAST relies on pairwise similarity, making it more sensitive to individual mismatches and potentially more prone to false negatives in fragmented or low-quality sequences. Additionally, the probabilistic nature of HMMs enables recognition of functionally conserved regions even when some residues vary, providing greater sensitivity to detect K antigen cluster genes under realistic genomic variation, including SNPs or partial sequences.

Action points: we included the results of this perturbation analysis in Extended Data Table 17 and described them in the Methods section (lines 949-959).

3) The designation of 2B as a new group is sound and supported by Fig 1C results. However, the authors must define in the text what constitutes the assignment of a new group. This is particularly important given that the level of sequence divergence observed can be influenced by recombination and the choice of representative sequences used, which can obscure the finding of new groups. For example, K150 is placed with Group 3 but includes *kpsF/U* genes defining it as Group 2A (as per Fig 1D). This suggests that it is a novel hybrid group. On the other hand, sequence divergence suggests that K15 is a new group. Hence, rules for classifying groups need to be defined and the K150 arrangement needs to be discussed in text.

We thank the reviewer for their support of the group 2B designation and for drawing attention to the fact that the groups do not represent independently evolving lineages but that the relationships are obscured by rearrangements and exchanges between and within the groups, potentially also with other species.

We have revised this section of the Results (lines 186-189) to highlight that group 2B was observed as an additional clade in the KpsDE phylogeny and that assigning K-types to groups (2A, 2B, 3A, or 3B) based on their position (clade) in the KpsDE phylogeny generally correlates (i.e. is not entirely consistent) with presence/absence of *kpsF* and *kpsU* genes in the locus as well as the organization of the conserved genes. We further highlight that K150 is an exceptional cluster with some properties of group 3A and others of group 2A (lines 191-193).

The reviewer also correctly highlights that a limitation of our work is the dependence on representative sequences for each K-type. Given the limited scope of our investigation on this topic, our intention is not to restrict the field with rules and definitions for group identification, but only to describe the major patterns of relatedness among the *kps* loci in a manner that builds upon the historical group 2 and 3 designations as well as the more recent literature on this topic. Accordingly, we have added text to the

discussion to acknowledge the limitation that our study of evolutionary relationships relies on representative clusters (lines 506-509). In this context, we would like to highlight that kTYPr's use of HMMs make it particularly well-suited as a basis for a more comprehensive study of these evolutionary relationships. Because HMMs allow for a better representation of the intrinsic diversity of genes within K-type groups, they are able to detect conserved features across divergent sequences that might be missed by simple pairwise alignment approaches (e.g. BLAST) against a single reference sequence, we highlight the advantage of kTYPr in this respect in the Discussion (lines 530-537).

Furthermore, the reviewer correctly suggests that the exceptional K15 and K150 clusters hint at the possibility of further groups and subgroups that may be identified by a more comprehensive exploration of *kps* locus diversity and phylogeny. We agree and have added corresponding text to the Discussion (lines 506-509).

4) The placement of *kpsM/T* genes in Group 2 loci is an interesting finding as it challenges the dogma that these genes are located at the terminal end of the locus, representing what is referred to as 'region 3' in the literature. The authors should highlight this more in the text to point out that there are, and may be more, variant forms of Group 2 where *kpsM* is not the locus boundary.

We thank the reviewer for highlighting this point. In the revised manuscript, we highlight that group 2 clusters are not always flanked by *kpsM* (lines 201-203).

5) The tool was tested on the same set of genomes used to catalogue loci, but the results show that ~5% with detectable HMMs could not be typed (L281). Were these genomes manually inspected to ensure that the full breadth of diversity was captured? How does the tool handle new cases where all genes for a K type are present (leading to the assignment of a known type) but the locus includes additional novel genes that could modify the structure?

We thank the reviewer for these questions. We would like to clarify that all genomes that were not assigned a K-type (n=25,602) had incomplete sets of K-type-specific genes. Of these, only 27 genomes (0.07% of the total 37,723) contained a complete set of conserved genes (*kpsEDCSMT*). We apologize for this misunderstanding and clarified the relevant lines in the main text (lines 301-310). We additionally edited Extended Data Fig. 3A to improve clarity on these numbers as suggested (see response to point 6 below).

To explore and catalog the diversity of K-types, we annotated, manually inspected, and compared all unique loci with a complete set of essential genes (*kpsEDCSMT*) with known and novel types in order to identify, as far as possible, the full breadth of *kps* locus diversity. Nevertheless, as we are not able to guarantee that all K-types were identified, we acknowledge in the manuscript that incomplete cases may represent (i) disrupted gene clusters, (ii) genes missing in draft genomes, or (iii) uncataloged *kps* cluster diversity (rephrased lines 314-330). These 27 genomes were included in this process and might fall in any of these categories.

To increase our confidence that we have not missed genuine *kps* locus diversity, we inspected the 27 unassigned genomes with a complete set of *kpsEDCSMT* genes. All 27 were assigned a predicted K-type by kTYPr whole-genome mode. Six of those had a complete set of K-type specific genes that were located at different sites in the genome. We inspected Kaptive results for the remaining 21 genomes (Extended Data Table 11). We note the following observations:

- 3 are assigned to known degenerate loci (KL121, KL128, KL138) (Extended Data Fig. 5 and Extended Data Table 11)
- The remaining 18 give hits to genuine K loci but Kaptive reports low coverage, truncated genes, or fragmentation across different contigs or genomic loci.

These results are consistent with the presence of disrupted gene clusters in these genomes. As we note in the response to reviewer 1 (and in Extended Data Fig. 5), such disrupted/degraded clusters are common and may be due to biological processes (e.g. slip strand mispairing or indels) or sequence quality (e.g. poly-N tracts).

While these 27 genomes do not appear to present novel capsule clusters, we acknowledge that the kTYPr catalog is incomplete. For example, six clusters that are either absent or exceedingly rare in the 37,723 genomes used to assemble the kTYPr catalog, are present in the EC-K-typing DB, but not in the kTYPr catalog (see response to reviewer 1, point 1). We leave the opportunity to explore this and other new biology to users who can easily compare kTYPr flanking and whole-genome modes and Kaptive results, for more conservative or explorative analyses, respectively, on their own datasets and on all collections profiled in this study. Importantly, users can include new K locus diversity in kTYPr by adding custom HMM sets that define novel types.

Regarding the handling of cases where additional genes are present in a K locus, these cases will be apparent in the clinker visualization that is part of the kTYPr output. The clinker output compares the identified query cluster with the respective reference cluster, allowing users to identify potentially novel genes and unexplored K-type diversity as pointed out by the reviewer (see an example clinker comparison of K10 clusters in the response to reviewer 1, point 2).

Action points: we clarified proportions of incomplete and unassigned genomes rephrasing the main text (lines 301-310) and edited Extended Data Fig. 3A to increase clarity. We provide here a systematic assessment of the 27 incomplete and unassigned cases that the reviewer refers to. We clarified our strategy to characterize known and novel K-type diversity here, in response to reviewer 1, in the main text (lines 314-330) and in the new Extended Data Figure 5. Our RefSeq collection profiled both in flanking and whole-genome mode will be accessible at https://github.com/SushiLab/kTYPr_EcoEpidem upon publication.

6) I am unable to follow the numbers on L276-286 vs Fig S3A:

Fig S3A - 25,879 plus the other columns add to 37,689. The total should be 37,723

L279 - 37,723 minus 9,607 is 28,116 but the text states that 28,082 were not assigned

L281 - 28,082 minus 25,879 = 2,203 (which matches the number in the figure) but the text states that 1,436 are remaining

We thank the reviewer for pointing this out: K-type assignment was not depicted in the figure, generating confusion. We updated the figure to include K-type assignment and rephrased the main text to clarify (lines 301-310).

In addition, we would like to note that during the revision process we introduced minor improvements in our K-type catalog, which led to small changes in numbers (but in none of the general trends described). This is why the new numbers in the figure are slightly different from our first submission.

7) While structures are available for some capsules, authors should note in the text that these structures were determined using a different set of isolates and therefore the correlations between sequence and structure are not direct. This could potentially explain the issue with K6, and other cases where genes outside the capsule locus are suspected.

We thank the reviewer for making this important point. We note this in Results lines 247-250 and the discussion lines 582-589.

8) It is not clear why K74 was not directly compared with K16 in the MS analyses as the structures are clearly related. MS analyses should also be performed on K74.

We agree with the reviewer that a direct comparison of the K16 and K74 structures should be performed. And we made considerable efforts to obtain a K74 strain for this purpose but were unsuccessful. In this respect, we note that the K74 antigen was published in 1988 from the lab of Jann and Jann at the MPI in Freiburg ([https://doi.org/10.1016/0008-6215\(88\)84120-X](https://doi.org/10.1016/0008-6215(88)84120-X)). They reported that the strain was acquired from "Drs. I. and F. Orskov (Copenhagen)." No institutional address was provided, but the reference strains that Drs Orskov collected were transferred to and commercialized by Staatens Serum Institute, Copenhagen. However, we acquired all available reference strains from SSI, and K74 was not among them (<https://ssid.com/disease-areas/gastrointestinal/e-coli-portfolio/e-coli-strains/>). Similarly, there are no strains reported as K74 in NCTC (<https://www.culturecollections.org.uk/>) or CCUG (<https://www.ccug.se/>). According to our literature searches, there are no reports of an *E. coli* K74 strain in at least 25 years. We did also reach out to the MPI in Freiburg but they reported that the people who could know anything about those strains either passed away or are very old.

The K74 sequence used for the catalog was obtained from a publicly available sequence database.

9) The Kdo residue in K16 and K74 is alpha linked and would require an inverting GT. This is different to the other types with the GT99 protein, where Kdo is beta linked and requires a retaining GT. This could explain the sequence divergence seen in Fig S2 but conflicts with the statement made on L199-200. Please revise.

We acknowledge in the revised text that the α -linked Kdo in the K16 and K74 polysaccharides would require an inverting Kdo transferase (lines 219-220).

10) The authors should use established gene nomenclature for previously characterised loci where possible to acknowledge the work of others. Additionally, ref 27 and others should be referred to for previous structure correlations.

We have made every effort to acknowledge the work of others by citation and by use of established gene nomenclature in the manuscript and figures. In development of kTYPr we have used HMM names (e.g. in the form of K3_5, K3_6, K3_7) which may appear as locus tags in GenBank files generated by kTYPr, however, these are not in the format of gene names and are not intended to be understood as such. As suggested, reference 29 (reference 27 in the original submission) and references 26-28 are cited in the introduction for linking *kps* loci with established capsule serotypes (line 85).

11) Please define what criteria and cutoffs were used to call types for each genome. For MAGs, if a low cutoff was used, types could potentially be novel or inaccurately assigned.

We thank the reviewer for highlighting this lack of clarity. We have clarified in the Methods (lines 1075-1077) that our analyses are based on genomes fulfilling two criteria: (i) a complete set of essential common genes (*kpsEDCSMT*) and (ii) a complete set of serotype-specific genes. We have furthermore clarified in the text (lines 400-401) that the same stringent criteria were used for calling K-types in MAGs as in the genome collection.

We note that kTYPr itself does not “call types” or indicate genomes as “typeable”. We would like to draw the reviewer’s attention to the relevant section of Methods (lines 898-904) which we edited for increased clarity. The kTYPr report includes an assessment of (a) the completeness of two sets of common genes *kpsEDCSMT* and *kpsFU*, and (b) the completeness of the set of genes which define each K-type, as well as (c) the “accumulated bit score” for assessing the overall quality of each set. A separate set of columns reports the “predicted” or most probable K-type based first on completeness and second on the highest accumulated bit score for the serotype-specific gene set.

In response to a similar point raised by reviewer 2 (point 2), we also now include in the tool output the “accumulated bit score” normalized to the highest score detected for that K-type in our collection. We furthermore provide all internal cutoffs manually curated for each HMM (as described in lines 885) in our catalog in Extended Data Table 16.

Finally, any predicted cluster is extracted and provided to the user as a GenBank file, and as an interactive visual comparison with the respective reference cluster in HTML format. This approach provides the user with comprehensive information to explore partial matches and to choose custom criteria for calling K-types.

Action point: we provide all internal cutoffs for each HMM in Extended Data Table 16.

12) Fig 2 - Nucleotide sequence identity between loci needs to be shown. Namely because it is not clear for example how different K51 is from K108, as well as the following pairs: K113/K148, K18/K100, K53/K93, K1/K92, K16/K107. If these are pairs have the same sequence, this will affect typing results.

We thank the reviewer for raising this point. We note firstly that none of the clusters reported in Fig. 2 have the same sequence, all can be distinguished by their nucleotide and encoded protein sequences. Because KTYPr is based on protein sequence similarity and functional annotation, we report amino acid sequence similarity from 30%-100% in grey scale lines that connect vertically adjacent ORFs in Figure 2. We were unable to fit numerical pairwise sequence identities between loci in Fig. 2 and we have therefore provided comprehensive tables with both amino acid and nucleotide percentage identity as Extended Data Table 6.

Action points: we now provide the % nucleotide and amino acid sequence identities as Extended Data Table 6.

13) Table S1 – Granted that these genomes are integral to the manuscript, please provide the genome accessions in this table.

We agree. Genome accession numbers are now provided as an additional column in Extended Data Table 1. We added a relevant explanation on this in the Table caption and in the Methods (lines 822-823).

14) Table S2 – please provide further justification on why the serotype-referenced isolates were not used for the representative locus sequences. Table S2 is referred to in-text and Fig 2 legend for locus accession numbers. However, only genome accessions are listed. Please provide the precise locus location (contig accession number with base positions).

We note that the majority of the K-types defined in the kTYPr catalog are novel, and thus, there is no serotype-reference strain. For K-types where K antigen reference strains are available, some have disruptions in non-essential genes in the *kps* locus (e.g. in genes which are redundant in the *E. coli* genome) which are not expected to impact capsule structure. Two examples are:

- *kpsF* is disrupted in the K19 reference strain
- *galE* is disrupted in the K14 reference strain

The reference clusters are intended for display in figures and for annotation of K-specific ORFs. For this purpose we have chosen to use intact gene clusters and clusters that are most representative of each particular type. The justification and procedure used for assigning reference locus sequences is outlined in the revised manuscript (lines 924-927).

We agree with the reviewer that including the coordinates of the clusters is valuable. We have added the following columns to the table: “RefSeq complete accession”, “Record locus accession”, and “kps locus coords”.

Action point: The justification and procedure for selecting representative locus sequences has been clarified (lines 924-927) and Extended Data Table 2 now includes additional information regarding genome versions, record presenting the cluster, and the start and end coordinates of the *kps* clusters. Where to find this data has also been included on Figure 2 caption.

Minor points

Throughout the text and Fig legends, “capsule-positive” is used to describe genomes that have *kps* genes. While this is defined in some places, it is not defined in others. This can lead to inferring that genomes with Group 1 and 4 genes are capsule negative. Please revise to “*kps*-positive” in all instances.

Thanks for highlighting this. All instances of “capsule-positive” have been replaced.

Were the genomes screened for species-specific markers to confirm the taxon?

The genomes in the collection reported in updated Extended Data Tables 12-13, used for Figures 3-4 and Extended Data Fig. 6-9, were taxonomically profiled in the following way: 32,043 genomes from NCBI were assigned to *E. coli* according to BV-BRC which in turn relies on NCBI Taxonomy (PMID: 36350631). 827 genomes from a UTI study (PMID: 37904175) were identified as *E. coli* in the original study experimentally via MALDI-TOF MS and computationally via rMLST. The 2,762 MAGs from 25 published studies, reported in updated Extended Data Table 13, were taxonomically profiled using mOTUs 4 (PMID 39526369), which clusters genomes into species-level taxonomic units (mOTUs) based on universal single-copy marker genes. Only MAGs assigned to the *E. coli* mOTU mOTUv4.0_000063 were retained.

Action point: we added information on taxonomic profiling in the “Genome collection preprocessing and annotation” section in the Methods (lines 1069-1073).

L55-64 – Please define K-antigen as it is not always CPS i.e. K88 and others are protein

Thanks for highlighting this important point. We have added a sentence in the discussion which acknowledge the fimbrial antigens assigned K-types (K88 and K99) and suggest that future assignment of K-types is restricted to acidic polysaccharide antigens belonging to groups 1-4 (lines 544-547).

L58 – Many capsule types are poorly immunogenic due to molecular mimicry with host structures as reported in the literature. In addition, Ref 10 does not mention *E. coli*. Please revise

We revised the text accordingly and changed reference 10 to PMID: 24372337 (line 58).

L65 – Suggest use of zero possessive i.e. “*E. coli* capsular polysaccharides”

Done.

L67 – Not all E. coli O-antigens are synthesised by the Wzx/Wzy-dependent pathway. Revise to “...in a manner analogous to many E. coli O-antigens”.

Thanks for catching this. Revised accordingly.

L72 – move “(Fig 1A)” to line 70.

Done.

L72 – Revise to “Further sugars and/or constituents are added to the conserved oligo-b-Kdo base...”

We have revised this sentence as follows and hope this meets the reviewer’s satisfaction:

“K-type-specific proteins extend the conserved oligo- β -Kdo primer with further sugars and other constituents, producing the serotype-specific part of the K antigen.”

L74-L76 – Please add reference(s) for this statement

We have added this reference:

Whitfield, C. Biosynthesis and assembly of capsular polysaccharides in Escherichia coli. Annu. Rev. Biochem 75, 39–68 (2006).

L82-83 – Please include references for other tools including SerotypeFinder, ECTyper and SRST2

Thanks for catching the missing references. We have added references to ECTyper and SerotypeFinder (line 85). SRST2 does not seem relevant since we detected sequence types with MLST (<https://github.com/tseemann/mlst>) as described in Methods (line 1048-1049).

L88 – Sequencing was not reported in this work. Revise sentence to “We recently reported the genome sequences for...”

Done

L90 – “...diversity of Group 2 and Group 3 capsule biosynthesis...”

Done

L115 – Fig. 1A does not show a locus – remove reference to figure or add a generic representation of a kps locus with regions 1, 2 and 3 marked.

We removed the reference to the figure as suggested.

L115 – Move Table S1 citation after “70 strains”

The sentence has been revised accordingly.

L121 - Fig. 1A – Please define yellow shape as Kdo in legend or key

Thanks. This has been added to the figure legend.

L123 - Fig. 1B legend – “...integrating Group 2 and 3 sequences for serotype reference strains...”

Revised to “...integrating *kps* loci from K antigen reference strains...”

L139 - Was the K2a mutation confirmed, or could it be a sequencing error?

Yes, the Oxford Nanopore long-read sequencing results (which are prone to such errors) were confirmed by short-read Illumina sequencing.

L153 – italicise *kps*

Done

L161 – please name the common flanking genes

Common flanking genes are listed in Extended Data Table 15, referenced in Methods (lines 842-843)

L166 – “HMM-based K-type definitions” seems to be referring to locus designations rather than the K type or structure

The reviewer has correctly identified our intentionally broad use of the term “K-type” to refer to both a set of genetic characteristics (genotype), a serotype in some cases, and an inferred polysaccharide structure (even if it remains unknown). We note that, for 55 of the mentioned “K-types”, there is currently no serotype or polysaccharide structure associated with designated K-number, thus the set of HMM that we provide are in a very practical sense the only “definition” for these K-types. Similarly, in the serological era, K-types were assigned based on serological observations. As discussed above under point (1) this is in keeping with current usage of such terms in the literature for genome-based typing tools of surface antigens, e.g. for O-types (see ECTyper, reference 1).

Action point: To improve clarity, we now specifically refer to *kps* loci whenever strictly referring to genetic loci, and K-type whenever referring to K-type identified based on genome-, serotype- or polysaccharide-structure.

L199 – Please provide numbers for how many encode the GT99 vs GT107 proteins, rather than the collective total of 29.

Done

L277 – “...were assigned a known (previously catalogued) transporter-dependent K-type...”

We rephrased as “were assigned one of the transporter-dependent K-types in our catalog” (line 301).

L300 – complete or known kps locus?

The sentence was edited as follows: “A complete kps locus was detected in 26% of the genomes...” (line 344).”

L312 – what is meant by novel? Types K104-K163? Or those not characterised in previous sections?

Thanks for raising this. Yes, we are referring to K104-K163, which we now state clearly in the text (line 356). Furthermore, we have defined “novel K-types” earlier in the manuscript as “...those that were not cataloged in the serological era.” (lines 174 and 356).

L369 – Table S3 not Table 3?

Corrected (now Extended Data Table 12).

L422 – replace “types” with “loci”

Done.

L608-609 – This section is misleading as the bacterial strains and sequencing/assembly are published in another paper. Suggest removing or revising it to collating the genome set for analyses.

This section has been removed.

L613 – Please include that the catalogue was prepared initially using the reference strains then extended by analysing RefSeq genomes.

Revised accordingly (lines 822-823).

L619-620 -if sequences were dereplicated at 90%, shouldn't the -min-seq-id parameter be 0.9 not 0.98?
Same comment for line 629

We thank the reviewer for pointing this out, we corrected this.

L635 – specify tool used to perform protein clustering and the tool used for identification of IS

IS were manually removed from the protein catalog when identified, either based on gene name or annotation.

L637 – Were HMMs trimmed?

HMMs were not trimmed.

L718 – “were” > “where”

Done.

L797 – please specify the tool and criteria used to pull and collate the metadata (geographical regions and isolation sources) associated with all genomes.

Metadata were retrieved from the supplementary materials of the published studies and from associated NCBI BioSample records and further manually curated to harmonize categories. We now add this information at lines 1054-1055.

Fig 2 – For some structures, acetyl groups are shown above sugars and in others they are shown below. Some structures are very close together making it hard to distinguish which structure has the acetyl group e.g. K13_K22 vs K20. Please correct.

Thanks for highlighting this. We have carefully revised the figure placing carbohydrate modifications closer to the respective sugar and at a position that is consistent with Symbol Nomenclature for Glycans suggestion to have substituents arranged in a clockwise manner around the sugar.

Fig 2 – it is difficult to distinguish blue and black italics on orange genes. Please revise.

Thanks for highlighting this. We have changed all numbers to bold, non-italics and the blue numbers are now grey.

Fig S4E and S5H – please make colour scheme (at least for common types) the same as panel B.

Done.

FigS5A – Please define grey colour

We now specify in figure captions that grey indicates *kps*-negative genomes in this and all other relevant figure panels.

Fig S6A – bottom plot – what is meant by “-” in first column? Are these the genomes with O-types represented in <25 genomes

These are genomes that lacked the O antigen encoding locus according to ECTyper. We now clarify this in the figure caption (updated Extended Data Fig. 9).

Fig S7 – not provided?

We apologize for this mistake. This figure (please see below) was meant to support the ANI threshold choice for the genome dereplication of the NCBI collection in a previous version of our analysis where we used 98% ANI. For 778,526,070 all-vs-all ANI comparisons, we showed the frequency distribution (normalized per million pairs) using 1,500 bins for all the ANI values on the x-axis. We ultimately decided for a more conservative threshold (99%), closer to most used cutoffs to define strain-level boundaries by the field literature (PMID: 38171007, PMID: 30504855). We think that this analysis is therefore not relevant or helpful for the reader and we decided to remove it.

Action point: we accordingly removed the figure caption in the main text and rephrased the relevant Methods section (line 1057).

Reviewer #1

I thank the authors for positively answering all my suggestions.

We thank Reviewer #1 for the positive feedback and for the thoughtful and constructive suggestions throughout the review process.

I particularly appreciate Fig. S8 (maybe consider including as a main figure?) and the associated discussion about k-types and ST. A small edit will be needed on the figure as the bottom left corner of panel C is not readable.

We implemented these changes.

The rigorous work comparing kTYPr and the EC-K-typing database will also very valuable for the readers and users of these tools.

I have also reviewed the author's revisions and thorough responses to all reviewers, and I have no further questions.

Reviewer #3

I have only one further comment for the authors to consider. I suggest retrospectively adding the six novel types found in the Kaptive database to the kTYPr catalog for future use if this hasn't already been done.

We thank Reviewer #3 for the positive feedback. Following their recommendations, we have included the six K-types found in the Kaptive database to be profiled with the latest version of kTYPr.